# Risk–reward trade-off during carbon starvation generates dichotomy in motility endurance among marine bacteria

Johannes M. Keegstra [1] ✉, Zachary C. Landry[1], Sophie T. Zweifel [1], Benjamin R. K. Roller [2,3,4], Dieter A. Baumgartner [1], Francesco Carrara[1], Clara Martínez-Pérez[1], Estelle E. Clerc[1], Martin Ackermann[2,4,5] & Roman Stocker [1] ✉

Copiotrophic marine bacteria contribute to the control of carbon storage in the ocean by remineralizing organic matter. Motility presents copiotrophs with a risk–reward trade-off: it is highly beneficial in seeking out sparse nutrient hotspots, but energetically costly. Here we studied the motility endurance of 26 marine isolates, representing 18 species, using video microscopy and cell tracking over 2 days of carbon starvation. We found that the trade-off results in a dichotomy among marine bacteria, in which risk-averse copiotrophs ceased motility within hours ('limostatic'), whereas risk-prone copiotrophs converted ~9% of their biomass per day into energy to retain motility for the 2 days of observation ('limokinetic'). Using machine learning classifiers, we identified a genomic component associated with both strategies, sufficiently robust to predict the response of additional species with 86% accuracy. This dichotomy can help predict the prevalence of foraging strategies in marine microbes and inform models of ocean carbon cycles.

There is a profound dichotomy in ecological strategies among marine bacteria between oligotrophic and copiotrophic bacteria[1]. This dichotomy is associated with a suite of ecological and behavioural adaptations that allow oligotrophic bacteria to more readily survive in the more oligotrophic regions of the ocean[2] and allow copiotrophic bacteria to proliferate through feast–famine cycles driven by encounters with resource-rich hotspots[3,4]. At these hotspots, the strong metabolic activity of copiotrophic marine bacteria substantially contributes to marine carbon cycling and to the attenuation of carbon storage in the ocean, which ultimately affects atmospheric carbon levels[5]. Outside of these hotspots, low concentrations of labile carbon[6] makes copiotrophs experience strong growth limitation due to nutrient or energy starvation[7,8].

Flagellar motility[9] can be highly beneficial for navigating heterogeneous environments[10] but is associated with a high demand on cellular resources[11–13], especially during starvation. Reports on the effect of starvation on motility have been mixed. It has been shown that some bacteria increase their investment in motility with decreasing nutrient-limited growth rate[14], and some species have been reported to remain motile during starvation[15,16]. However, most experiments so far show that starvation hampers motility[17–22]. Despite the high energetic requirements, motility potentially brings great rewards in the marine environment, by enhancing the encounter rate with localized nutrient hotspots, such as phytoplankton cells[23] or organic matter particles[24], by $10^2$- to $10^3$-fold[25–28]. These hotspots provide marine bacteria with rich nutrient resources, meaning a successful colonization of a sub-mm

[1]Institute for Environmental Engineering, Department of Civil, Environmental and Geomatic Engineering, ETH Zurich, Zurich, Switzerland. [2]EAWAG - Swiss Federal Institute of Aquatic Science and Technology, Dübendorf, Switzerland. [3]Division of Microbial Ecology, University of Vienna, Vienna, Austria. [4]Department of Environmental Systems Science, ETH Zurich, Zurich, Switzerland. [5]School of Architecture, Civil and Environmental Engineering, École Polytechnique Fédérale de Lausanne (EPFL), Lausanne, Switzerland. ✉e-mail: keegstra@ifu.baug.ethz.ch; romanstocker@ethz.ch

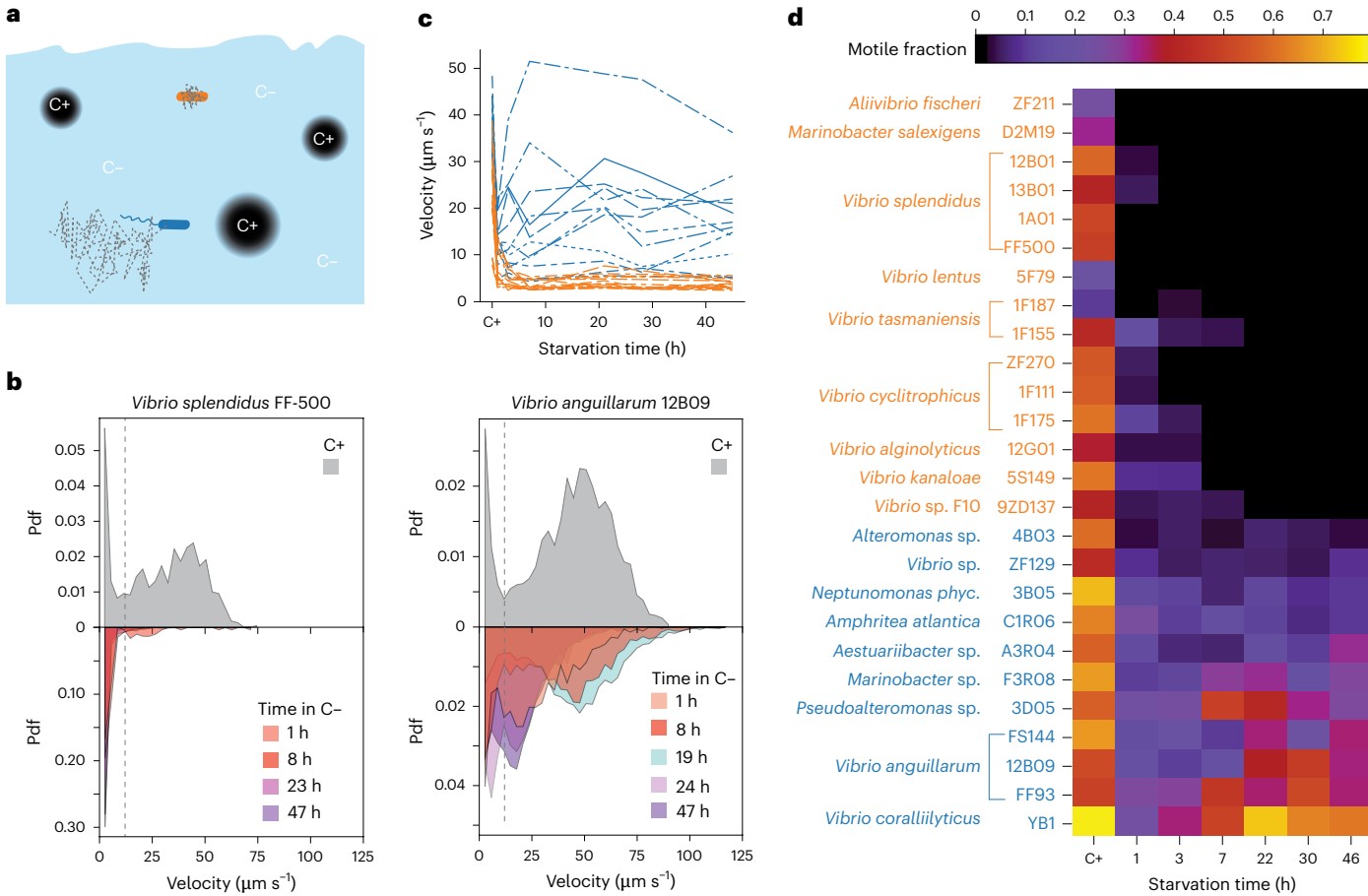

**Fig. 1 | Marine bacteria exhibit a dichotomy in their motility response to carbon starvation. a**, Marine bacteria often experience carbon starvation (C−) during the time between encounters with carbon-replete hotspots that support growth (C+, dark circles). During starvation, bacteria may opt to cease motility (orange) to conserve resources, or sustain motility (blue) to increase chances of encountering a hotspot. **b**, Distribution of cellular velocities in *V. splendidus* FF-500 (left) and *V. anguillarum* 12B09 (right) before starvation (C+; top) and at different times during carbon starvation (1 h to 47 h; bottom). Dashed grey lines mark the velocity of 12 μm s⁻¹, used to differentiate motile from non-motile cells (Supplementary Fig. 2b). Pdf, probability density function. **c**, Average cellular velocity of the population as a function of starvation time for 26 marine strains; 15 strains show a rapid decrease of velocity (orange), to on average 5 ± 2 μm s⁻¹, whereas 11 strains retain a high velocity (blue), with an average of 18 ± 9 μm s⁻¹. **d**, Fraction of motile cells (as given by the colour bar at top) for the 26 strains as a function of starvation time. C+ denotes the condition before starvation. Fifteen strains reveal a rapid decline of the motile fraction during starvation (orange species names), 11 strains show persistent motility during starvation (blue species names). A total of 580 video microscopy experiments were performed; the number of experiments for each condition (2–5 per strain) are given in Extended Data Fig. 2.

sized particle may lead to a manyfold increase in biomass[24,29]. This high potential search reward, combined with the risk of wasting limited cellular resources, makes bacterial motility under starvation subject to a risk–reward trade-off, and raises the question of which strategy is adopted by marine bacteria.

Here we report on the motility behaviour upon carbon starvation for 26 strains of 18 species of copiotrophic marine bacteria. We did not find a continuum of endurance timescales, but rather a behavioural split between species that cease motility within a few hours and species that retain motility for multiple days, revealing an ecological dichotomy among motile copiotrophic bacteria. This dichotomy reflects a different risk assessment of starvation by different bacteria: risk-averse foragers cease motility to conserve resources until conditions improve, whereas risk-prone foragers retain motility to enhance their chance of large search rewards.

## Results

### Behavioural split in motility endurance upon carbon starvation

We measured the motility response of different marine bacteria to carbon starvation. Carbon starvation was imposed experimentally

by growing the cells in carbon-replete marine broth (MB), then washing and placing the cells in carbon-depleted starvation medium. This procedure models, for example, the rapid loss of access to nutrients that cells experience when leaving a nutrient hotspot (Fig. 1a). We sampled cells immediately before washing and then at 1 h and ~3, 7, 22, 30 and 46 h after the onset of starvation. For every time point, we used video microscopy and cell tracking to quantify the cellular velocity (the velocity averaged over the cell's trajectory) of ~300 cells. Our measurements reveal a striking divergence in the motility response to starvation, even among closely related species. As an example between closely related species (see the phylogenetic tree in Extended Data Fig. 1), in carbon-replete medium, *Vibrio splendidus* FF-500 and *Vibrio anguillarum* 12B09 (previously known as *Vibrio ordalii*[30]) were both highly motile, with population-averaged velocities of 29 ± 18 μm s⁻¹ and 41 ± 21 μm s⁻¹, respectively (Fig. 1b, and Supplementary Videos 1 and 2). However, their motility upon entering carbon starvation was strikingly different. Within 1 h of starvation, the velocity of *V. splendidus* FF-500 diminished to 4 ± 4 μm s⁻¹, whereas the velocity of *V. anguillarum* 12B09 during the 2 days of starvation remained high, with an average of 31 ± 21 μm s⁻¹ (Fig. 1b, and Supplementary Videos 3 and 4). Experiments with an additional pair of strains from the same two

species showed similar results (Extended Data Fig. 2a). These observations show that bacterial species can have strongly divergent motility responses upon carbon starvation.

We performed these carbon starvation experiments to measure the motility of 26 strains from 18 species belonging to the Gammaproteobacteria class (Supplementary Videos 5–10). For each strain, we computed the population-averaged velocity (Fig. 1c) and the fraction of motile cells (Fig. 1d) as a function of starvation time (Methods). Following carbon starvation, the fraction of motile cells revealed a clear dichotomy: for some strains the motile fraction decreased rapidly to near zero, whereas for other strains it remained considerably above zero throughout starvation (Fig. 1d and Supplementary Videos 5–10).

To have an objective criterion to determine which strains retained and which strains relinquished motility, we computed the kernel density estimate (KDE) of the log-transformed motile fraction, averaged for all starvation times exceeding 1 h, for each strain. The KDE exhibits a bimodal distribution with a minimum at a motile fraction of 0.033, providing a clear separation into two classes (Extended Data Fig. 2c). We used this criterion to separate the motility response of each strain into two classes: 15 out of 26 strains had a motile fraction below this threshold upon starvation (on average $0.01 \pm 0.01$; unless noted otherwise computed as the average $\pm 1$ s.d. of the average value per strain), whereas the remaining 11 out of 26 strains retained a higher motile fraction than this threshold (on average $0.23 \pm 0.16$) (Fig. 1d). By contrast, in a carbon-replete environment, the 26 strains exhibited no dichotomy in motility behaviour even though motility-retaining strains were on average more motile than motility-relinquishing strains (Extended Data Fig. 1d–f, Fig. 1d and Supplementary Table 1). We propose to call the motility-retaining response 'limokinetic' (from the Greek $\lambda\iota\mu\acute{o}\sigma$ meaning 'starvation') and the motility-relinquishing response 'limostatic'.

To test the robustness of the observed dichotomy, we repeated experiments by using a different treatment to impose carbon starvation. Instead of washing the cells, we measured the motility of cells during nutrient-limited stationary phase. For 12 strains, we compared the time-averaged motile fraction for each strain in the stationary phase to that obtained in the previous experiments in starvation medium, and found that these were highly correlated (Extended Data Fig. 3, Pearson's $\rho = 0.91$). Furthermore, the classification into limokinetic and limostatic strains (based on the motile fraction criterion) was the same under the two treatments, with a single exception (*Alteromonas* sp. 4B03). Together, these results indicate that the loss of motility is not specific to our starvation medium, and the dichotomy is robust to differences in the mode in which carbon starvation is imposed and is primarily a species-specific trait.

### Differential flagellar loss indicates commitment to non-motile and motile lifestyles

Loss of flagellar filaments during nutrient limitation has been reported for other bacteria[21,22], prompting us to investigate the flagellation of limokinetic and limostatic strains during carbon starvation. We first measured the flagellation of 11 strains in nutrient-replete conditions. The average fraction of cells with 0 or 1 flagella was 97.2%, indicating that the dominant mode of flagellation was a single polar flagellum (Extended Data Fig. 4), as is common for marine bacteria[31]. This is also consistent with bacteria performing run-reverse or run-reverse-flick random walks in our tracking experiments (Supplementary Videos 1–10), the hallmark of single flagellation[32,33].

We then used scanning electron microscopy (SEM) to quantify the flagellation in response to starvation in two limostatic strains and three limokinetic strains (Fig. 2a). The flagellation during starvation revealed a strong difference between limostatic and limokinetic strains. In a carbon-replete environment, the flagellation was similar in the two classes, with an average flagellated fraction of $0.65 \pm 0.11$ and $0.69 \pm 0.26$ for limostatic and limokinetic strains, respectively. After 24 h of starvation, the fraction of flagellated cells was only $0.04 \pm 0.01$

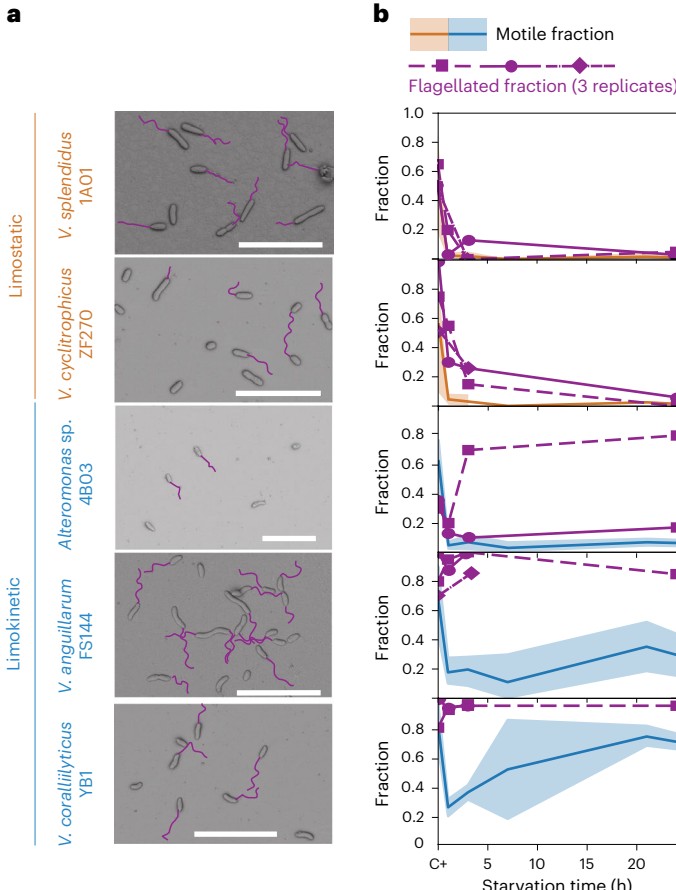

**Fig. 2 | Limokinetic and limostatic strains differ in the prevalence of flagellation upon starvation. a**, Representative SEM images from 5 strains, 2 limostatic and 3 limokinetic, from exponentially growing cultures (that is, before the onset of starvation). Flagellar filaments are highlighted (purple). Scale bars, 10 μm. **b**, Fraction of flagellated cells (purple lines) determined from SEM images as a function of starvation time, for 2–3 replicates per strain (symbols). The number of cells imaged by SEM per strain and time point was at least 34. For comparison, the average fraction of motile cells for the same strains are shown (orange for limostatic, blue for limokinetic; data from Fig. 1d) along with 95% confidence intervals (CI; shaded areas).

in limostatic strains, whereas in limokinetic strains it was $0.75 \pm 0.20$ (Fig. 2b). The average filament length (3.9 μm) did not show a difference between the two classes, or between starving or growing conditions (Extended Data Fig. 4), indicating that during flagellar loss, the filaments are lost in their entirety. Additional experiments showed that the flagellar loss was not due to shear stress[34,35] (Extended Data Fig. 4). Overall, this shows that during starvation, limostatic strains lose flagellar filaments, whereas limokinetic strains retain them.

Comparing the fraction of flagellated cells with the fraction of motile cells shows that bacteria can cease motility without losing flagellar filaments: for all strains the fraction of flagellated cells was higher than the fraction of motile cells (Fig. 2b). The difference is especially strong for the limokinetic strains *V. anguillarum* FS144 and *V. corallilyticus* YB1, where the respective flagellated fractions were 3.8 and 1.9 times larger than the motile fractions (Fig. 2b). This suggests that limokinetic bacteria have the ability to pause motility, by temporarily stopping flagellar rotation, without flagellar loss.

Flagellar loss prevents bacteria from rapidly responding when conditions improve, as flagellar synthesis is slow: even a relatively short flagellar filament of 1.5-μm length requires at least 30 min to be synthesized[34,36,37]. Indeed, additional experiments on strains starved for 24 h revealed that limostatic strains only recovered motility 30–60 min

after nutrient addition (Extended Data Fig. 5). These observations indicate that limostatic strains not only stop swimming but also commit to a non-motile lifestyle.

## Limokinetic strains convert biomass into energy to fuel motility

During starvation, the synthesis of new motility machinery diminishes and the dominant cost of motility is the operation of the flagellar motor for propulsion[11]. The average power spent on motility per cell can be estimated as $\epsilon = f_s\Omega/(\eta N)\sum_i v_i^2$, where $v_i$ is the average swimming velocity of motile cell $i$, $N$ the number of motile cells, $f_s$ the fraction of motile cells, $\eta$ the efficiency of the flagellum (2%)[38] and $\Omega$ the resistance coefficient of the bacterium including its flagellum ($\Omega = 4.1 \times 10^{-8}$ Ns m$^{-1}$ (ref. 39)). We used the motile fraction and the swimming velocities (Fig. 1d, Extended Data Fig. 6a,b and Supplementary Table 1) to compute the power spent on motility per cell for each strain as a function of time (Fig. 3a). The energy expenditure of limokinetic strains on average decreased more than 3-fold during starvation (from $4.1 \pm 0.4 \times 10^4$ ATP s$^{-1}$ before starvation to $1.2 \pm 0.4 \times 10^4$ ATP s$^{-1}$ during starvation, mean $\pm$ s.e.m.), assuming a conversion factor of $8 \times 10^{-20}$ J ATP$^{-1}$ (ref. 11). The maintenance energy flux during starvation is estimated at $1 \times 10^4$ ATP s$^{-1}$ per cell[40,41] (this estimate is for *E. coli*; the higher starvation survival rates of some marine bacteria[42,43] do suggest that much lower maintenance energies are possible). Hence, by remaining motile, limokinetic strains at least double their energy requirements during carbon starvation compared with limostatic strains.

We hypothesized that cells use internal energy sources to fuel motility, sacrificing part of their biomass to generate energy[44]. To test this hypothesis, we performed starvation experiments for 7 days with 3 limostatic strains and 3 limokinetic strains, which were motile over the 7-day period (motile fraction of $0.27 \pm 0.20$, average swimming velocity $40 \pm 7$ μm s$^{-1}$, Fig. 3b). We measured the biomass of six strains using optical density (OD)[45] during the starvation experiment. The 3 limokinetic strains lost on average $62 \pm 3\%$ of their biomass over 7 days. In contrast, the biomass of the limostatic strains remained approximately constant ($99 \pm 16\%$) (Fig. 3c) during the same period. A linear fit of $\text{OD}_n = 1 - \gamma t$ over all individual measurements on limokinetic strains yielded a good fit ($R^2 = 0.88$), with a biomass decay rate rate of $\gamma = 0.094$ day$^{-1}$ (95% CI: [0.085, 0.103]). The same fit yielded $\gamma = -0.011$ day$^{-1}$ (95% CI: [$-0.025$, 0.003], $R^2 = 0.04$) for the limostatic strains. These data indicate that motility endurance was associated with a biomass loss of 9.4% per day.

We confirmed that the biomass decrease is due to a conversion of biomass to energy, rather than a decrease in the number of cells. Flow cytometry measurements of the cell number during the 7-day starvation experiment revealed that the number of cells increased or remained constant compared to the onset of starvation (Fig. 3d). Alternative estimates based on colony counts and the number of cell tracks confirmed that the number of cells did not decrease during starvation (Extended Data Fig. 6). The increase in cell number was probably due to reductive divisions, a well-known starvation response where the population biomass is redistributed over more, but smaller, cells[7,22]. With a decreasing population biomass and non-decreasing cell number, this implies that on top of any reductive divisions, the conversion of biomass to energy in limokinetic strains reduces the cellular biomass in limokinetic strains.

To further investigate the biomass loss at the single-cell level, we measured the dry mass distributions from quantitative phase imaging (QPI) on individual bacterial cells[46,47]. We compared the biomass of limokinetic *Vibrio coralliilyticus* YB1 and limostatic *Vibrio cyclitrophicus* ZF270. During growth we found respective average biomasses of $308 \pm 100$ fg (mean $\pm$ s.d. of 3 biological replicates; 1 fg = $1 \times 10^{-15}$ g) and $333 \pm 60$ fg (Extended Data Fig. 6f). We then measured the biomass at -5, 76 and 125 h after starvation onset (Fig. 3e). At 5 h after starvation onset, the biomasses of YB1 and ZF270 were similar ($P = 0.9$, Tukey's

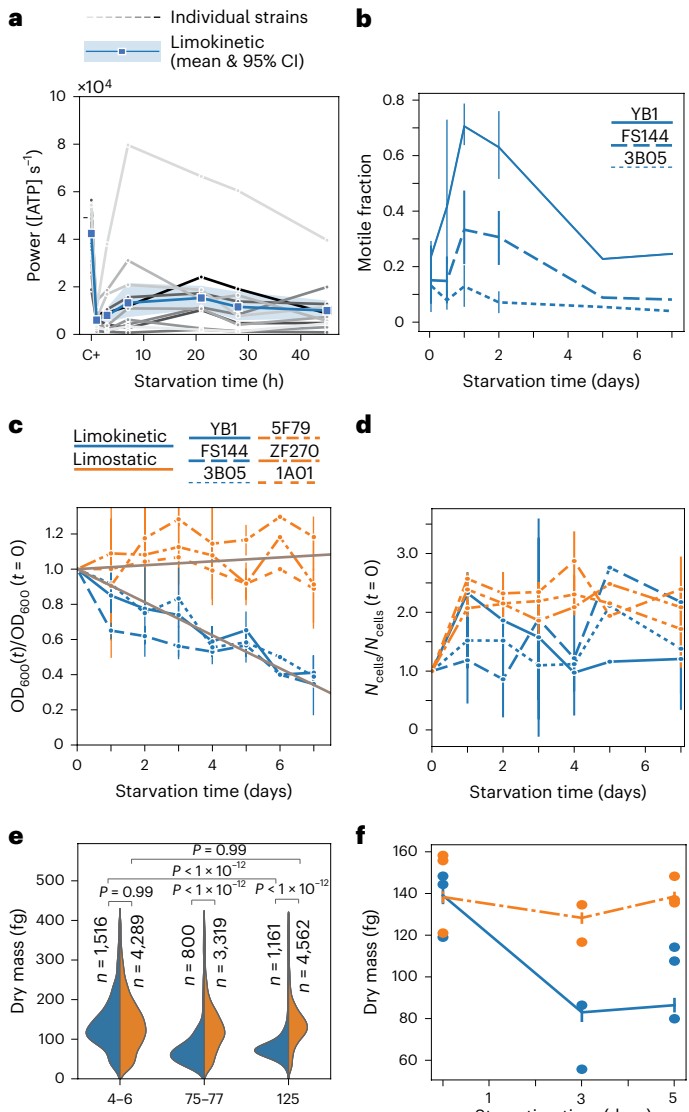

**Fig. 3 | Limokinetic bacteria convert biomass to energy to fuel motility.**
**a**, Estimated motility power requirement per cell for each of the 11 limokinetic strains (grey lines) and average of the 11 limokinetic strains (blue, 95% CI shown as shaded area) as a function of starvation time for 2 days of starvation. **b**, Fraction of motile cells as a function of starvation time for 3 limokinetic strains. Shown are the mean $\pm$ 95% CI (vertical lines). Number of biological replicates for $t > 2$ days is 1 (FS144) or 2 (3B05 and YB1). For number of replicates for $t < 2$ days, see Extended Data Fig. 2. **c**, Optical density as a function of starvation time, normalized by the optical density at the onset of starvation ($t = 0$), for 3 limostatic (orange) and 3 limokinetic (blue) strains for 7 days of starvation. Shown are the mean $\pm$ s.d. (vertical lines) of 3 independent experiments. Different strains are denoted by different line types. Grey lines indicate linear fits to the change in optical density of limokinetic and limostatic strains, with respective slopes of $-0.094$ day$^{-1}$ and $+0.011$ day$^{-1}$ (see main text). **d**, The number of cells $N_{cells}$ after prolonged starvation, normalized by the number of cells at the onset of starvation ($t = 0$), for 3 limostatic (orange) and 3 limokinetic (blue) strains for 7 days of starvation. Strains and error bars as in **c**. **e**, Single-cell dry mass distributions for YB1 (blue) and ZF270 (orange) as measured using quantitative phase imaging. Number of cells per condition ($n$) and statistical significance using a two-sided post hoc Tukey's HSD test are indicated. **f**, Average dry mass of a population (solid lines) as a function of starvation time for ZF270 (orange) and YB1 (blue). Error bars denote the 95% CI on the average of all the single-cell data, and the circles denote the average of a single biological replicate.

honestly significant difference (HSD)) at 139 fg and 138 fg, respectively. The reduction in dry mass compared with the dry mass during growth indicates that the cells have engaged in at least 1 reductive division

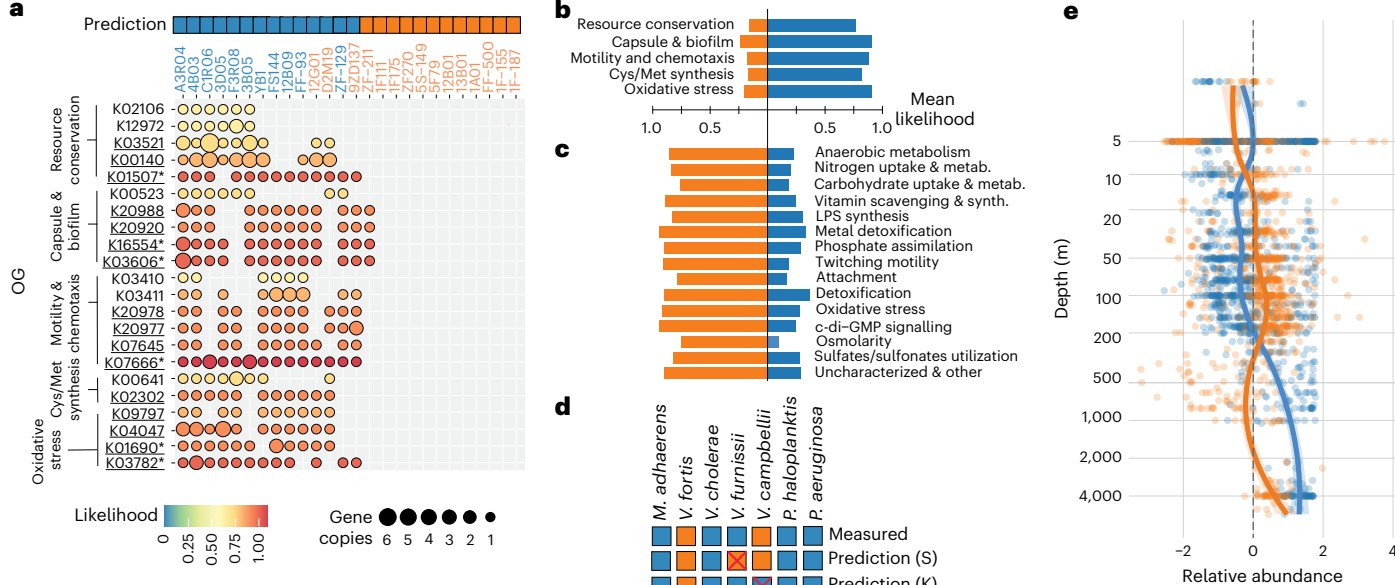

**Fig. 4 | Genomic basis of the limokinetic–limostatic dichotomy. a**, Bayesian classifier for the prediction of limokinetic behaviour. Top: classifier prediction of the limokinetic (blue) and limostatic (orange) behaviour for all strains. Strain names are colour-coded according to their experimentally determined classification (Fig. 1). Bottom: prevalence of orthogonal groups (OG) associated with a limokinetic response for both limokinetic and limostatic strains, as obtained by RFE (Methods) and clustered into 5 functional categories (Supplementary Note 3). Circles indicate the gene copy number of each OG (size) and the probability of association with the limokinetic response (colour). Underlined OGs indicate significance corrected for phylogeny of $P < 0.10$, *$P < 0.05$ (full list of $P$ values in Supplementary Table 3, Methods and Supplementary Note 4). **b**, Mean likelihood averaged over all OGs in each functional category for both limokinetic (blue) and limostatic (orange) strains as predicted by the classifier for the limokinetic response (**a**). **c**, As in **b**, but for a classifier based on genes associated with the limostatic response (classifier features in Extended Data Fig. 10). **d**, Prediction and measurement of the motility response to starvation based on the genomic classifiers for 7 strains not included in classifier training (squares) for both the limokinetic (K) and limostatic (S) classifiers. Red crosses indicate that the prediction deviates from the experimental result. **e**, Predicted relative abundance (z-score) of limokinetic (blue) and limostatic (orange) taxa as a function of depth, computed by applying the classifiers to 1,038 field samples from the Ocean Microbiomics Database[51]. For limostatic taxa, the z-score of individual samples (dots) was computed as the abundance of limostatic taxa after subtraction of the depth-averaged abundance and normalization to 1 s.d. The same procedure was applied for limokinetic taxa. For both strategies, a moving average (solid lines) computed with a locally estimated scatterplot smoothing (LOESS) filter using a window of 2/3 of the data is also shown with 95% CI (shaded area). The relative abundances of limokinetic and limostatic taxa are anti-correlated with depth (Pearson's $\rho = -0.79$).

during the first hours of starvation. However, after 5 days of starvation, the average dry mass of limostatic ZF270 did not significantly change ($P = 0.9$, Tukey's HSD), but the dry mass of limokinetic YB1 decreased to 86 fg ($P < 0.001$, Tukey's HSD) (Fig. 3e). Thus, the limokinetic strain lost on average 54 fg per cell (Fig. 3f), corresponding to ~11 fg cell$^{-1}$ day$^{-1}$. Such a biomass loss would be insurmountable for marine oligotrophs, with typical cell mass of 20 fg[48]. For copiotrophs, however, it represents a daily loss of only ~11/139 = 8% (close to the estimate based on population biomass loss).

Additional experiments allowed us to exclude three alternative energy sources for motility. First, we considered the recycling of necromass[49,50]. Live/dead staining showed that the fraction of dead cells was comparable between the two classes ($0.07 \pm 0.06$ and $0.11 \pm 0.07$ for limostatic and limokinetic strains, respectively) when averaged over the week of starvation (Supplementary Note 1 and Extended Data Fig. 6). Given the small difference in death rates, and considering that necromass recycling is typically inefficient (10–20%)[50], this means necromass recycling does not represent a large energy source for motility in our starvation experiments. Second, we found that the energy source is not photonic in nature, as limokinetic strains lack rhodopsin genes and remained motile when starved in the dark (Supplementary Note 1 and Extended Data Fig. 6g,h). Third, we excluded the effect of any residual nutrients in the starvation buffer, by showing that there was no negative dependence of the motile fraction on cell concentration (Supplementary Note 1 and Extended Data Fig. 6i).

The biochemical nature of the biomass conversion remains to be determined. Assuming all converted biomass is stored as glucose

(yielding ~30 ATP per molecule), 11 fg day$^{-1}$ would yield an energy flux of $1 \times 10^4$ ATP s$^{-1}$, close to the average motility power requirement of limokinetic strains during starvation (Fig. 3a), indicating that biomass conversion can fuel bacterial motility for several days. Fluorescence staining of storage compounds indicated no significant accumulation of polyphosphate, but a potential role for polyhydrobutyrate (PHB) to act as an energy source in limokinetic strains (Supplementary Note 2 and Extended Data Fig. 7).

**The genomic basis of the limokinetic and limostatic lifestyles**

We investigated the genomic basis of the difference between limokinetic and limostatic behaviours using assembled genomes of all strains to identify further differences between limokinetic and limostatic strategies and to predict the motility behaviour under starvation in other bacterial species. We constructed a Bayesian classifier, selecting genetic features that are associated with limokinetic behaviour through recursive feature elimination (RFE; Methods). The classifier was able to separate the two behavioural classes with good accuracy (88%, Fig. 4a and Supplementary Fig. 3), defined as the fraction of correctly predicted strains.

The classifier relies on a set of 22 orthologous groups (OG), or genes with conserved function, associated with a limokinetic response to carbon starvation (Fig. 4a). We grouped the genes into 5 functional categories on the basis of their annotated function (Supplementary Note 3). Genes selected by RFE include mechanisms for resource conservation ($N_O = 5$ genes, average likelihood $\mathcal{L} = 0.76$), capsule and biofilm formation ($N_O = 5$, $\mathcal{L} = 0.91$), regulatory elements of motility

and chemotaxis ($N_O$ = 6, $\mathcal{L}$ = 0.88), cysteine/methionine synthesis ($N_O$ = 3, $\mathcal{L}$ = 0.82) and oxidative stress response ($N_O$ = 3, $\mathcal{L}$ = 0.91) (Fig. 4b and Supplementary Note 3). Despite the high associated likelihood of oxidative stress in the environment, we found no significant differences in oxidative stress sensitivity between limokinetic and limostatic strains (Extended Data Fig. 8 and Supplementary Note 3).

We also trained the classifier in the inverse direction, selecting for genes that are associated with a limostatic response to starvation. This yielded a much larger list of genes (121) in a range of cellular functions (Fig. 4c), although with similar accuracy (88%) to the limokinetic classifier. Comparing these genomic signatures suggests that the limokinetic/limostatic dichotomy may be connected to other traits associated with oxidative stress defence, metabolism, uptake and surface-associated lifestyles (Fig. 4b,c).

We used the classifiers to predict the motility response to starvation of 7 additional marine strains not included in the training of the classifiers. The limokinetic and limostatic classifiers each predicted all but one correctly (86%) (Fig. 4d). Interestingly, the classifier predictions for the enteric species *Escherichia coli* and *Salmonella typhimurium* were ambiguous in that these strains were predicted to be both limokinetic and limostatic. In experiments, their motile fraction decreased more gradually (11–14 h) compared with the limostatic marine strains (2.4 h, $P < 1 \times 10^7$, $t$-test) (Extended Data Fig. 10). This contrast with enteric bacteria indicates that the dichotomy we have described is a feature of marine bacterial communities: the extent to which it may occur in other microbiomes will require dedicated investigation.

Finally, we used the Global Ocean Microbiomes dataset[51] in combination with our classifiers to predict the prevalence of the two strategies among assembled metagenomes in the ocean (Methods), as the classifier was originally trained and tested using gammaproteobacterial taxa, and we limited our prediction to this group. Limokinetic taxa were predicted to dominate in 97.3% of 1,038 field samples (Extended Data Fig. 10). Certain environments may favour a limostatic strategy, as indicated by the fact that the samples with predicted limostatic dominance (2.8%) all come from the euphotic zone (geometric mean depth of 70 m). The relative abundances of limokinetic and limostatic taxa as a function of depth are anti-correlated (Pearson's $\rho = -0.79$, Fig. 4e), suggesting the presence of environmental variables that affect the abundance of both strategies. This could, for example, be due to the concentration of dissolved and/or particulate nutrients, yet more work is needed to determine the environmental drivers of the prevalence of one versus the other strategy.

## Discussion

Copiotrophic marine bacteria contribute to the marine carbon cycle by remineralizing a large fraction of the carbon stored in sinking marine particles before they reach the ocean floor[52,53]. Since these particles are sparse, the bacterial contribution to particle degradation depends not only on their degradation activity but also on their ability to localize and colonize the particles (Supplementary Discussion 1). However, bacterial behaviours are rarely explicitly included in oceanic carbon flux models[54], making these models less predictive. This is in part due to the challenge of accounting for the enormous diversity among microbes in models. Dichotomies are widely used as simplifying principles to help understand the daunting diversity of microbes in natural environments, permitting generalization across traits from behaviour to cell physiology. Our results reveal an important dichotomy that separates motile copiotrophs into limokinetic and limostatic species: we propose this to be a useful concept for more explicitly including microbial behaviour in models of marine particle dynamics and microbial ecology overall.

Traditionally, bacterial motility and chemotaxis have been understood as strategies to enhance foraging[11,55], particularly in nutrient-poor environments[56,57], yet recent work has emphasized the benefit of motility in nutrient-replete environments[58–60]. Limostatic strains appear to use motility to disperse and colonize hotspots only during growth, when motile fractions are high. This strategy may be especially effective under algal bloom conditions, when the number of hotspots is high and so is the background level of dissolved organic matter, alleviating starvation. Limokinetic species probably also use motility for this purpose, but unlike limostatic species, we propose that they also use motility to actively search for hotspots in oligotrophic environments, even at the expense of sacrificing a sizeable fraction of their biomass to fuel motility, which could reduce typical search times from months to a day (Supplementary Discussion 2). Search times could be even further reduced if cells would suppress reorientations, but our data show no indication for this (Supplementary Discussion 3). Chemotaxis may add a further reduction in search times, but less than the boost from random encounters as gradients do not extend far beyond the particles (Supplementary Discussion 4).

In oligotrophic environments, limostatic copiotrophs will cease to be motile and conserve biomass until conditions improve again. Marine bacteria have been observed to survive starvation for periods of up to several months[42], suggesting that limostatic strains could be specialists in overcoming large temporal intervals of oligotrophy, whereas limokinetic strains could be specialists in overcoming larger spatial distances in oligotrophic environments. While certain environments will favour one or the other phenotype (Fig. 4), some of the limokinetic and limostatic strains studied here were isolated together (Supplementary Table 1), suggesting that the two behaviours can co-exist.

Testing our predictions of the motility in natural environments requires more direct observations of motility from the field[11]. Our current prediction of prevalence of the limokinetic strategy stands in contrast to the findings from most laboratory-based studies predicting motility loss upon nutrient depletion[17–22]. Direct measurements of motility in field samples report variable fractions of motile cells (<10% up to 70%)[57,61,62] but are limited to coastal surface waters where the concentration of dissolved nutrients is probably higher. Therefore, a more systematic mapping is required that extends the horizon of motility sampling to the open ocean as well as the ocean interior.

Our results show that limostatic strains lose flagella, but limokinetic strains mostly retain them. The flagellar retention confirms that it is not necessary for marine bacteria to cease motility and indicates pausing behaviour within limokinetic populations (Supplementary Discussion 5). Therefore, flagellar loss in limostatic strains[21,63] seems wasteful from a resource perspective. There must thus be non-energetic reasons for the ejection of flagella under starvation. One example could be the avoidance of predation, which plays a significant role in oligotrophic environments[64]. Motility can affect predation by increasing encounter rates with predators[65,66], but it is also possible that even the mere presence of a flagellum could increase predation risk, for example, by bacterivores[67] and phages[68,69]. Therefore, it is possible that cells eject their flagella to decrease predation risk. This indicates that the dichotomy between limokinetic and limostatic behaviour is shaped not only by energetics, but probably also predation pressure. Our results highlight a dichotomy in bacterial motility behaviour that results from a risk assessment between the anticipated biomass gain of motile behaviour and the biomass loss due to conversion to energy and possibly predation. The dichotomy serves as a simplifying principle that can help predict the ecological and biogeochemical functions of marine microorganisms in the face of their astounding degree of diversity.

## Methods

### Bacterial cell culture and starvation protocol

Cells were inoculated from a frozen (−80 °C) glycerol stock and grown overnight in 100% 2216 Marine Broth (BD Difco, Fisher Scientific, hereafter 'MB') at 27 °C on a rotary shaker (200 rpm). On the day of

the experiment, cells were diluted 1/100 into half-strength MB with 50% artificial seawater (Instant Ocean, Aquarium Systems, hereafter 'ASW'). After 3.5–4 h, the cultures reached mid-exponential phase (OD 0.1–0.5) and were collected by centrifugation (5,000 × *g* for 6 min). Pelleted cells were resuspended in starvation medium consisting of f/2 minimal medium without carbon (made by supplementing ASW with nitrogen, phosphorous, trace metals and vitamins (Provasoli-Guillard f/2 Media kit, NCMA), with added 1 mM NH$_4$Cl). This washing protocol was repeated three times, after which bacterial cells were diluted ten-fold compared with the original culture (leading to a cell concentration of ~10$^7$ cells per ml) and placed in a shaking incubator (175 rpm) at room temperature for the duration of the experiment. Bacteria were sampled from the medium immediately before washing the cells and at 1 (1), 2–4 (3), 5–9 (7), 19–24 (22), 28–32 (30) and 43–48 (46) h after the washing protocol started, where the number in brackets refers to the weighted average of each time window, rounded to 1 h, that was used for averaging over multiple experiments. Bacteria were observed within 15 min after sampling (their motility parameters were relatively stable during this period of time, Supplementary Fig. 1c). The optical density of bacterial cultures was measured with a cuvette-based spectrometer (WPA Biowave Cell Density Meter, Biochrom) on samples that were starved as described above but without the final dilution step (leading to an OD of 0.05–0.4, corresponding to ~10$^8$ cells per ml).

## Choice of bacterial strains

The strains used in experiments originated mainly from two principal collections. First, a collection of Vibrionaceae isolated off the Massachusetts coast[70], which has been extensively characterized for antagonistic interactions[71], colonization–dispersal behaviour[72] and alginate degradation[73]. Second, a collection of coastal seawater isolates associated with chitin particles[29]. We also included *Vibrio coralliitycus* YB1, a highly motile strain isolated from corals[74]. Using publicly available genomes, we selected those likely to be motile, on the basis of their number of motility and chemotaxis genes. Of the 107 available strains, we selected 36 strains to test for motility and growth, some of which were from the same species to encompass intraspecies and interspecies variation, and all with both chemotaxis and motility genes. Of the 30 remaining strains (4 strains did not grow in marine broth and 2 strains did not show motility during growth in marine broth), we randomly selected 26 strains to be used in this study.

The following species were selected to test classifier predictions (but were not used to train the classifier): *Vibrio fortis* KT626460, isolated from a healthy coral[75]; *Pseudoalteromonas haloplanktis* ATCC 700530, a model organism for chemotaxis studies in marine bacteria[76]; *Vibrio cholerae* C6706 was a gift from K. Drescher (U. Basel); *Vibrio campbellii* BB-120 (ATCC BAA-1116) was a gift from K. Jung (LMU, Munich); *Vibrio furnissii* DSM 14383 (NCTC 11218) was obtained through the German Collection of Microorganisms and Cell Cultures (DSMZ); *Marinobacter adhaerens* HP15 is a model organism for algae–bacteria interactions[77] and a gift from M. Ullrich (Jakobs University, Bremen). The HP15 strain contained a YFP-encoding plasmid but was grown and measured as the other, non-fluorescent bacteria.

## Microscopy and cell tracking

Cell samples of 45 µl were placed in the centre of a chamber (created by fixing a coverslip on a standard microscopy slide separated by silicone rubber of 1 mm thickness) and observed mid-plane using phase-contrast microscopy (Nikon) with a ×20 (0.45 NA) air objective (S Plan Fluor ELWD, Nikon). For very high cell densities and very low densities, ×10 (0.30 NA) and ×40 (0.60 NA) objectives were used, respectively. Videos recorded at ×40 were processed with a fast radial symmetry transform algorithm to remove diffraction rings[78] before applying the tracking routine. Videos with acquisition rate of 25–30 frames per second were recorded using a CMOS camera (ORCA Flash 4.0, Hamamatsu) for 30 s, at a resolution of 2,044 × 2,048 pixels

(0.326 µm pixel$^{-1}$ for ×20). Cell tracking was performed using TrackPy (v.0.4.2 and v.0.5.0)[79] after removing the background from each image by subtracting the median image computed over the entire video (Supplementary Fig. 1). In the analysis, a maximum displacement per frame of 31 pixels (corresponding to a swimming velocity of ~200 µm s$^{-1}$) and minimum separation between particles of 51 pixels were allowed. Trajectories shorter than 15 frames were removed from the analysis. Trajectories were then corrected for drift and cell positions were averaged over a time window of 5 frames in the calculation of velocity. Cellular velocity was defined as the velocity averaged over its trajectory, and the population-averaged velocity is the mean cellular velocity of a population. Cells with a cellular velocity lower than 12 µm s$^{-1}$ were classified as non-motile (Fig. 1a and Supplementary Fig. 1b). A velocity of 12 µm s$^{-1}$ corresponds to an approximate apparent displacement of 1 pixel between frames due to diffusion and/or localization inaccuracy. For each frame *i*, the number of motile ($N_{m,i}$) and non-motile ($N_{s,i}$) cells was determined. The motile fraction was then defined as $(1/T)\sum_i N_{m,i}/(N_{s,i} + N_{m,i})$, where *T* is the total number of frames in the video. The average swimming velocity was defined as the average cellular velocity of all motile cells. Videos with a motile fraction lower than 0.075 were inspected and corrected manually. Reorientation events were detected as described previously[32,80]. First, cellular positions were processed with a second-order Savitzky–Golay filter[81] with a time window of 5 frames to compute the angle and velocity between frames. For each trajectory, reorientation events were identified as time points at which both (1) the absolute change in angle exceeded 25° and (2) the velocity was lower than 75% of the average velocity of the trajectory. The minimal time between two reorientation events was limited to 2 frames (60–80 ms). The run time was defined as the time between detected reorientation events. The first run (from the start of the trajectory to the first event) and the last run (from the last detected event to trajectory length) were used as lower bound estimates of the run time. The reorientation frequency per cell was calculated as the inverse of the mean average run time per cell. To prevent detection of spurious reorientation events in slowly moving cells, the analysis was only applied to trajectories with a minimum length of 30 frames and a minimum velocity of 12 µm s$^{-1}$ based on average-filtered positional data with a time window of 9 frames. Data analysis was performed in Python (v.3.7 or newer) and visualization was performed using the packages Matplotlib (v.3.5.0) and Seaborn (v.0.11.2).

## Phylogenetic tree construction

A phylogenetic tree was constructed using Phylophlan (3.0)[82]. Phylophlan was run according to instructions using all reference genomes for all 4,788 strains as the initial dataset. Reference genomes were downloaded from RefSeq. Three outgroups were also added from RefSeq (GCA 000012345, GCA 000168995 and GCA 002355955). Additional references were added using 'phylophlan_get_reference' with the '-g c_Gammaproteobacteria -n -1' options. 'phylophlan_write_config_file' was run with the '-d a –db_aa diamond –map_dna diamond –map_aa diamond –msa mafft –trim trimal –tree1 iqtree' options, and the final phylophlan run was executed using 'phylophlan' with the options '-d phylophlan –diversity medium –accurate -t a'. The resulting IQTree file was used for phylogenetic analysis and as the basis for the tree in Supplementary Fig. 1, and Shimodaira–Hasegawa values were added by re-running the dataset in IQTree using the original seed (857,918) and resampling 10,000 times.

## Electron microscopy

Flagellation was measured using SEM. SEM was rendered using an extreme high resolution (XHS) TFS Magellan 400 (ScopeM, ETH Zurich) outfitted with a field emission gun and operated at 2.00 kV and 50 pA. Images of fixed bacteria were obtained using a secondary electron through-the-lens detector. Liquid culture samples were collected at

different time points and fixed with 1% (w/v) glutaraldehyde. Samples were then deposited on hydrophilized silicon wafers treated with 0.01% poly-L-lysine. The wafers were successively submerged in 2.5% glutaraldehyde (salinity 27 psu), seawater, 1% osmium tetroxide and ASW for 5 min each. This was followed by an ethanol drying series (30%, 50%, 70%, 90% and 100%, with samples submerged for 2 min for each step), followed by final washing three times in water-free ethanol. The samples were then critical-point dried using the cell-monolayer programme (CPD 931 Tousimis, ScopeM) and mounted with silver paint to aluminum stubs. The stubs were then sputter coated with 4 nm of Pt-Pd (CCU-010 Metal Sputter Coater Safematic, ScopeM) to prevent sample charging. Cell and flagellar lengths were determined using ImageJ.

### Cell counting and viability measurements

Cell counting was performed by diluting cells by a factor of 100 and staining them with SYBR Green I (Thermo Fisher). For samples where the dead fraction was determined, a second sample was stained with with SYTOX Green (Thermo Fisher). Cells were stained at a final concentration of 5 μM for both stains and incubated in the dark for 10 min at room temperature. After staining, cells were counted using a flow cytometer (Beckman Coulter, CytoFLEX S) equipped with a 488 nm laser. Cell counts were determined after gating on the basis of the fluorescence and forward scatter signals (Supplementary Fig. 2). Plate counts to determine the colony-forming units were performed on MB (1.5% agar) plates with 15 ml liquid per plate. Only plates with 20–350 colonies were included in the analysis.

### Staining of storage granules

DAPI staining to probe polyphosphate levels was based on methods described previously[83,84]. A 5 mg ml$^{-1}$ DAPI (4′,6-diamidino-2-phenylindole dihydrochloride, Thermo Scientific) solution in filtered (milliQ) water, stored at −20 °C, thawed and diluted to 25 μg ml$^{-1}$ in ASW as a working stock on the day of the experiment. Samples of 1 ml cell suspension with ~1 × 10$^7$ cells (OD 0.01) were fixed with 3.7% paraformaldehyde for 1 h and then added to a filter tower pulled through a filter column by pressure difference. The filters (25 mm diameter) consisted of a nitrocellulose backing filter (0.4 μm, Thermo Scientific) covered by a black Isopore membrane filter (0.2 μm). After filtration, the black membrane filter was placed on 500 μl DAPI solution for 10 min in the dark. The filter was then gently washed by sweeping it through a drop of milliQ water and dried for 10 min. The filter was then placed on a standard glass slide under a 24 × 50 mm coverslip, with a small drop (20 μl) of a photostability mixture consisting of 4 parts Citifluor (Citifluor) and 1 part Vectashield (Vector Laboratories). Samples were measured using an oil immersion objective (×100, 1.4 NA, Nikon) and a Canon EOS 80D DSLR camera (ISO 800, 0.25 s exposure), with excitation by a broad-spectrum mercury lamp (Prior Scientific) with DAPI filter cube (Chroma, ex: 350/50 nm, di: 400 nm, em: LP420 nm).

The neutral lipid stain Bodipy 493/503 (Thermo Fisher) was used to visualize PHB granules[85]. A stock solution of 1 mg ml$^{-1}$ in DMSO was diluted 10-fold in DMSO to obtain a working stock of 100 μg ml$^{-1}$. To stain cells, 5 μl dye solution was added to 0.5 ml cell suspension with ~1 × 10$^8$ cells per ml (OD 0.1), briefly vortexed and incubated on ice for 10 min in the dark. Cells were then immobilized by placing them on poly-L-lysine (Sigma) coated coverslips for 30 min. Cells were imaged using an oil immersion objective (×100, 1.4 NA, Nikon) and CMOS camera (ORCA Flash 4.0, Hamamatsu) under epifluorescent illumination provided by a mercury lamp (Prior Scientific) with Chroma EGFP filter cube (ex: 470/40 nm, di: 495 nm, em: 525/50 nm). To determine the PHB content per cell, the raw fluorescence signal $F^*$ of a rectangle with one cell was integrated, and the background fluorescence value of an area without cells in the same image subtracted. For each cell, $F^*$ was then normalized to $F$ according to $F = F^*/\langle F^*_{auto}\rangle - 1$, with the autofluorescence $\langle F^*_{auto}\rangle$ value averaged over all cells of a strain without PHB synthesis genes.

### Quantitative phase imaging

Quantitative phase images were obtained using a microscope equipped with digital holography microscopy (Lynceetec, Switzerland)[46]. For each measurement, 100 μl of cell suspension was placed on dry ice for 2 min to cease motility. Then the sample was thawed and 20 μl was placed in an observation well consisting of two coverslips separated by a single parafilm layer, for at least 15 min to ensure cells were settled. Quantitative phase images were obtained at 100 positions per sample, with each position consisting of an averaged stack of 25 autofocused images, each displaced ~2 μm to average out aberrations due to the optical path. Simultaneously, bright-field (BF) images were recorded for each of the 100 positions. Image segmentation was performed using Ilastik[86]. Objects were first detected using BF and the cell contour was then determined by a watershed detection on the QPI. Only objects identified both in QPI and BF were analysed. The dry mass for each object was computed from the integrated intensity of the QPI and using a refractive index increment of $\alpha = 0.175$ ml g$^{-1}$, as described elsewhere[47]. Objects with a pixel area >500 pixels and a mass density <0.1 (w/v) % and >0.8 were removed from the analysis, as well as objects with a mass <20 fg and >400 fg (1,000 fg for growing cells).

### Training and usage of a naive Bayesian classifier

Protein-coding sequences from all strains were re-annotated using EGGNOG-mapper[87]. KEGG orthologous group (KOG) assignments[88] from EGGNOG annotations were tabulated for all strains. KOGs were filtered to remove KOGs with representation only from a single strain or KOGs with uniform representation. In the limokinetic classifier, genes with higher relative abundance in the limostatic strains were excluded as potential features (and vice-versa). The feature matrix was then binarized, reducing counts of each KOG to presence/absence data for each strain. Recursive feature elimination was implemented using the 'FeatureTerminatoR' package and attempting training from 2 to 1,000 features using 'leave-one-out' cross-validation for each strain, assuming a Poisson distribution and using a Laplacian smoothing value of 1. A value of 22 features was chosen as the smallest, high-accuracy feature set that would not be prone to overfitting (Extended Data Fig. 9). Training of the final Bayesian classifier was performed by the 'naivebayes' and 'caret'[89] packages in R. Training was performed using 128 train/test splits, training on 2/3 of the data and reserving 1/3 of the dataset for prediction.

For the depth profiling using the classifiers, the limokinetic and limostatic classifiers were applied to field data in the Ocean Microbiome Database (OMD1; https://microbiomics.io/ocean,[51]) to profile patterns of occurrence for each phenotype as a function of depth. Feature presence or absence was extracted from pre-calculated KOG for each metagenome-assembled genome (MAG) in the dataset. Abundance was calculated as the coverage of unambiguously (limokinetic positive AND limostatic negative or vice-versa) gammaproteobacterial MAGs divided by the total coverage or total coverage of gammaproteobacterial MAGs. For the normalized depth profiles, the fractional abundance in each sample was log-normalized, mean-centred and scaled independently for each group.

Logistic regression analysis to test for phylogenetic bias of feature selection was performed using the R package phylolm (v.2.6.2)[90]. Of the 22 and 121 OGs in the limokinetic and limostatic classifiers, respectively, 6 and 56 OGs have relationships with classification outcome that cannot be explained by phylogeny alone with high certainty ($P < 0.05$), and 9 and 5 with moderate certainty ($P < 0.10$) (Fig. 4, Supplementary Table 3 and Supplementary Note 4). The $P$ values for all features were computed using a two-sided $z$-test (logistic regression) or two-sided linear regression for each feature individually (no multiple comparisons adjustment). We found that the limokinetic and limostatic classifiers performed better than a taxonomic classifier ($P = 0.07$) and were able to accurately predict variation within the Vibrionaceae (Supplementary Note 4).

## Reporting summary

Further information on research design is available in the Nature Portfolio Reporting Summary linked to this article.

## Data availability

All data used to support statements in this manuscript, including all bacterial cell trajectories, are available through figshare at https://doi.org/10.6084/m9.figshare.26195339 (ref. 91). This repository includes figure source data. Raw microscopy data (>6 TB) can be obtained upon request. Genome accession numbers of the bacterial strains are listed in Supplementary Table 1. Ocean Microbiome Database v.1.1., used in this study for model prediction, is available at https://microbiomics.io/ocean/.

## Code availability

Code for cell tracking, analysis, classifier training and testing is available on figshare at https://doi.org/10.6084/m9.figshare.26195339 (ref. 91).

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

## Acknowledgements

We thank N. Norris, G. Meijer, E. Ledieu, Y. Yawata, N. Blitvic, V. Fernandez and J. Yan for advice and discussions; R. Naisbit for scientific editing and J.-B. Raina for a critical reading of the manuscript; L. Paoli and A. Stubbusch for help with the Ocean Microbiomics Database; L. Moor and L. Schocher for help with experiments; K. Drescher (U. Basel), G. D'Souza (EAWAG), K. Jung and S. Brameyer (LMU Munich), M. Ullrich (U. Bremen), S. Charlton and I. Short (ETH Zurich) for sharing strains; and the microscopy facility ScopeM at ETHZ for SEM training, imaging and facilities. This work was supported by the Simons Collaboration on Principles of Microbial Ecosystems (PriME; grants 542395FY22 to R.S. and 542379FY22 to M.A.), the Swiss National Science Foundation (grant 205321_207488 to R.S.), NCCR Microbiomes, a National Center of Competence in Research funded by the Swiss National Science Foundation (grant numbers 51NF40_180575 and 51NF40_225148 to R.S.) and a Gordon and Betty Moore Foundation Symbiosis in Aquatic Systems Initiative Investigator Award (grant GBMF9197 to R.S.; https://doi.org/10.37807/GBMF9197).

## Author contributions

J.M.K., F.C. and R.S. conceived the work. J.M.K., F.C., R.S., M.A. and Z.C.L. designed experiments. J.M.K., S.T.Z., D.A.B. and E.E.C. performed experiments, with help from C.M.-P.; J.M.K., D.A.B. and S.T.Z. analysed experimental data. Z.C.L., B.R.K.R. and C.M.-P. performed genetic analyses. Z.C.L. performed model training and testing. All authors discussed and interpreted results. J.M.K., Z.C.L., B.R.K.R. and R.S. wrote the paper with input from all authors.

## Funding

## Competing interests

The authors declare no competing interests.

## Additional information

**Extended data** is available for this paper at https://doi.org/10.1038/s41564-025-01997-7.

**Correspondence and requests for materials** should be addressed to Johannes M. Keegstra or Roman Stocker.

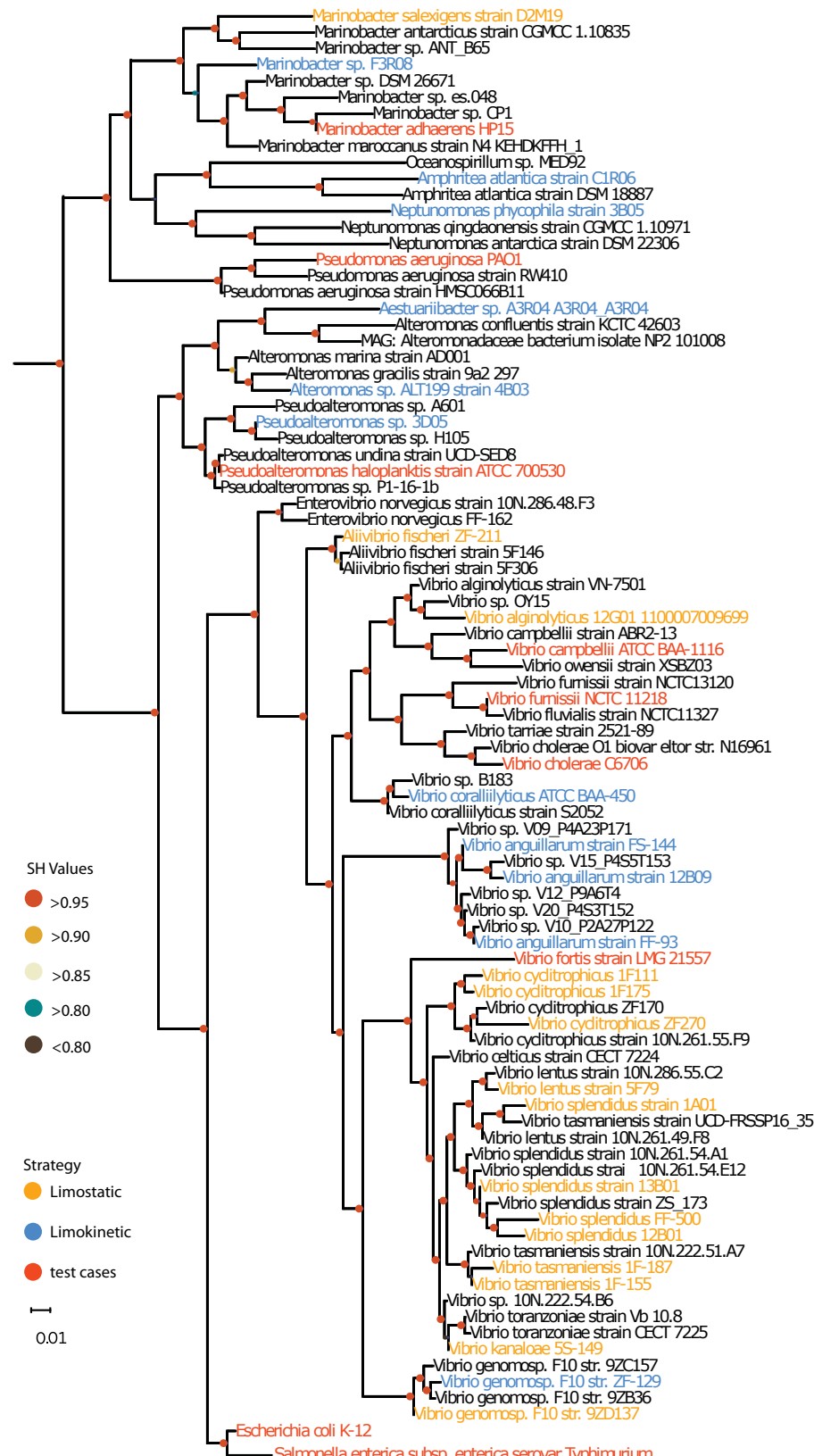

**Extended Data Fig. 1 | Phylogenomic tree of all strains used in this study.** Isolates spanning 3 orders of gammaproteobacteria were used in this study, with high representation among the Vibrionaceae. The names of studied strains are colored according to their behavioral response to starvation: limokinetic strains in blue; limostatic strains in orange. The names of additional strains used to test the predictive ability of the classifier are shown in red ('test cases', see Fig. 4 and Main Text). Node values represent Shimodaira-Hasegawa (SH)-like test values with 10,000 resamplings (obtained using the -alrt option in IQTree2[92]). Phylogenomic tree was constructed using PhyloPhlAn 3.0[82] The tree is rooted on Pelagibacter ubique 1062 (SAR11) (NCBI accession: CP000084.1; not shown). Tree pruning, ordering and aesthetics were carried out using ETE3. Scale bar represents 0.01 nucleotide substitutions per site. For *Vibrio fortis* KT626460, strain LMG21557 has been used for prediction.

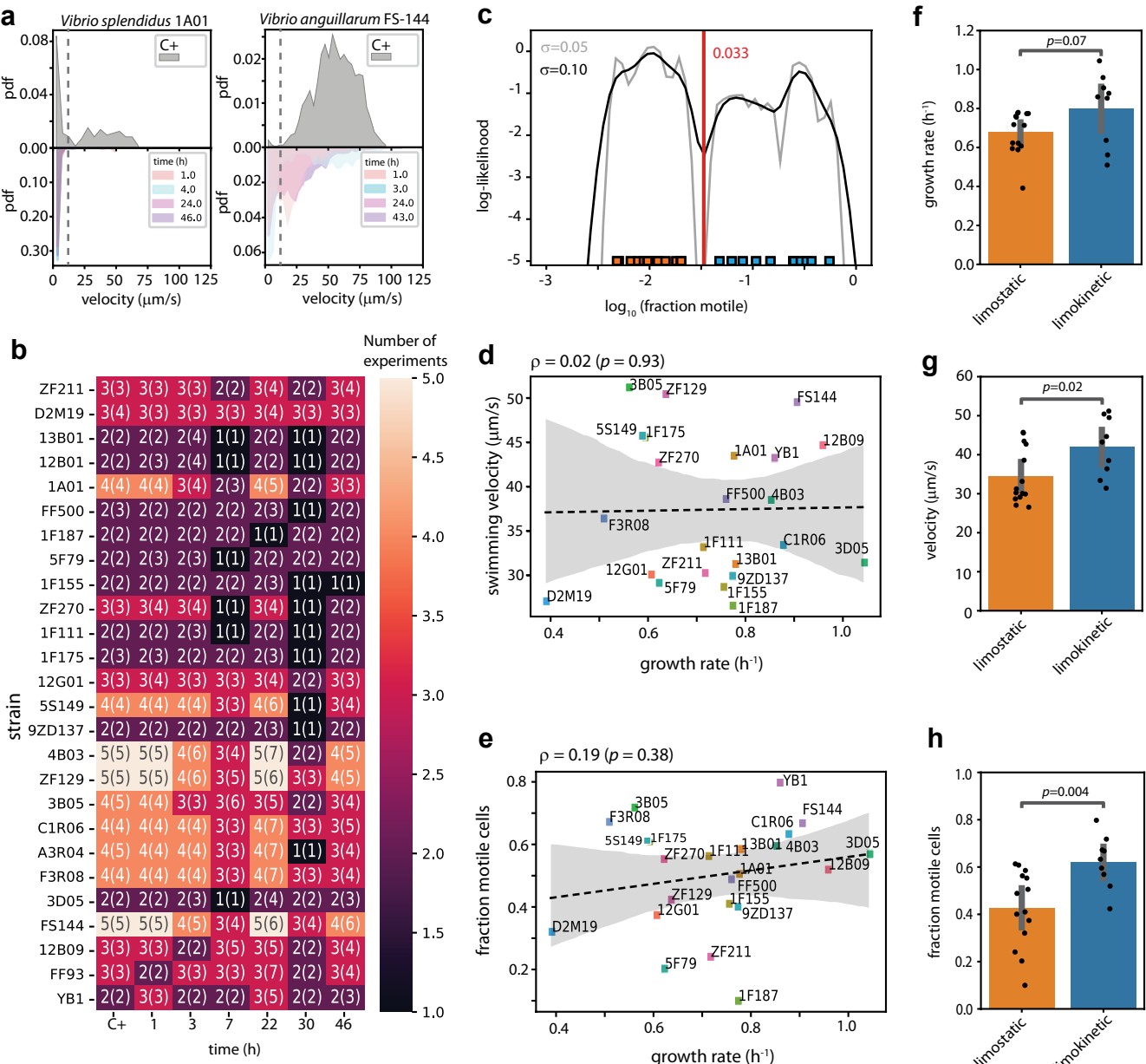

**Extended Data Fig. 2 | Response of marine bacteria to starvation reveals a dichotomy in motility endurance. a**: Distribution of average swimming velocities in *V*. splendidus 1A01 (left) and *V*. anguillarum FS-144 (right) prior to starvation ('C+'; top) and at different times during carbon starvation (1 h to 46 h; bottom). Dashed gray lines mark the velocity of 12 µm/s, used to differentiate motile from non-motile cells (Supplementary Fig. 1). 'pdf': probability density function. Data originate from identical experiments as used to produce Fig. 1b, for a different pair of strains from the same species. **b**: The number of observations for each of the 26 strains during the starvation experiment (Fig. 1). For each time bin, the number of independent experiments (performed on different days) is shown (number and heatmap), with the total number of videos (including replicates taken on the same day) indicated in brackets. **c**: Separation of limokinetic and limostatic strains based on a kernel-density estimate (KDE) on the logarithm of the time-averaged fraction of motile cells per strain for starvation times > 1 h (for example the average of each row in Fig. 1d). Results for two different bandwidths σ are shown, σ = 0.05 (gray) and σ = 0.10 (black). A single local minimum (at 0.033) of the KDE indicates the fraction of motile cells that

best separates two behavioral classes. **d**: Relationship between specific growth rate (ln(2)/doubling time) and average speed of motile cells during growth in 50% Marine Broth for 23 strains. Pearson's correlation coefficient, $\rho$ = 0.02 (dashed line), is shown as band center with 95% confidence interval (shaded area). **e**: Relationship between specific growth rate and fraction of motile cells for 23 strains. Pearson's correlation coefficient, $\rho$ = 0.29 (dashed line), is shown as band center with 95% confidence interval (shaded area). In panel d and e, the specific growth rate was computed from 4 technical replicates of a single biological replicate. The number of replicates for the motility parameters is shown in panel **b**. **f**: Average growth rates (from panels **d**,**e**) during growth for limokinetic and limostatic strains are not significantly different (two-sided Mann-Whitney U test (M.W.U.): $p$ = 0.07). In panels **f**-**h**, the error bars represent 95% CI. **g**: Average swimming velocities (from panel **d**) of motile cells during growth for limokinetic and limostatic strains. The average velocities of the two behavioral classes differ significantly (M.W.U.: $p$ = 0.02). **h**: Fraction of motile cells (from panel **e**) during growth for limokinetic and limostatic strains. The average motile fractions of the two behavioral classes differ significantly (two-sided M.W.U.: $p$ = 0.004).

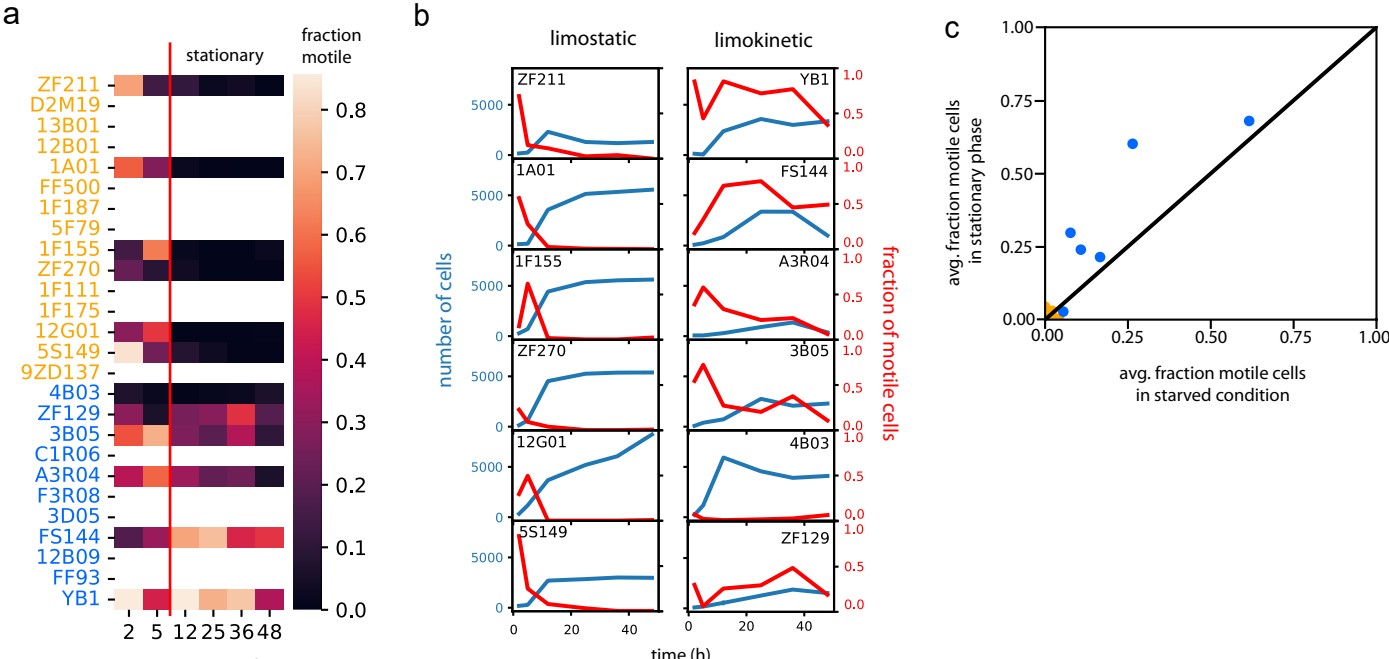

**Extended Data Fig. 3 | The motility response to nutrient limitation is similar during stationary phase and after washing.** To quantify motility during stationary phase, we diluted cells (1/2000) from a culture grown overnight in 100% MB into 2% MB in artificial seawater. The concentration of MB was chosen to obtain a final concentration of cells that is low enough to remain compatible with tracking (OD ~ 0.02). **a**: Heat map showing the motile fraction for different strains at various time points in 2% in Marine Broth. The red vertical line indicates the approximate transition between growth and stationary phase, based on the plateau in the cell number in panel b. Strain names are shown in blue for limokinetic strains and orange for limostatic strains, based on their motility response after washing and transfer to carbon-depleted medium. **b**: Number of cells (measured as the average number of trajectories per frame; blue) and fraction of motile cells (red) as a function of time in 2% MB. **c**: The time-averaged fraction of motile cells per strain in starvation medium and in stationary phase is highly correlated (Pearson's $\rho = 0.91$, CI: 0.67-0.97). Blue indicates limokinetic strains and orange limostatic.

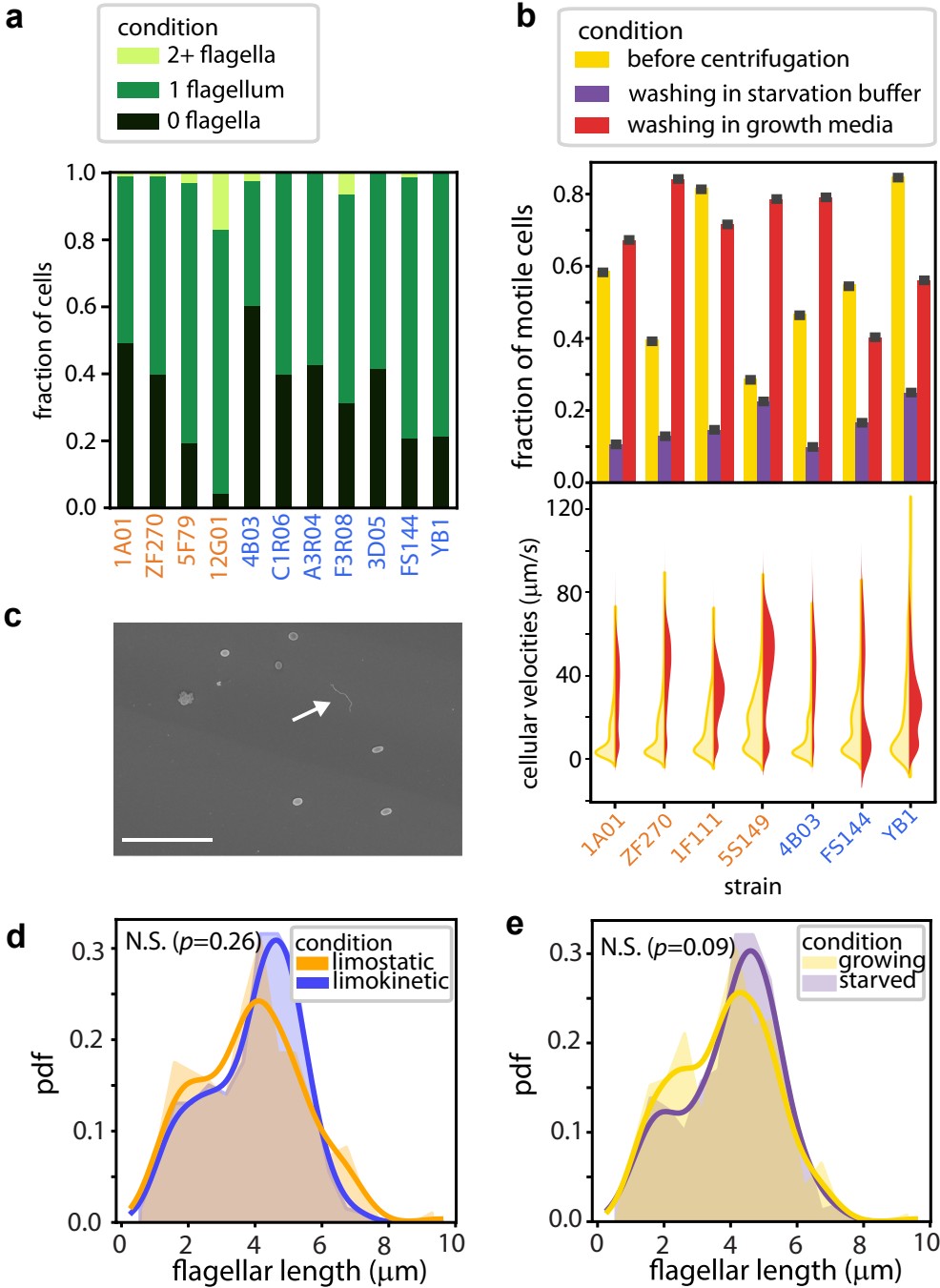

**Extended Data Fig. 4 | Loss of flagellar filaments is a result of exposure to nutrient starvation, not mechanical stress. a**: Fraction of cell population with 0 flagella (black), 1 flagellum (dark green) or multiple flagella (light green) per cell for 11 different strains measured with scanning electron microscopy (SEM). All cells were grown in Marine Broth. **b**: The fraction of motile cells (top) and average velocity per cell (bottom) for cells growing in 50% Marine Broth ('before centrifugation', yellow), after washing in starvation buffer (purple), and after washing in fresh 50% Marine Broth (red), for 7 strains (4 limostatic strains labeled in orange and 3 limokinetic in blue). **c**: Example image of 24 h-starved *Vibrio cyclitrophicus* ZF270 cells with an isolated flagellar filament. Scale bar 10 μm.

**d**: Distribution (shaded area) and KDE-estimate (lines) of filament length for flagella of all measured limokinetic (blue, *n* = 189) and limostatic (orange, *n* = 388) cells. All Filaments measured by SEM in an experiment where the cells were grown in carbon-replete media and then starved for up to 24 h (Fig. 2). Distributions are not significantly different (two-sided M.W.U: *p* = 0.26). **e**: Distribution (shaded area) and KDE-estimate (lines) of flagellar length for all cells during growth in carbon-replete media (yellow, *n* = 268) and during carbon starvation (purple, *n* = 309). Distributions are not significantly different (two-sided M.W.U: *p* = 0.09). Data originates from the same experiment as in panel d.

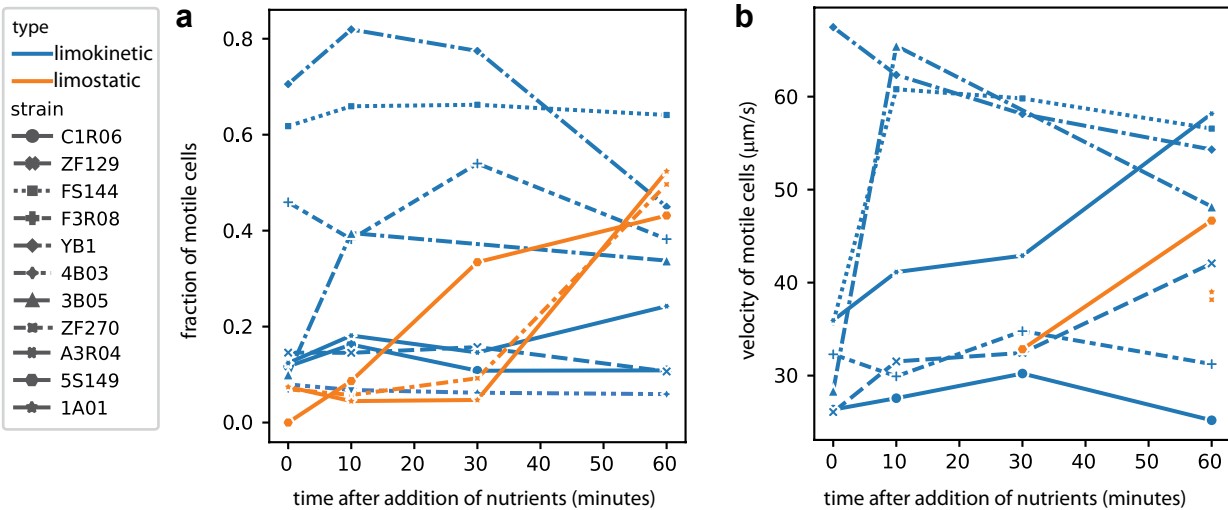

**Extended Data Fig. 5 | Delayed motility recovery in limostatic compared to limokinetic strains. a,b**: Fraction of motile cells (**a**) and average velocities of motile cells (**b**) for limostatic (orange) and limokinetic (blue) strains after the addition of 1% Marine Broth to cultures starved for 24 h. Velocities are shown for strains in which the fraction of motile cells was greater than 10%.

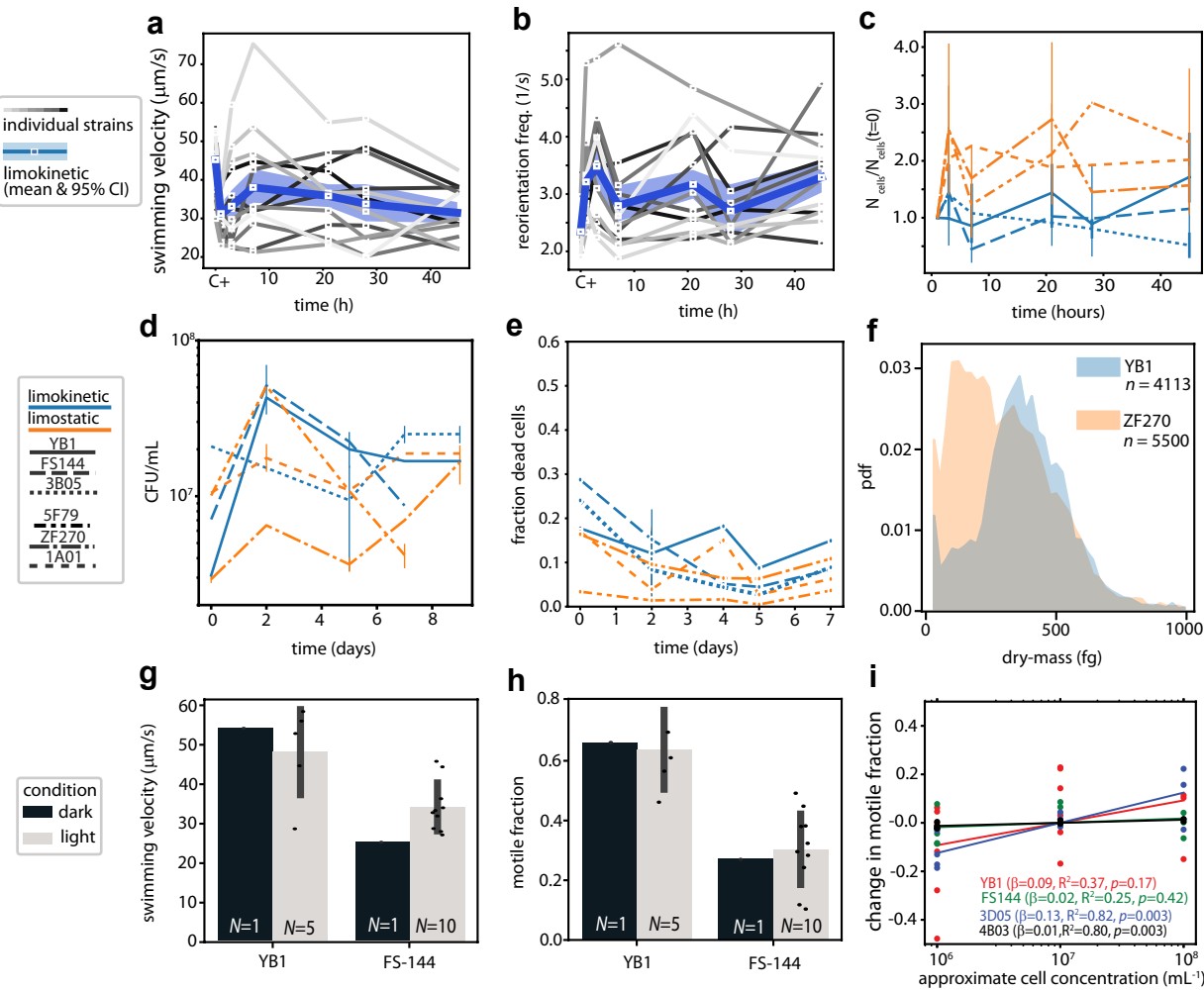

**Extended Data Fig. 6 | The energetic cost of motility during starvation.**
**a**: Population-averaged swimming velocity of motile cells as a function of time
for single strains (gray lines) and the average over all strains (blue line) with
95% confidence interval (shaded area). In panel **a** and **b**, 'C+' denotes the
condition prior to starvation. **b**: Population-averaged reorientation frequency
as a function of time. Colors as in panel **a**. **c**: Number of cells as a function of
time for 3 limokinetic and 3 limostatic strains (see legend next to panel **d**), as
estimated from the number of trajectories per frame, normalized to the estimate
at the first time point after washing. Error bars represent mean with 95% CI.
**d**: Viable cell concentration as a function of time for 3 limokinetic and 3 limostatic
strains, measured by colony counting on Marine Broth (1.5% agar) plates. Only
plates with 20-350 colonies were included in the analysis. Error bars denote the
standard deviation of 2-3 plates per condition (technical replicates). **e**: Fraction
of cells with a compromised membrane as a function of time. The number of
dead cells was estimated using SYTOX Green staining and the total number
of cells by SYBR Green staining (Fig. 3C). Each line is computed as the average
of 2 biological replicates (1 replicate for day 0 and 3 for day 2, with error bars
representing 1 s.d.). **f**: Single cell mass distribution measured with quantitative
phase imaging for cells growing in Marine Broth, for *Vibrio coralliilyticus* YB1

and *Vibrio cyclitrophicus* ZF70, before starvation. **g,h**: Limokinetic cells are also
motile in the dark. Average swimming velocity (**g**) and motile fraction (**h**) of cells
during starvation t > 24 h in the dark (dark gray, single replicate) and with normal
light exposure (pale gray, see Fig. 1). To test the effect of light on swimming,
*Vibrio anguillarum* FS-144 and *Vibrio coralliilyticus* YB1 cultures were grown and
starved following the standard protocol (Methods) but then kept in culture tubes
wrapped in aluminum foil. Microscopy samples were prepared in the dark and
cell motility was quantified immediately upon placing them on the microscope.
Without covering the tubes, the cells experienced a diel cycle (approx. 16 h of
light per day) with the starvation process starting in the afternoon. Error bars
indicate mean +/- one s.d. **i**: The motile fraction of cells, for starvation times
> 24 h, as a function of cell concentration for 4 limokinetic strains (colored
points). For each strain, the cell concentrations were obtained by diluting the
same culture, where 1 indicates the standard dilution in our starvation protocol,
corresponding to ~ $10^7$ mL⁻¹. Solid lines represent linear fits to the data, with the
slope $\beta$, residuals $R^2$ and associated probability $p$ indicated in the figure legend.
The motile fraction is not negatively correlated with the cell concentration,
which excludes residual nutrients in the medium having a large influence on
motility during starvation.

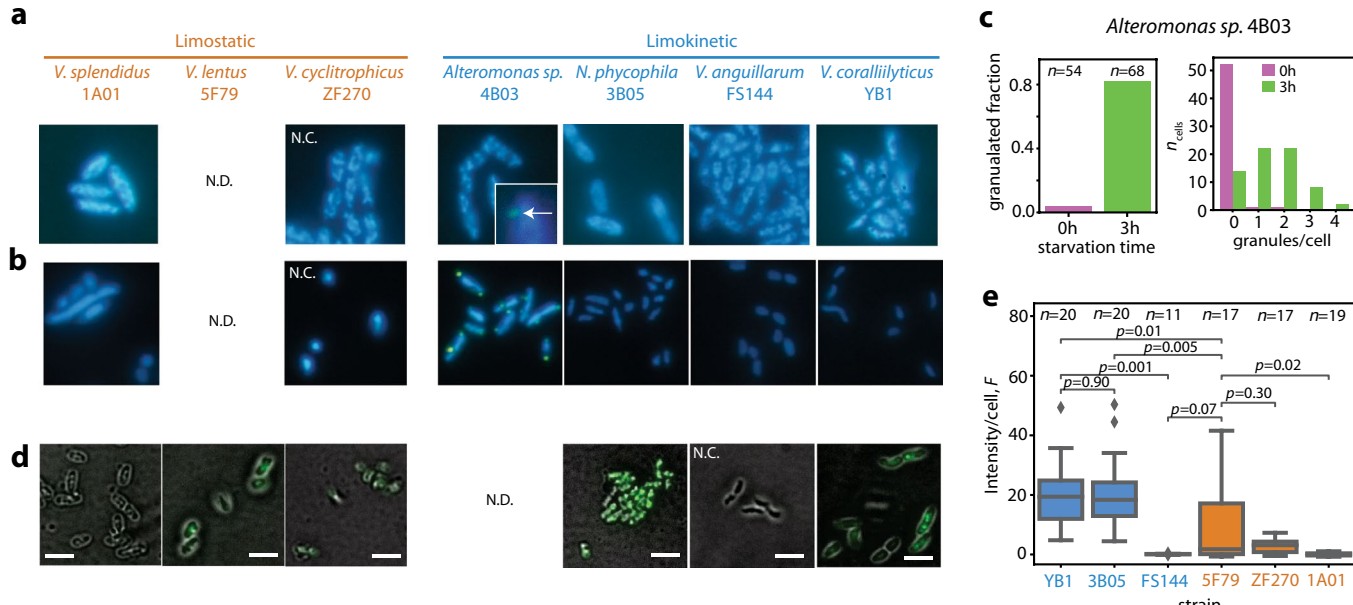

**Extended Data Fig. 7 | Accumulation of energy storage compounds in carbon-replete medium.** Absence of polyphosphate (polyP) storage during exponential growth in rich medium in 2 limostatic and 4 limokinetic strains, measured by DAPI staining. PolyP granules are visible in green, DNA in blue. None of the inspected cells showed polyP granules, except for 2 out of 54 cells of strain 4B03 (one cell in inset, granule indicated by white arrow). N.C.: negative control (strain lacks storage compound synthesis genes, see Supplementary Table 2). N.D.: Not determined. **b**: As in panel **a**, but then for strains starved for carbon for 3h. Carbon starvation induces polyP granule formation in strain 4B03. In the carbon-limited medium, phosphate is in excess (Methods). **c**: Fraction of cells observed in 4B03 containing at least 1 polyphosphate granule per cell (left) and histogram with number of polyP granules per cell (right) for cells stained during growth (0 h) and cells in carbon-limited medium (3 h). Each time point corresponds to a single biological replicate. **d**: Polyhydrobutyrate (PHB) storage in 3 limokinetic and 3 limostatic strains during exponential growth in rich medium, measured by Bodipy staining. Scale bars: 3 μm. **e**: Fluorescence intensity per cell from staining of PHB with Bodipy, for each strain. Box plots represent the values of the first, second (median) and third quartiles. Whiskers indicate minimum and maximum values of the distribution, limited by 1.5 times the difference between the first and the third quartile. Diamonds indicate outliers. Sample sizes indicate the number (n) of individual cells measured. Significance based on two-sided Tukey HSD-test.

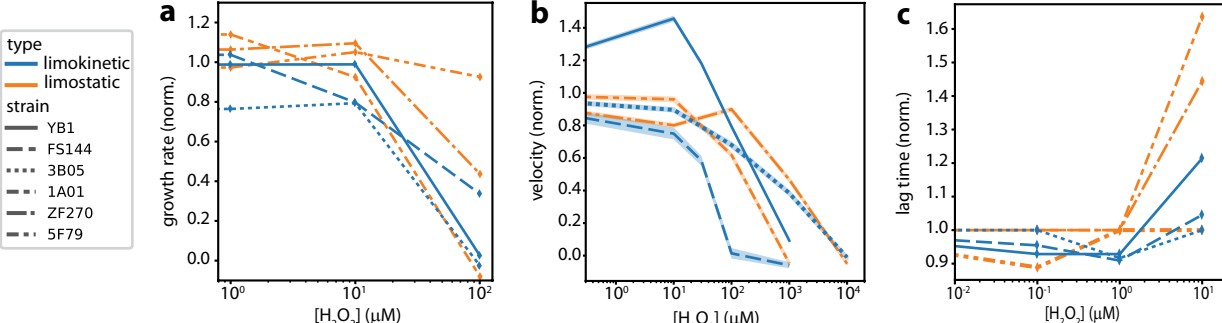

**Extended Data Fig. 8 | Experimental tests of oxidative stress defense, a prominent classifier feature. a**: Growth rate as a function of the added external hydrogen peroxide concentration [$H_2O_2$] for 3 limokinetic (blue) and 3 limostatic strains (orange). Growth rates were normalized to the growth rate of each strain measured without the addition of hydrogen peroxide. **b**: Average swimming velocity as a function of [$H_2O_2$] for 3 limokinetic (blue) and 2 limostatic strains (orange). Velocities were normalized to the velocity of each strain without the addition of hydrogen peroxide. **c**: Lag time as a function of the added [$H_2O_2$] for

3 limokinetic (blue) and 3 limostatic strains (orange). Cultures were starved for 24 h and lag time was measured after adding Marine Broth (final concentration 50%). Lag time was defined as the time until the culture reached an OD of 0.05. Values were normalized by the lag time of each strain without the addition of hydrogen peroxide. The average lag time at [$H_2O_2$] = 10 µm was not significantly different (two-sided $t$-test, $p = 0.24$) between limostatic (1.36) and limokinetic (1.09) strains.

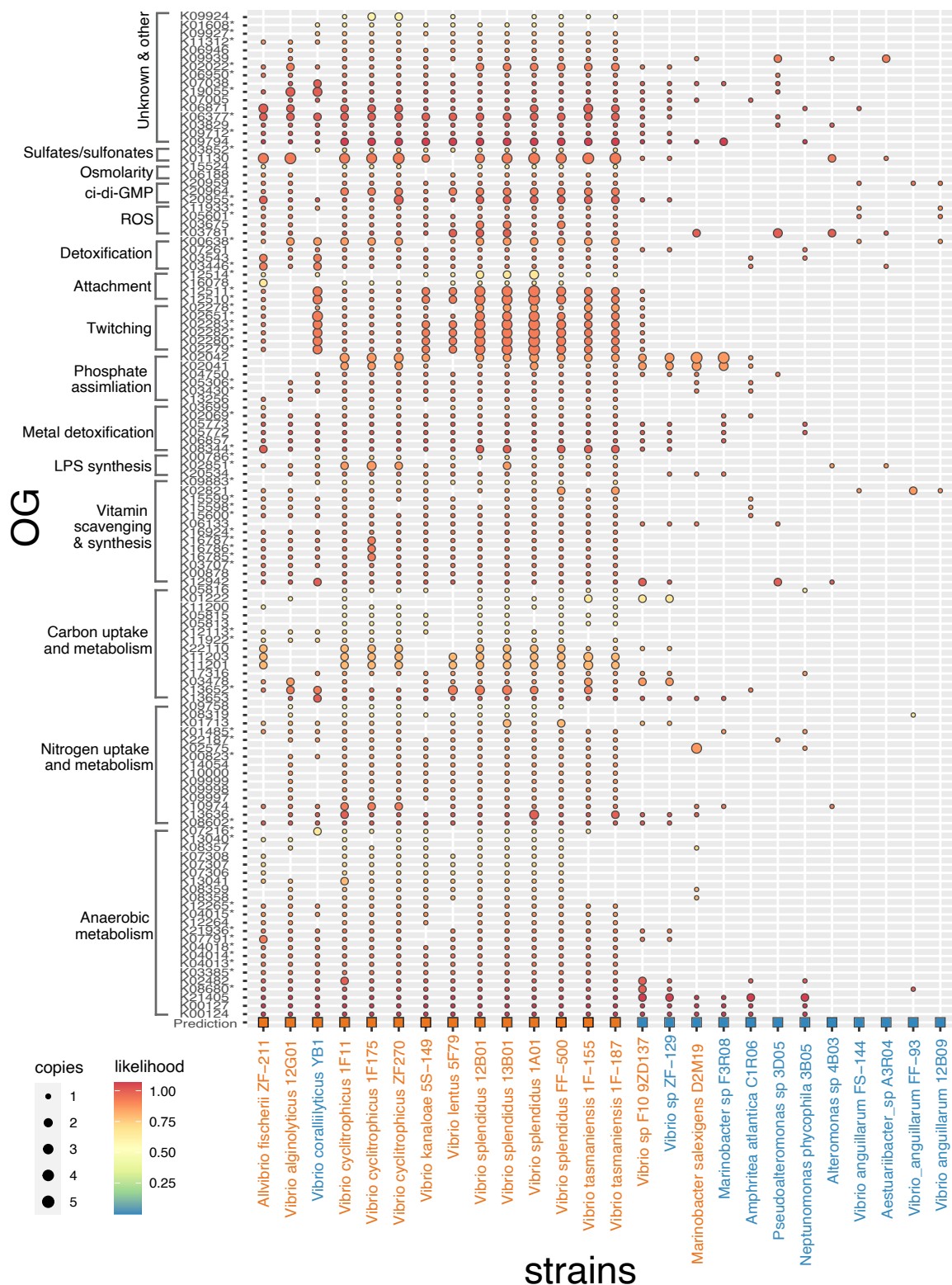

**Extended Data Fig. 9 | Feature selection of the limostatic classifier.** Prevalence of orthologous groups (OG) associated with a limostatic response for both limokinetic and limostatic strains, as obtained by RFE (Methods) and clustered into 15 functional categories. OG with significance $p < 0.05$ from a regression analysis that includes phylogeny (Methods) are marked with *. Complete list of $p$-values is available in Supplementary Table S4. Circles indicate the gene copy number of each OG (size) and the probability of the association with the limostatic response (color). 'Prediction' (bottom row) is the predicted class of each strain by the classifier.

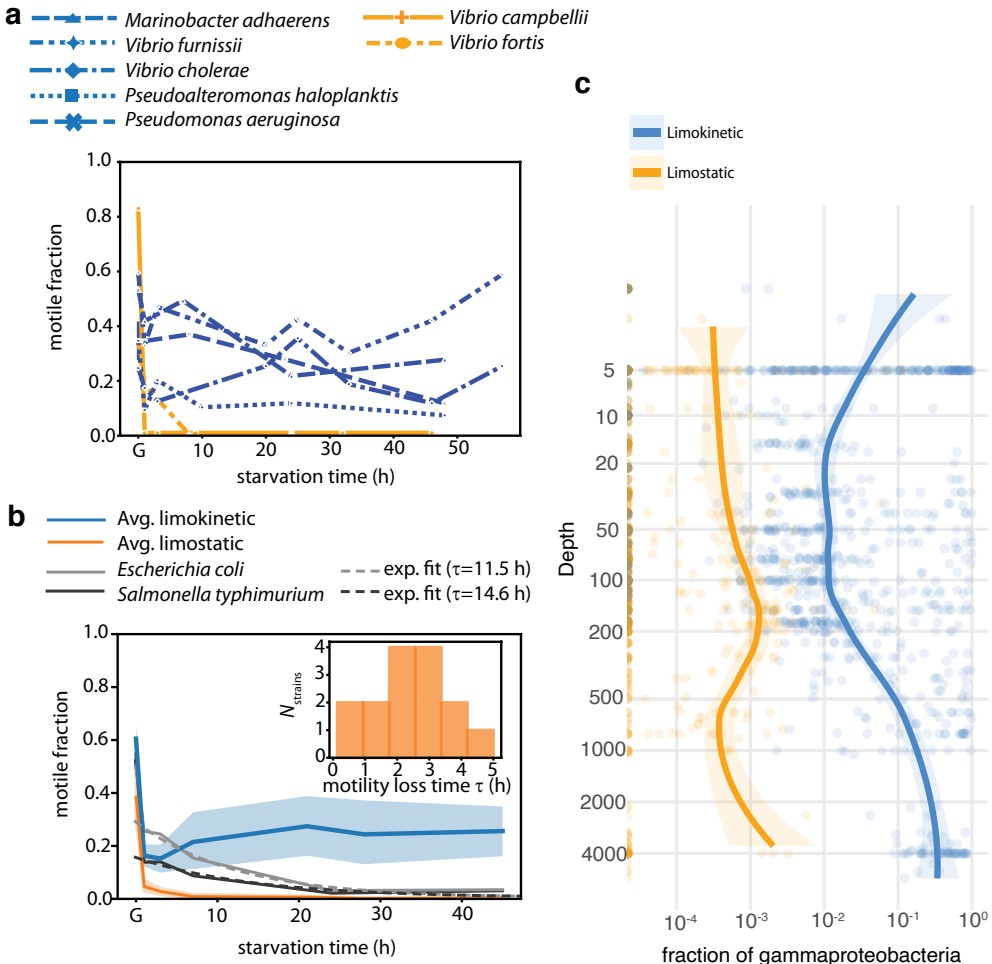

**Extended Data Fig. 10 | Classifer predictions on laboratory experiments and field samples. a**: Testing of classifier prediction in other marine strains not used for initial training. Lines show the observed motile fraction as a function of starvation time under the same conditions as in Fig. 1, providing experimentally-determined classification (colors). 'G' denotes the condition before starvation. **b**: Testing of classifier prediction in enteric bacteria. Shown are the measured motile fraction of *E. coli* RP437 gray and *Salmonella typhimurium* LT2 (black) and fits to a single exponential decay (dashed lines). For these strains, the standard starvation protocol was adapted by replacing Marine Broth with Tryptone broth (10 g/L Bacto tryptone and 5 g/L NaCl), and as starvation buffer adapted motility buffer[93] was used without potential energy sources lactic acid and methionine (10 mM potassium phosphate at pH 7.0, 0.1 mM EDTA). Each experiment was replicated twice. The behaviour of the enteric bacteria deviates from the strong dichotomy found in marine strains: their motile fraction steadily decreases with starvation time, representing limostatic behaviour, but with a timescale much longer than observed in marine strains (orange). Their motile fraction is much lower compared to the average of all limokinetic marine strains (blue). Shaded areas represent the mean with 95% CI. Dashed lines indicate an exponential fit of the motile fraction as a function of starvation time (excluding the datapoint for growth). Inset: motility loss timescale for limostatic strains, obtained from single exponential fits to the motile fraction during starvation. **c**: Predicted fraction of limokinetic (blue) and limostatic (orange) taxa for different ocean sampling time points (circles), normalized by all gamma proteobacterial taxa. Prediction is based on the limostatic and limokinetic classifier (Fig. 4 and Extended Data Figure 9), including only the taxa where both classifiers have identical prediction (for example excluding ambiguous predictions). The line represents a LOESS smoothed average with 95% CI (shaded area).

# Reporting Summary

## Statistics

For all statistical analyses, confirm that the following items are present in the figure legend, table legend, main text, or Methods section.

| n/a | Confirmed | |
|---|---|---|
| ☐ | ☒ | The exact sample size (*n*) for each experimental group/condition, given as a discrete number and unit of measurement |
| ☐ | ☒ | A statement on whether measurements were taken from distinct samples or whether the same sample was measured repeatedly |
| ☐ | ☒ | The statistical test(s) used AND whether they are one- or two-sided<br>*Only common tests should be described solely by name; describe more complex techniques in the Methods section.* |
| ☐ | ☒ | A description of all covariates tested |
| ☐ | ☒ | A description of any assumptions or corrections, such as tests of normality and adjustment for multiple comparisons |
| ☐ | ☒ | A full description of the statistical parameters including central tendency (e.g. means) or other basic estimates (e.g. regression coefficient) AND variation (e.g. standard deviation) or associated estimates of uncertainty (e.g. confidence intervals) |
| ☐ | ☒ | For null hypothesis testing, the test statistic (e.g. $F$, $t$, $r$) with confidence intervals, effect sizes, degrees of freedom and $P$ value noted<br>*Give P values as exact values whenever suitable.* |
| ☐ | ☒ | For Bayesian analysis, information on the choice of priors and Markov chain Monte Carlo settings |
| ☒ | ☐ | For hierarchical and complex designs, identification of the appropriate level for tests and full reporting of outcomes |
| ☐ | ☒ | Estimates of effect sizes (e.g. Cohen's *d*, Pearson's *r*), indicating how they were calculated |

*Our web collection on statistics for biologists contains articles on many of the points above.*

## Software and code

Policy information about availability of computer code

| | |
|---|---|
| Data collection | Nikon Elements v5.02, CytExpert v2.4, Koala Acquisition & Analysis 8.5 |
| Data analysis | Data analysis was performed in Python 3.7 or higher. Tracking was performed using Trackpy (v0.4.2 and v0.5.0). Electron microscopy image analysis was performed using imageJ v2.1.0/1.53c. FlowCytometry analysis was performed using CytExpert v2.4. DHM image reconstruction was performed using Koala Acquisition & Analysis 8.5 (Lyncee Tech). Data visualization was performed using IQTree 2.0, Matplotlib 3.5.0 and Seaborn 0.11.2. Genomic model training and testing were performed in R 4.4.2, using packages FeatureTerminatoR, 1.0.0 naivebayes 1.0.0 and caret 7.0-1. Phylogenetic tree was constructed using Phylophlan 3.0 and post-hoc phylogenetic bias testing was performed using the R package phylolm v2.6.2. All custom code is available in the Figshare repository: https://doi.org/10.6084/m9.figshare.26195339 |

For manuscripts utilizing custom algorithms or software that are central to the research but not yet described in published literature, software must be made available to editors and reviewers. We strongly encourage code deposition in a community repository (e.g. GitHub). See the Nature Portfolio guidelines for submitting code & software for further information.

## Data

Policy information about availability of data

All manuscripts must include a data availability statement. This statement should provide the following information, where applicable:

- Accession codes, unique identifiers, or web links for publicly available datasets
- A description of any restrictions on data availability
- For clinical datasets or third party data, please ensure that the statement adheres to our policy

All data used to support statements in this manuscript, including all bacterial cell trajectories, are available through a Figshare repository: https://doi.org/10.6084/m9.figshare.26195339. This repository includes figure source data. Raw microscopy data (>6TB) can be obtained upon request. Genome accession numbers of the bacterial strains are listed in Table S1. Ocean Microbiome Database v1.1., used in this study for model prediction, is available through https://microbiomics.io/ocean/.

## Research involving human participants, their data, or biological material

Policy information about studies with human participants or human data. See also policy information about sex, gender (identity/presentation), and sexual orientation and race, ethnicity and racism.

| | |
|---|---|
| Reporting on sex and gender | n/a |
| Reporting on race, ethnicity, or other socially relevant groupings | n/a |
| Population characteristics | n/a |
| Recruitment | n/a |
| Ethics oversight | n/a |

Note that full information on the approval of the study protocol must also be provided in the manuscript.

# Field-specific reporting

Please select the one below that is the best fit for your research. If you are not sure, read the appropriate sections before making your selection.

☐ Life sciences        ☐ Behavioural & social sciences        ☒ Ecological, evolutionary & environmental sciences

For a reference copy of the document with all sections, see nature.com/documents/nr-reporting-summary-flat.pdf

# Ecological, evolutionary & environmental sciences study design

All studies must disclose on these points even when the disclosure is negative.

| | |
|---|---|
| Study description | This study investigated how a risk-reward trade-off in bacterial motility determines the motility endurance of marine bacteria. Different marine bacterial isolates were cultured and subsequently starved for carbon, and these bacterial strains were characterized using video microscopy and cell tracking, scanning electron microscopy, flow cytometry and chemical staining. The outcome of these experiments revealed a dichotomy in motile behavior during carbon starvation, and the experiments were used to train a genomic classifier that can predict the outcome in other strains not included in training. |
| Research sample | All samples are culturable marine bacterial isolates from different field deployments or mesocosm experiments. Experiments were performed on 26 strains from 18 species belonging to the Gammaproteobacteria class. Model testing was performed on 7 additional marine strains from 7 different species and 2 additional non-marine species. |
| Sampling strategy | Of the 107 available strains with motility and/or chemotaxis genes, we selected 36 strains to test for motility and growth, some of which were from the same species to encompass intra-species and inter-species variation, and all with both chemotaxis and motility genes. Of the 30 remaining strains (four strains did not grow in marine broth and two strains did not show motility during growth in marine broth), we randomly selected 26 strains to be used in this study. No statistical method was used to pre-determine sample size. |
| Data collection | Bacterial motility was measured using video microscopy and cell tracking (J.M.K. and S.T.Z.). Bacterial flagellation was measured using Scanning Electron Microscopy (S.T.Z.). Bacterial cell count and viability were measured using flow cytometry (S.T.Z.), optical density (J.M.K) and colony counting (J.M.K). Presence of storage compounds were tested using fluorescent stains and inspected using microscopy (J.M.K). Single-cell mass measurements were performed using digital holographic microscopy (D.A.B). |
| Timing and spatial scale | The principal timescale used to assess motility endurance was 2 days as theoretical estimates indicate that most motile bacteria will encounter at least one nutrient hotspot within this timescale (see Discussion). From each culture with one strain, bacteria were |

sampled from the medium immediately before starving the cells and at {1 (1), 2--4 (3), 5--9 (7), 19--24 (22), 28--32 (30), 43--48 (46)} hours after the washing protocol started, where the number in brackets refers to the weighted average of each time window, rounded to 1 h, that was used for averaging over multiple experiments. For experiments to determine the differential biomass loss and cell viability, one sampling per 24 +/- 1 h was performed for a period of 7 days.

| | |
|---|---|
| Data exclusions | No videomicroscopy experiments were excluded from the analysis. Bacterial trajectories were inspected manually and individual trajectories that were the result of tracking errors were removed. For colony counting, only plates with 20-350 colonies were used. |
| Reproducibility | All the laboratory experiments were repeated three times (unless noted otherwise) , where for each repeat the cells were cultured and starved independently. |
| Randomization | The selection of the bacterial strains was random, apart from the criterion that they could be grown and showed motility in rich media (2216 Marine Broth). When a subset of bacterial strains was chosen for further investigation, one strain per species were chosen. For the physiological characterizations of starvation, strains from the same genus were compared (when possible). |
| Blinding | Blinding was not pertinent to our study because it did not include any animals and/or human research participants. In addition, blinding was not possible since many analyses were also carried out by the person in charge of sampling. |

Did the study involve field work? ☐ Yes ☒ No

# Reporting for specific materials, systems and methods

We require information from authors about some types of materials, experimental systems and methods used in many studies. Here, indicate whether each material, system or method listed is relevant to your study. If you are not sure if a list item applies to your research, read the appropriate section before selecting a response.

## Materials & experimental systems

| n/a | Involved in the study |
|---|---|
| ☒ ☐ | Antibodies |
| ☒ ☐ | Eukaryotic cell lines |
| ☒ ☐ | Palaeontology and archaeology |
| ☒ ☐ | Animals and other organisms |
| ☒ ☐ | Clinical data |
| ☒ ☐ | Dual use research of concern |
| ☒ ☐ | Plants |

## Methods

| n/a | Involved in the study |
|---|---|
| ☒ ☐ | ChIP-seq |
| ☐ ☒ | Flow cytometry |
| ☒ ☐ | MRI-based neuroimaging |

## Plants

| | |
|---|---|
| Seed stocks | n/a |
| Novel plant genotypes | n/a |
| | |
| Authentication | n/a |

## Flow Cytometry

### Plots

Confirm that:

☒ The axis labels state the marker and fluorochrome used (e.g. CD4-FITC).

☒ The axis scales are clearly visible. Include numbers along axes only for bottom left plot of group (a 'group' is an analysis of identical markers).

☒ All plots are contour plots with outliers or pseudocolor plots.

☒ A numerical value for number of cells or percentage (with statistics) is provided.

## Methodology

**Sample preparation**
Cell counting was performed by diluting cells by a factor of 100 and staining them with Syber Green (Sigma Aldrich). For samples where the dead fraction was determined, a second sample was stained with with Sytox Green (Thermofischer). Cells were stained at a final concentration of 5 microM for both stains and incubated in the dark for 10 min at room temperature.

**Instrument**
CytoFLEX S (Beckman Coulter, USA)

**Software**
CytExpert Version 2.4

**Cell population abundance**
The bacteria were the only cell population present in the samples.

**Gating strategy**
The gating settings were set to the size of the measured objects (Forward scatter FSC) and Fluorescent intensity (FITC-A)

☒ Tick this box to confirm that a figure exemplifying the gating strategy is provided in the Supplementary Information.

