## [Peer Review File · Nature Microbiology]

Risk-reward trade-off during carbon starvation generates dichotomy in motility endurance among marine bacteria

Corresponding Authors: Roman Stocker and Johannes M. Keestra

Version 0:

Reviewer comments:

Reviewer #1

(Remarks to the Author)

The manuscript is like the kinetic and static bacteria the authors report. It is bimodal with strong points and a weak point. The primary strength is the report of bimodal limokinetic and limostatic bacteria. There are some excellent microbiology techniques and analyses. The weakness is the connection between the change of bacterial motility among strains and carbon storage, which is mentioned in the first and last sentences of the abstract and never mentioned again.

The weakness is easily resolved by removing those sentences and emphasising the significance of the dichotomy, which is a next step in our understanding of how bacteria make decisions and which pathways and strategies they have available to survive and thrive. This is a significant conceptual advance buttressed by multifaceted experimental tests of the concept. This should generate a whole sub-sub-discipline of papers examining this concept in multiple species, with different media and in the environment.

The results are particularly significant if the dichotomy holds for most motile taxa; chemotaxis does not blur or erase the effect and is maintained beyond lab cultures. These are the major conceptual points for and against publication. Below are technical strengths and weaknesses.

The dichotomy is strong and clear. Exploring alternative energy sources is excellent and demonstrates that the motile bacteria are not 'cheating'. However, their cutoff could force the dichotomy in that the bacteria are rated as motile or non-motile. They have many excellent tracking videos, and Figure 1 shows more of a bimodal distribution than solely the binary static versus kinetic outcome. The dichotomy is conceptually useful, but in future papers on this topic, it may weaken or blur the distribution of strategies. While they have described the dichotomy, it may be worth the authors pointing out that Figure 1 suggests intermediate strategies. The revelation of this work is that there are multiple strategies. The flagellar loss is a strategy, but do some bacteria use proteases to remineralise the flagella and take up the amino acids? This question is for a future paper but illustrates just one line of enquiry this paper reveals.

The authors have clearly considered alternative strategies, albeit from the dichotomy viewpoint. They convincingly rule out alternative energy sources, such as necromass recycling and light. This confirms that motility is fueled by internal biomass conversion. Showing that cell numbers remain the same but that biomass decreases is a worthwhile result rarely or never highlighted in the literature, adding significance to the manuscript. Their meticulous approach to this and the entire manuscript strengthens the study's conclusions.

The authors' ability to predict the strategy of other taxa is outstanding and speaks to the thoroughness of their work. Their investigation of their own biases further underscores the meticulousness of their approach.

Extrapolating carbon cycling and adaptations to oligotrophic environments is an overreach without experiments on bacteria directly from the ocean. Having the genetic database is excellent. However, their experiments minimised or eliminated nutrient gradients, and the ocean and other environments are full of them. This leads to the question of whether chemotaxis in the ocean alters behaviour, possibly breaking down the dichotomy.

(Remarks on code availability)

Reviewer #2

(Remarks to the Author)

In this manuscript, the authors ask if and how long (copiotrophic) marine bacteria retain motility after sudden starvation. They highlight that motility (and chemotaxis), although metabolically extremely costly, may allow bacteria to find nutrient hotspots in a dilute environment. Therefore, there is likely to be a high risk/reward trade-off between retaining motility or not during starvation. To investigate the strategy adopted by copiotrophic marine bacteria, the authors selected 26 isolates (18 species) and measured their motile fraction and swimming speed just before and after carbon starvation (up to almost 2 days). This approach is a major strength of the paper, as motility studies are often limited to either (i) different isolates of the same species (with more homogeneous behaviors), (ii) only a few species, or (iii) laboratory type strains.

Interestingly, rather than a continuous behavioral landscape, they observed two distinct poles: either cells 'persist' and retain flagellation for long periods of time, or they rapidly lose their flagella. They suggest names for these two categories and propose a genome-based classifier with reasonable performance against phenotypic determination to assign a marine species to one of the two categories. While I will have comments on such a classification, I note here that in microbiome fields it is much easier to simply highlight diversity than to propose overarching categories in a reasonably convincing way, as the authors do here. The authors then extend their work by noting that the distribution of these strategies (as detected by their classifier in the GOMicrobiome) is not homogeneous, suggesting that local environmental constraints shape the precise risk/reward trade-off.

Overall, the writing is clear, the methods and results are extremely well described, and the discussion is of high interest. I would like to thank the authors for making it such an easy and insightful read. The supplementary material is also of high quality and interest, with careful considerations on numerous topics, and further compensates for the need for simplifications in the main text (in part imposed by the heterogeneous dataset of isolates).

Major remarks

(1) Authors provide an excellent technical summary of the experiments done to support their claims, in Figure S2C.

I notice that two strains only have one experiment (replicate), and these two strains show a very low motility fraction even before carbon starvation. Given the important variability observed in motile fraction even in showcase examples (Fig.2B), it seems to me like insufficient data.

(2) Methods for tracking:

a. (l. 386) "Bacteria were observed within 15 minutes after sampling."

Cells can stay up to 15 minutes inside the closed motility chamber before an acquisition is done, somewhere in that chamber. Where? In the bulk or along glass surfaces? The movie suggests the first option, but the Methods should mention such information. More importantly, I am concerned about biases induced by the time for a non-negligible part of the motile population to accumulate at surfaces. Bacteria tend to remain swimming along surfaces once they reach them, exiting occasionally (especially due to turns): altogether, this usually biases bulk acquisition (made several minutes after injecting the cells) against fast low-turning-rate bacteria and will also bias (lower) the bulk motile/non-motile ratio. Did the author assess such biases? I believe it could explain part of the variability they observe. How would that impact their conclusion?

b. (ll.417-418) "Cell tracking was performed [...] after removing the background from each 418 image by subtracting the median image computed over the entire video."

If using a background correction on the whole movie, aren't the authors removing a part of the non-motile cells? The described method takes the median value of each pixel over ~14s (25-30 fps, 418 frames), a duration I do not expect to be sufficient for a non-motile cell to diffuse enough to not be erased by the background correction. How is that not biasing the motile fraction computation?

(3) (ll. 82-83) "All strains exhibited a run-reverse-flick motility pattern (Supplementary Videos S5-S10), a hallmark of single-flagellated bacteria [30, 31]."

While most of the video do show cases run-reverse-flick (RRF) noticeable by eye, it does not appear to me as sufficient proof to sustain an exclusive claim. Indeed, the authors will know that under a speed threshold (dependent on species) the buckling instability causing the flick every other turn will not happen, therefore leading to a run-reverse (RR) motility pattern. RR patterns can cohabit with RRF patterns. RR patterns notably have a different (lower) diffusion coefficient.

(4) Authors make use of an equation from Taktikos et al. (eqn 23 in the article) for computing the diffusion coefficient from rotational diffusion, average swimming speed and turning frequency.

This equation assumes RRF motility with identical forward and backward run duration and constant speed. I find the usage of the diffusion coefficient as a quantitative metric in this manuscript very dubious due to (1) the likelihood of RR patterns coexisting with RRF patterns, (2) the potential biases in turning frequency and swimming speed mentioned above, (3) the absence of justification regarding a symmetrical pattern (likely beyond capabilities of the tracking method here), (4) the usage of a single rotational diffusion value despite strong reasons to think it would vary from species to species (e.g. different body sizes as shown in Figure S10 A). I understand the need for estimates of such value in order to develop supplementary discussion, and I appreciate it as done in Supplementary Discussion 2.2, but it is clearly a stretch to provide the diffusion coefficient of all strains at different time points (Supplementary Discussion 2.1, SFigure 7C) and comment on differences between limostatic and limokinetic strains.

Minor remarks

(5) Could the author comment on the (less likely) possibility that the dichotomy limostatic/limokinetic they observe is just a difference in response to the sudden withdrawal of an unknown trace elements initially present in MB, rather than carbon source?

(6) Is there any hint within the data for a difference in dispersal strategy between different limokinetic strains?

Nonconsequential remarks

(7) Export issues that could alter understanding, find the "¿" symbols (in SI)

(8) Bottom panels of Fig 5 are hard to read on paper.

(Remarks on code availability)

Reviewer #3

(Remarks to the Author)

This is a fascinating study with a clear hypothesis, a suitable experimental approach, and interesting results. The text is clear and well supported by figures, videos and other supplementary material.

Specific comments:

In some videos, some cells were practically immotile while others were actively swimming. I presume each video represents one strain. Is there also a systematic dichotomy of motility and starvation response within individual strains?

Line 16: It sounds contradictory that bacteria contribute to carbon storage by remineralizing organic carbon.

Line 49: "a marine particle" is so poorly defined that the calculation of carbon supply for 103 to 104 new bacteria is not meaningful.

Line 136: "bacteria can control motility independently from flagellation" sounds misleading since cells are immotile without flagella (but in the further context it becomes clear).

Line 338: How does the additional energy cost by swimming faster balance out with the increasing chance/frequency of encountering a food particle?

Line 340: What is the unit of reorientation frequency?

Line 411 etc.: How does the vertical component of swimming affect the calculated swimming speed?

Fig. 1A legend: It is not clear how orange color illustrates a ceased motility in this frame.

Fig. 1D legend: Mention the meaning of orange vs. blue species names.

Fig. 2A: It is difficult to distinguish the purple color of flagellar filaments.

Fig. 5D: Please explain the red X's.

Videos: The names of the videos (e.g. 29201_0_video_336844_sg1jty.mp4) do not relate to their names in the text (e.g. Supplementary Video S1).

(Remarks on code availability)

I am not sure which code is referred to here.

Decision Letter:

13th September 2024

Dear Roman,

Thank you for your patience while your manuscript "Risk-reward trade-off in motility endurance generates dichotomy in search strategies among copiotrophic marine bacteria" was under peer-review at Nature Microbiology. I'm sorry this took quite so long-- this summer has proven to be a difficult time to find and secure referees. Nevertheless, your work has now been seen by 3 referees, whose expertise and comments you will find at the end of this email. Despite the delay, I am happy to say that they find your work of considerable potential interest. They have, however, raised a number of technical and editorial concerns that will need to be addressed before we can consider publication of the work in Nature Microbiology.

In particular, Reviewer #2 raised a number of technical questions regarding the set up and analysis of the tracking experiments, and these will need to be cleared up with additional explanation or data. On the more editorial side, Reviewer #1 was concerned that the extrapolations to ocean carbon storage, carbon cycling and adaptations to oligotrophic environments were a stretch without additional data from the field to support this, and we agree that this should be toned down and adequately caveated when discussed.

Should further experimental data allow you to address these criticisms, we would be happy to look at a revised manuscript.

We strongly support public availability of data. Please place the data used in your paper into a public data repository, if one exists, or alternatively, present the data as Source Data or Supplementary Information. If data can only be shared on request, please explain why in your Data Availability Statement, and also in the correspondence with your editor. For some data types,

deposition in a public repository is mandatory - more information on our data deposition policies and available repositories can be found at <https://www.nature.com/nature-research/editorial-policies/reporting-standards#availability-of-data>.

Please include a data availability statement as a separate section after Methods but before references, under the heading "Data Availability". This section should inform readers about the availability of the data used to support the conclusions of your study. This information includes accession codes to public repositories (data banks for protein, DNA or RNA sequences, microarray, proteomics data etc...), references to source data published alongside the paper, unique identifiers such as URLs to data repository entries, or data set DOIs, and any other statement about data availability. At a minimum, you should include the following statement: "The data that support the findings of this study are available from the corresponding author upon request", mentioning any restrictions on availability. If DOIs are provided, we also strongly encourage including these in the Reference list (authors, title, publisher (repository name), identifier, year). For more guidance on how to write this section please see: <http://www.nature.com/authors/policies/data/data-availability-statements-data-citations.pdf>

* If you have not done so already we suggest that you begin to revise your manuscript so that it conforms to our Article format instructions at <http://www.nature.com/nmicrobiol/info/final-submission>. Refer also to any guidelines provided in this letter.

When submitting the revised version of your manuscript, please pay close attention to our [href="https://www.nature.com/nature-portfolio/editorial-policies/image-integrity">Digital Image Integrity Guidelines.](https://www.nature.com/nature-portfolio/editorial-policies/image-integrity) and to the following points below:

Link Redacted

Note: This url links to your confidential homepage and associated information about manuscripts you may have submitted or be reviewing for us. If you wish to forward this e-mail to co-authors, please delete this link to your homepage first.

Nature Microbiology is committed to improving transparency in authorship. As part of our efforts in this direction, we are now requesting that all authors identified as 'corresponding author' on published papers create and link their Open Researcher and Contributor Identifier (ORCID) with their account on the Manuscript Tracking System (MTS), prior to acceptance. This applies to primary research papers only. ORCID helps the scientific community achieve unambiguous attribution of all scholarly contributions. You can create and link your ORCID from the home page of the MTS by clicking on 'Modify my Springer Nature account'. For more information please visit [please visit www.springernature.com/orcid](http://www.springernature.com/orcid).

If you wish to submit a suitably revised manuscript we would hope to receive it within 6 months. If you cannot send it within this time, please let us know. We will be happy to consider your revision, even if a similar study has been accepted for publication at Nature Microbiology or published elsewhere (up to a maximum of 6 months).

Yours sincerely,

Reviewer Expertise:

Referee #1: marine microbiology, biophysics, chemotaxis
Referee #2: chemotaxis, biophysics, tracking experiments
Referee #3: marine microbiology, biological oceanography, biogeochemistry

Reviewer Comments:

Reviewer #1 (Remarks to the Author):

The manuscript is like the kinetic and static bacteria the authors report. It is bimodal with strong points and a weak point. The primary strength is the report of bimodal limokinetic and limostatic bacteria. There are some excellent microbiology techniques and analyses. The weakness is the connection between the change of bacterial motility among strains and carbon storage, which is mentioned in the first and last sentences of the abstract and never mentioned again.

The weakness is easily resolved by removing those sentences and emphasising the significance of the dichotomy, which is a next step in our understanding of how bacteria make decisions and which pathways and strategies they have available to survive and thrive. This is a significant conceptual advance buttressed by multifaceted experimental tests of the concept. This should generate a whole sub-sub-discipline of papers examining this concept in multiple species, with different media and in the environment.

The results are particularly significant if the dichotomy holds for most motile taxa; chemotaxis does not blur or erase the effect and is maintained beyond lab cultures. These are the major conceptual points for and against publication. Below are technical strengths and weaknesses.

The dichotomy is strong and clear. Exploring alternative energy sources is excellent and demonstrates that the motile bacteria are not 'cheating'. However, their cutoff could force the dichotomy in that the bacteria are rated as motile or non-motile. They have many excellent tracking videos, and Figure 1 shows more of a bimodal distribution than solely the binary static versus kinetic outcome. The dichotomy is conceptually useful, but in future papers on this topic, it may weaken or blur the distribution of strategies. While they have described the dichotomy, it may be worth the authors pointing out that Figure 1 suggests intermediate strategies. The revelation of this work is that there are multiple strategies. The flagellar loss is a strategy, but do some bacteria use proteases to remineralise the flagella and take up the amino acids? This question is for a future paper but illustrates just one line of enquiry this paper reveals.

The authors have clearly considered alternative strategies, albeit from the dichotomy viewpoint. They convincingly rule out alternative energy sources, such as necromass recycling and light. This confirms that motility is fueled by internal biomass conversion. Showing that cell numbers remain the same but that biomass decreases is a worthwhile result rarely or never highlighted in the literature, adding significance to the manuscript. Their meticulous approach to this and the entire manuscript strengthens the study's conclusions.

The authors' ability to predict the strategy of other taxa is outstanding and speaks to the thoroughness of their work. Their investigation of their own biases further underscores the meticulousness of their approach.

Extrapolating carbon cycling and adaptations to oligotrophic environments is an overreach without experiments on bacteria directly from the ocean. Having the genetic database is excellent. However, their experiments minimised or eliminated nutrient gradients, and the ocean and other environments are full of them. This leads to the question of whether chemotaxis in the ocean alters behaviour, possibly breaking down the dichotomy.

Reviewer #2 (Remarks to the Author):

In this manuscript, the authors ask if and how long (copiotrophic) marine bacteria retain motility after sudden starvation. They highlight that motility (and chemotaxis), although metabolically extremely costly, may allow bacteria to find nutrient hotspots in a dilute environment. Therefore, there is likely to be a high risk/reward trade-off between retaining motility or not during starvation. To investigate the strategy adopted by copiotrophic marine bacteria, the authors selected 26 isolates (18 species) and measured their motile fraction and swimming speed just before and after carbon starvation (up to almost 2 days). This approach is a major strength of the paper, as motility studies are often limited to either (i) different isolates of the same species (with more homogeneous behaviors), (ii) only a few species, or (iii) laboratory type strains.

Interestingly, rather than a continuous behavioral landscape, they observed two distinct poles: either cells 'persist' and retain flagellation for long periods of time, or they rapidly lose their flagella. They suggest names for these two categories and propose a genome-based classifier with reasonable performance against phenotypic determination to assign a marine species to one of the two categories. While I will have comments on such a classification, I note here that in microbiome fields it is much easier to simply highlight diversity than to propose overarching categories in a reasonably convincing way, as the authors do here.

The authors then extend their work by noting that the distribution of these strategies (as detected by their classifier in the GOMicrobiome) is not homogeneous, suggesting that local environmental constraints shape the precise risk/reward trade-off.

Overall, the writing is clear, the methods and results are extremely well described, and the discussion is of high interest. I would like to thank the authors for making it such an easy and insightful read. The supplementary material is also of high quality and interest, with careful considerations on numerous topics, and further compensates for the need for simplifications in the main text (in part imposed by the heterogeneous dataset of isolates).

Major remarks

(1) Authors provide an excellent technical summary of the experiments done to support their claims, in Figure S2C.

I notice that two strains only have one experiment (replicate), and these two strains show a very low motility fraction even before carbon starvation. Given the important variability observed in motile fraction even in showcase examples (Fig.2B), it seems to me like insufficient data.

(2) Methods for tracking:

a. (l. 386) "Bacteria were observed within 15 minutes after sampling."

Cells can stay up to 15 minutes inside the closed motility chamber before an acquisition is done, somewhere in that chamber.

Where? In the bulk or along glass surfaces? The movie suggests the first option, but the Methods should mention such information. More importantly, I am concerned about biases induced by the time for a non-negligible part of the motile population

to accumulate at surfaces. Bacteria tend to remain swimming along surfaces once they reach them, exiting occasionally (especially due to turns): altogether, this usually biases bulk acquisition (made several minutes after injecting the cells) against fast low-turning-rate bacteria and will also bias (lower) the bulk motile/non-motile ratio. Did the author assess such biases? I believe it could explain part of the variability they observe. How would that impact their conclusion?

b. (II.417-418) "Cell tracking was performed [...] after removing the background from each 418 image by subtracting the median image computed over the entire video."

If using a background correction on the whole movie, aren't the authors removing a part of the non-motile cells? The described method takes the median value of each pixel over ~14s (25-30 fps, 418 frames), a duration I do not expect to be sufficient for a non-motile cell to diffuse enough to not be erased by the background correction. How is that not biasing the motile fraction computation?

(3) (II. 82-83) "All strains exhibited a run-reverse-flick motility pattern (Supplementary Videos S5-S10), a hallmark of single-flagellated bacteria [30, 31]."

While most of the video do show cases run-reverse-flick (RRF) noticeable by eye, it does not appear to me as sufficient proof to sustain an exclusive claim. Indeed, the authors will know that under a speed threshold (dependent on species) the buckling instability causing the flick every other turn will not happen, therefore leading to a run-reverse (RR) motility pattern. RR patterns can cohabit with RRF patterns. RR patterns notably have a different (lower) diffusion coefficient.

(4) Authors make use of an equation from Taktikos et al. (eqn 23 in the article) for computing the diffusion coefficient from rotational diffusion, average swimming speed and turning frequency.

This equation assumes RRF motility with identical forward and backward run duration and constant speed. I find the usage of the diffusion coefficient as a quantitative metric in this manuscript very dubious due to (1) the likelihood of RR patterns coexisting with RRF patterns, (2) the potential biases in turning frequency and swimming speed mentioned above, (3) the absence of justification regarding a symmetrical pattern (likely beyond capabilities of the tracking method here), (4) the usage of a single rotational diffusion value despite strong reasons to think it would vary from species to species (e.g. different body sizes as shown in Figure S10 A). I understand the need for estimates of such value in order to develop supplementary discussion, and I appreciate it as done in Supplementary Discussion 2.2, but it is clearly a stretch to provide the diffusion coefficient of all strains at different time points (Supplementary Discussion 2.1, SFigure 7C) and comment on differences between limostatic and limokinetic strains.

Minor remarks

(5) Could the author comment on the (less likely) possibility that the dichotomy limostatic/limokinetic they observe is just a difference in response to the sudden withdrawal of an unknown trace elements initially present in MB, rather than carbon source?

(6) Is there any hint within the data for a difference in dispersal strategy between different limokinetic strains?

Nonconsequential remarks

(7) Export issues that could alter understanding, find the "¿" symbols (in SI)

(8) Bottom panels of Fig 5 are hard to read on paper.

Reviewer #3 (Remarks to the Author):

This is a fascinating study with a clear hypothesis, a suitable experimental approach, and interesting results. The text is clear and well supported by figures, videos and other supplementary material.

Specific comments:

In some videos, some cells were practically immotile while others were actively swimming. I presume each video represents one strain. Is there also a systematic dichotomy of motility and starvation response within individual strains?

Line 16: It sounds contradictory that bacteria contribute to carbon storage by remineralizing organic carbon.

Line 49: "a marine particle" is so poorly defined that the calculation of carbon supply for 103 to 104 new bacteria is not meaningful.

Line 136: "bacteria can control motility independently from flagellation" sounds misleading since cells are immotile without flagella (but in the further context it becomes clear).

Line 338: How does the additional energy cost by swimming faster balance out with the increasing chance/frequency of encountering a food particle?

Line 340: What is the unit of reorientation frequency?

Line 411 etc.: How does the vertical component of swimming affect the calculated swimming speed?

Fig. 1A legend: It is not clear how orange color illustrates a ceased motility in this frame.

Fig. 1D legend: Mention the meaning of orange vs. blue species names.

Fig. 2A: It is difficult to distinguish the purple color of flagellar filaments.

Fig. 5D: Please explain the red X's.

Videos: The names of the videos (e.g. 29201_0_video_336844_sg1jty.mp4) do not relate to their names in the text (e.g. Supplementary Video S1).

Reviewer #3 (Remarks on code availability):

I am not sure which code is referred to here.

Version 1:

Reviewer comments:

Reviewer #2

(Remarks to the Author)

In their rebuttal and revised manuscript, the authors have convincingly addressed my comments and made all the changes that I still consider important after our exchange. The limitations of the methods and the approximations made are now clear and clearly do not affect the conclusions.

I apologise for the misunderstanding in my comment 2b, which was due to an error on my part. The authors have nevertheless assessed the technical bias I feared, despite a much lower probability than I originally thought. I think this is a good example of the rigorous work done here.

(Remarks on code availability)

Reviewer #3

(Remarks to the Author)

The authors have carefully responded to all comments by the reviewers and the manuscript is now in a fine shape.

(Remarks on code availability)

I expect reviewer #2, who clearly has expertise in this specific field, to review the code. This is not my competence.

Reviewer #4

(Remarks to the Author)

The authors provided a fascinating analysis of bacterial strategy with a large set of experiments to evaluate their prediction. I would like to thank the editor in appointing a reviewer specifically to have a closer look at the code in our data-centric biology. I thank the authors for providing the data and the code to the reviewers as part of the review process. However, the code lacks documentation in its current state to allow for an easy replication of the code or understanding of the shared data. I understand that the main research outcome of the authors is not a software, but all items in the Code and Software Submission Checklist from Nature have been ticked when not all the information is actually provided. A couple of improvements regarding the documentation and sharing the model would benefit the transparency of the methods as well as increase the scientific outreach of the work. I answered to the short feedback questions first and in more details below.

Code review feedback

- Are you satisfied that all data and source code needed to reproduce the results of the paper have been made available? Yes
- Are you satisfied that the results can be replicated using the code/software and dataset provided in the study? No
- Were you able to run the tool successfully? Yes
- Was the code sufficiently documented to allow another researcher to follow the algorithm? No
- Can the software be run on a widely available operating system? Yes
- To your knowledge, do available tools or software exist that perform in a similar way to the reported software? No

Charlie Pauvert

(Remarks on code availability)

I had a particular look at the R code for the classifier as it support a main claim of the paper and I could run the scripts after manual corrections.

Whilst the R packages are cited, there is no indication of the versions used, neither in the manuscript nor in the code nor the README.

A specific suggestion to help here would be for the authors to use the `renv` package (<https://cloud.r-project.org/package=renv>) that tracks versions of packages within a project. It makes installing the set of packages way much easier for collaborators, reviewers and readers.

This would also spot earlier on two issues I faced:

- a missing R dependency not specified in the code: the package `klaR`. Error in { : task 1 failed "there is no package called 'klaR'"
- a remaining typo in the file ``classifier/classifier/training_prediction/limostatic.R`` L6 "library(caret)m"

Documentation is scarce and scattered in multiple readme files (`readme.rtf`, `README.md`, `readme_DHM_images.rtf`, `readme_20241128.rtf`), making it hard for reviewers and readers to know where to start. The main readme file at the root of the code and data archive should point to others READMEs (perhaps with consistent naming and format) to guide the reviewers and readers of the type of input needed, how to install the dependencies, what are the scripts and which output files are going to be written.

Many files present within the code and data archives are actually not described in the documentation/README files when entering an analysis directory.

I found the readme for tracking code to be nicely written as a guide, but the authors could have provided a list of Python packages to install the dependencies (in the form of a command in the README to copy in order to create a conda/mamba environment for instance).

There are some nestedness problems with the directories that needs to be fixed to evaluate the code. A specific suggestion to help here would be for the authors to use the `here` package (<https://cloud.r-project.org/package=here>) to refer to paths in the archives.

For example, the script ``classifier/classifier/with_depth/depth.R`` requires data that is not available in the working directory.

- L25 or L31: `load("limostatic.RData")` when it should be `load("../training_prediction/limostatic.RData")`
- L36-38: where `tsv` files are expected to be under "depth" directory which had been renamed to "with_depth".

Precomputed data in the form of `RData` files is presumably shared which is a good idea. However, it is made worthless without explanation (nor in any README or in code comments), for instance loading `limokinetic.RData` in R allows to explore the following objects:

- master
- nb
- x

These objects should be documented in the README within the `training_prediction` to understand them and to allow reviewers and readers that could not re run the model fit to evaluate the model.

In that regard, if the authors provided few key instructions on how to predict the studied motility behaviour using their model, this would extend the scientific outreach of the work and help understanding the methods. For instance, making available a small demo file of KEGG orthologs to be loaded, next to a precomputed model, from which the motility behaviour could be predicted. This detailed steps would add smaller use case compared to the ones on the enteric species (that is included in the training) or the MAGs. This would allow other groups to easily use their model to assess the prediction with different strains, taxa or biomes.

A minor point: The authors could have excluded unnecessary files from the archives for clarity (e.g., the ones generated by Mac OSX: `._MACOSX,.DS_Store`)

Decision Letter:

6th January 2025

Dear Roman,

Happy new year! I hope you and yours had a nice holiday break. Thank you as always for your patience while your manuscript "Risk-reward trade-off in motility endurance generates dichotomy in search strategies among copiotrophic marine bacteria" was under peer-review at Nature Microbiology. Your paper has been seen again by Reviewers #2 and #3, who are happy with the revisions and have no additional comments. This round we also brought in a new reviewer, Reviewer #4, to delve into the code to ensure that it is documented properly and reproducible (this is a relatively new policy that we have started at NMicro). You will see from Reviewer #4's comments below that while they did not have any major issues with the code, there are a number points that will be important to correct before we can move forward. My sense is that for you and your team this will be very quick and easy (I hope!), and of course we are still very interested in the possibility of publishing your study in Nature Microbiology, but we would like to consider your response to these concerns in the form of a revised manuscript before we make a final decision on publication. Once again...my hope is that this will be quick and painless! Let me know if you have any questions or concerns.

If you have not done so already please begin to revise your manuscript so that it conforms to our Article format instructions at <http://www.nature.com/nmicrobiol/info/final-submission/>

The usual length limit for a Nature Microbiology Article is six display items (figures or tables) and 3,000 words. We have some flexibility, and can allow a revised manuscript at 3,500 words, but please consider this a firm upper limit. There is a trade-off of ~250 words per display item, so if you need more space, you could move a Figure or Table to Supplementary Information.

Some reduction could be achieved by focusing any introductory material and moving it to the start of your opening 'bold' paragraph, whose function is to outline the background to your work, describe in a sentence your new observations, and explain your main conclusions. The discussion should also be limited. Methods should be described in a separate section following the discussion, we do not place a word limit on Methods.

Nature Microbiology titles should give a sense of the main new findings of a manuscript, and should not contain punctuation. Please keep in mind that we strongly discourage active verbs in titles, and that they should ideally fit within 90 characters each (including spaces).

Please include a data availability statement as a separate section after Methods but before references, under the heading "Data Availability". This section should inform readers about the availability of the data used to support the conclusions of your study. This information includes accession codes to public repositories (data banks for protein, DNA or RNA sequences, microarray, proteomics data etc...), references to source data published alongside the paper, unique identifiers such as URLs to data repository entries, or data set DOIs, and any other statement about data availability. At a minimum, you should include the following statement: "The data that support the findings of this study are available from the corresponding author upon request", mentioning any restrictions on availability. If DOIs are provided, we also strongly encourage including these in the Reference list (authors, title, publisher (repository name), identifier, year). For more guidance on how to write this section please see: <http://www.nature.com/authors/policies/data/data-availability-statements-data-citations.pdf>

To improve the accessibility of your paper to readers from other research areas, please pay particular attention to the wording of the paper's opening bold paragraph, which serves both as an introduction and as a brief, non-technical summary in about 150 words. If, however, you require one or two extra sentences to explain your work clearly, please include them even if the paragraph is over-length as a result. The opening paragraph should not contain references. Because scientists from other sub-disciplines will be interested in your results and their implications, it is important to explain essential but specialised terms concisely. We suggest you show your summary paragraph to colleagues in other fields to uncover any problematic concepts.

If your paper is accepted for publication, we will edit your display items electronically so they conform to our house style and will reproduce clearly in print. If necessary, we will re-size figures to fit single or double column width. If your figures contain several parts, the parts should form a neat rectangle when assembled. Choosing the right electronic format at this stage will speed up the processing of your paper and give the best possible results in print. We would like the figures to be supplied as vector files - EPS, PDF, AI or postscript (PS) file formats (not raster or bitmap files), preferably generated with vector-graphics software (Adobe Illustrator for example). Please try to ensure that all figures are non-flattened and fully editable. All images should be at least 300 dpi resolution (when figures are scaled to approximately the size that they are to be printed at) and in RGB colour format. Please do not submit Jpeg or flattened TIFF files. Please see also 'Guidelines for Electronic Submission of Figures' at the end of this letter for further detail.

Figure legends must provide a brief description of the figure and the symbols used, within 350 words, including definitions of any error bars employed in the figures.

When submitting the revised version of your manuscript, please pay close attention to our [href="https://www.nature.com/nature-research/editorial-policies/image-integrity">Digital Image Integrity Guidelines](https://www.nature.com/nature-research/editorial-policies/image-integrity) and to the following points below:

EXTENDED DATA FIGURES

Please include a statement before the acknowledgements naming the author to whom correspondence and requests for materials should be addressed.

Finally, we require authors to include a statement of their individual contributions to the paper -- such as experimental work, project planning, data analysis, etc. -- immediately after the acknowledgements. The statement should be short, and refer to authors by their initials. For details please see the Authorship section of our joint Editorial policies at http://www.nature.com/authors/editorial_policies/authorship.html

* include a point-by-point response to any editorial suggestions and to our referees. Please include your response to the editorial suggestions in your cover letter, and please upload your response to the referees as a separate document.

* ensure it complies with our format requirements for Letters as set out in our guide to authors at www.nature.com/nmicrobiol/info/gta/

* state in a cover note the length of the text, methods and legends; the number of references; number and estimated final size of figures and tables

* resubmit electronically if possible using the link below to access your home page:

Link Redacted

*This url links to your confidential homepage and associated information about manuscripts you may have submitted or be reviewing for us. If you wish to forward this e-mail to co-authors, please delete this link to your homepage first.

Please ensure that all correspondence is marked with your Nature Microbiology reference number in the subject line.

Nature Microbiology is committed to improving transparency in authorship. As part of our efforts in this direction, we are now requesting that all authors identified as 'corresponding author' on published papers create and link their Open Researcher and Contributor Identifier (ORCID) with their account on the Manuscript Tracking System (MTS), prior to acceptance. This applies to primary research papers only. ORCID helps the scientific community achieve unambiguous attribution of all scholarly contributions. You can create and link your ORCID from the home page of the MTS by clicking on 'Modify my Springer Nature account'. For more information please visit please visit www.springernature.com/orcid.

We hope to receive your revised paper within three weeks. If you cannot send it within this time, please let us know.

Yours sincerely,

Reviewers Comments:

Reviewer #2 (Remarks to the Author):

In their rebuttal and revised manuscript, the authors have convincingly addressed my comments and made all the changes that I still consider important after our exchange. The limitations of the methods and the approximations made are now clear and clearly do not affect the conclusions.

I apologise for the misunderstanding in my comment 2b, which was due to an error on my part. The authors have nevertheless assessed the technical bias I feared, despite a much lower probability than I originally thought. I think this is a good example of the rigorous work done here.

Reviewer #3 (Remarks to the Author):

The authors have carefully responded to all comments by the reviewers and the manuscript is now in a fine shape.

Reviewer #3 (Remarks on code availability):

I expect reviewer #2, who clearly has expertise in this specific field, to review the code. This is not my competence.

Reviewer #4 (Remarks to the Author):

The authors provided a fascinating analysis of bacterial strategy with a large set of experiments to evaluate their prediction. I would like to thank the editor in appointing a reviewer specifically to have a closer look at the code in our data-centric biology. I thank the authors for providing the data and the code to the reviewers as part of the review process. However, the code lacks documentation in its current state to allow for an easy replication of the code or understanding of the shared data. I understand that the main research outcome of the authors is not a software, but all items in the Code and Software Submission Checklist from Nature have been ticked when not all the information is actually provided. A couple of improvements regarding the documentation and sharing the model would benefit the transparency of the methods as well as increase the scientific outreach of the work. I answered to the short feedback questions first and in more details below.

Code review feedback

- Are you satisfied that all data and source code needed to reproduce the results of the paper have been made available? Yes
- Are you satisfied that the results can be replicated using the code/software and dataset provided in the study? No
- Were you able to run the tool successfully? Yes
- Was the code sufficiently documented to allow another researcher to follow the algorithm? No
- Can the software be run on a widely available operating system? Yes
- To your knowledge, do available tools or software exist that perform in a similar way to the reported software? No

Charlie Pauvert

Reviewer #4 (Remarks on code availability):

I had a particular look at the R code for the classifier as it support a main claim of the paper and I could run the scripts after manual corrections.

Whilst the R packages are cited, there is no indication of the versions used, neither in the manuscript nor in the code nor the README.

A specific suggestion to help here would be for the authors to use the renv package (<https://cloud.r-project.org/package=renv>) that tracks versions of packages within a project. It makes installing the set of packages way much easier for collaborators, reviewers and readers.

This would also spot earlier on two issues I faced:

- a missing R dependency not specified in the code: the package klaR. Error in { : task 1 failed "there is no package called 'klaR'"
- a remaining typo in the file ``classifier/classifier/training_prediction/limostatic.R`` L6 "library(caret)m"

Documentation is scarce and scattered in multiple readme files (readme.rtf, README.md, readme_DHM_images.rtf, readme_20241128.rtf), making it hard for reviewers and readers to know where to start. The main readme file at the root of the code and data archive should point to others READMEs (perhaps with consistent naming and format) to guide the reviewers and readers of the type of input needed, how to install the dependencies, what are the scripts and which output files are going to be written.

Many files present within the code and data archives are actually not described in the documentation/README files when entering an analysis directory.

I found the readme for tracking code to be nicely written as a guide, but the authors could have provided a list of Python packages to install the dependencies (in the form of a command in the README to copy in order to create a conda/mamba environment for instance).

There are some nestedness problems with the directories that needs to be fixed to evaluate the code. A specific suggestion to help here would be for the authors to use the here package (<https://cloud.r-project.org/package=here>) to refer to paths in the archives.

For example, the script ``classifier/classifier/with_depth/depth.R`` requires data that is not available in the working directory.

- L25 or L31: `load('limostatic.RData')` when it should be `load('../training_prediction/limostatic.RData')`
- L36-38: where tsv files are expected to be under "depth" directory which had been renamed to "with_depth".

Precomputed data in the form of RData files is presumably shared which is a good idea. However, it is made worthless without explanation (nor in any README or in code comments), for instance loading `limokinetic.RData` in R allows to explore the following objects:

- master
- nb
- x

These objects should be documented in the README within the training_prediction to understand them and to allow reviewers and readers that could not re run the model fit to evaluate the model.

In that regard, if the authors provided few key instructions on how to predict the studied motility behaviour using their model, this would extend the scientific outreach of the work and help understanding the methods. For instance, making available a small demo file of KEGG orthologs to be loaded, next to a precomputed model, from which the motility behaviour could be predicted. This detailed steps would add smaller use case compared to the ones on the enteric species (that is included in the training) or the MAGs. This would allow other groups to easily use their model to assess the prediction with different strains, taxa or biomes.

A minor point: The authors could have excluded unnecessary files from the archives for clarity (e.g., the ones generated by Mac OSX: ._MACOSX,.DS_Store)

Version 2:

Reviewer comments:

Reviewer #4

(Remarks to the Author)

I thank the authors for considering my comments regarding the code and data, and I consider they clearly did improve the documentation and addressed my concerns regarding the transparency of the methods.

(Remarks on code availability)

The authors had made the code available earlier, it is now associated with a better documentation and examples.

Decision Letter:

24th February 2025

Dear Roman,

Thank you for your patience while your manuscript "Risk-reward trade-off in motility endurance generates dichotomy in search strategies among copiotrophic marine " was under peer-review at Nature Microbiology. It has now been seen the final referee, whose and comments you will find at the of this email. I have also now had the chance to discuss your work with my colleagues, and I am happy to say that we would like to offer to publish your work, in principle... however, before we can move forward and send official notice of this, and proceed with our final editorial and reproducibility checks, there is one more thing that I need to ask of you. In going over your manuscript files, I noticed that the Extended Data figures are bundled into the SI PDF. We will need these to be uploaded individually to our system as ED figures, rather than being in that PDF--this is because they will now need to be run through our research integrity imagine screening protocol (which is standard for papers that enter the pre-accept pipeline, as yours is about to!).

So, I am going to send a "revise" decision here, which will bounce the paper back to you. If you could just upload those ED figs separately and resubmit, that would be great. As soon as you do that, I will get your the official Accept In Principle letter. Please do not change anything else about the manuscript at this point--we'll get back to you with final editorial revisions in due time. No need to submit a new cover letter or anything...just re-upload everything else as it was. Drop me a line once this is done and resubmitted and I will move it along as quickly as I can.

More details...

EXTENDED DATA FIGURES

* resubmit electronically if possible using the link below to access your home page:

Link Redacted

*This url links to your confidential homepage and associated information about manuscripts you may have submitted or be reviewing for us. If you wish to forward this e-mail to co-authors, please delete this link to your homepage first.

Thanks!

Reviewers Comments:

Reviewer #4 (Remarks to the Author):

I thank the authors for considering my comments regarding the code and data, and I consider they clearly did improve the documentation and addressed my concerns regarding the transparency of the methods.

Reviewer #4 (Remarks on code availability):

The authors had made the code available earlier, it is now associated with a better documentation and examples.

Version 3:

Decision Letter:

Our ref: NMICROBIOL-24071965C

25th February 2025

Dear Roman,

Thank you for submitting your revised manuscript "Risk-reward trade-off in motility endurance generates dichotomy in search strategies among copiotrophic marine bacteria" (NMICROBIOL-24071965C). It has now been seen by the original referees and their comments are below. The reviewers find that the paper has improved in revision, and therefore we'll be happy in principle to publish it in Nature Microbiology, pending minor revisions to comply with our editorial and formatting guidelines.

If the current version of your manuscript is in a PDF format, please email us a copy of the file in an editable format (Microsoft Word or LaTeX)-- we can not proceed with PDFs at this stage (probably should have asked for that before--sorry to keep coming up with new things I need...I have done this before, I swear!!).

Thank you again for your interest in Nature Microbiology Please do not hesitate to contact me if you have any questions.

Sincerely,

Version 4:

Decision Letter:

25th March 2025

Dear Roman and Johannes,

I am pleased to accept your Article "Risk-reward trade-off during carbon starvation generates dichotomy in motility endurance among marine bacteria" for publication in Nature Microbiology. Thank you for having chosen to submit your work to us and many congratulations.

Once your paper is typeset, you will receive an email with a link to choose the appropriate publishing options for your paper and our Author Services team will be in touch regarding any additional information that may be required. Once your paper has been

scheduled for online publication, the Nature press office will be in touch to confirm the details.

You may wish to make your media relations office aware of your accepted publication, in case they consider it appropriate to organize some internal or external publicity. Once your paper has been scheduled you will receive an email confirming the publication details. This is normally 3-4 working days in advance of publication. If you need additional notice of the date and time of publication, please let the production team know when you receive the proof of your article to ensure there is sufficient time to coordinate. Further information on our embargo policies can be found here:

<https://www.nature.com/authors/policies/embargo.html>

Authors may need to take specific actions to achieve [compliance](https://www.springernature.com/gp/open-research/funding/policy-compliance-faqs) with funder and institutional open access mandates. If your research is supported by a funder that requires immediate open access (e.g. according to [Plan S principles](https://www.springernature.com/gp/open-research/plan-s-compliance)) then you should select the gold OA route, and we will direct you to the compliant route where possible. For authors selecting the subscription publication route, the journal's standard licensing terms will need to be accepted, including [self-archiving policies](https://www.nature.com/nature-portfolio/editorial-policies/self-archiving-and-license-to-publish). Those licensing terms will supersede any other terms that the author or any third party may assert apply to any version of the manuscript.

With kind regards,

P.S. Click on the following link if you would like to recommend Nature Microbiology to your librarian
<http://www.nature.com/subscriptions/recommend.html#forms>

** Visit the Springer Nature Editorial and Publishing website at http://editorial-jobs.springernature.com?utm_source=ejP_NMicro_email&utm_medium=ejP_NMicro_email&utm_campaign=ejp_NMicro for more information about our career opportunities. If you have any questions please click [here](mailto:editorial.publishing.jobs@springernature.com).

Response to Reviewer 1

Reviewer comments in black, Author response in blue

Text citations indented, with highlighted changes in bold.

The manuscript is like the kinetic and static bacteria the authors report. It is bimodal with strong points and a weak point. The primary strength is the report of bimodal limokinetic and limostatic bacteria. There are some excellent microbiology techniques and analyses. The weakness is the connection between the change of bacterial motility among strains and carbon storage, which is mentioned in the first and last sentences of the abstract and never mentioned again. The weakness is easily resolved by removing those sentences and emphasizing the significance of the dichotomy, which is a next step in our understanding of how bacteria make decisions and which pathways and strategies they have available to survive and thrive. This is a significant conceptual advance buttressed by multifaceted experimental tests of the concept. This should generate a whole sub-sub-discipline of papers examining this concept in multiple species, with different media and in the environment. The results are particularly significant if the dichotomy holds for most motile taxa; chemotaxis does not blur or erase the effect and is maintained beyond lab cultures.

We thank the Reviewer for their criticism and praise and we agree the connection between motility and carbon storage could have been better explained. However, understanding how microscale interactions contribute to macro scale processes is a major motivation for this work, and therefore we decided instead of removing those sentences to expand on them, and we have added a paragraph to the introduction (LL 35):

The association of marine bacteria with nutrient hotspots affects important oceanic elemental fluxes. An important type of hotspots are sinking marine particles, consisting of organic matter fixed by primary producers, that sink to the bottom of the ocean and thereby effectively sequester carbon from the atmosphere [8]. However, copiotrophic marine bacteria colonize and degrade these particles before they reach the ocean floor, and are responsible for up to 29 % of the carbon flux attenuation [9] and act as gatekeepers of the vertical carbon flux [10, 11]. In many areas, such as in the open ocean [12], particles are very dilute and search times can extend to days or weeks. There has been considerable effort in obtaining a more mechanistic understanding of the exploitation of the organic matter in marine particles [13, 14, 15], but the degradation process ultimately hinges on the successful colonization of these particles. Unsuccessful colonization is suggested to be a key driver for some particles escaping degradation [16]. Therefore, a better understanding of the bacterial search process for particles may better constrain models of the microbial contribution to carbon flux in the ocean.

We share excitement about the potential for future studies.

These are the major conceptual points for and against publication. Below are technical strengths and weaknesses.

To facilitate the response, we have separated the comments into a numbered list.

1. The dichotomy is strong and clear. Exploring alternative energy sources is excellent and demonstrates that the motile bacteria are not ‘cheating’. However, their cutoff could force the dichotomy in that the bacteria are rated as motile or non-motile. They have many excellent tracking videos, and Figure 1 shows more of a bimodal distribution than solely the binary static

versus kinetic outcome. The dichotomy is conceptually useful, but in future papers on this topic, it may weaken or blur the distribution of strategies. While they have described the dichotomy, it may be worth the authors pointing out that Figure 1 suggests intermediate strategies. The revelation of this work is that there are multiple strategies.

We thank the Reviewer for recognizing that the dichotomy is a strong and clear feature of our dataset. As indicated by the Reviewer, Fig. 1 suggests a bimodal distribution: this bimodality naturally supports the primary classification of the strains in two behavioral categories (i.e., limokinetic and limostatic). Of course, there may be secondary variation within these modes that can be interesting, but the point of the figure remains that there are two main classes: strains that lose motility rapidly and strains that retain motility for at least 2 days.

The dichotomy is strengthened if we observe the results from Fig. 1 (dichotomy in motile fraction during starvation) through the lens of the later figures in the manuscript, especially of the biomass loss in limokinetic strains (Fig. 3) and the flagellar loss in limostatic strains (Fig. 2). Flagellar loss in limostatic species indicates they do not just pause motility, but commit to a nonmotile lifestyle. And the fraction of flagellated cells is higher than the motile fraction, especially in the limokinetic strains tested. This indicates that, although not all cells are motile, they can be motile at a different time.

Of course it cannot be ruled out that individual strains could show intermediate behavior or behavior that depends on environmental conditions. One example could be *Alteromonas* sp. 4B03, the limokinetic strain with the lowest motile fraction, and therefore we have added a statement to describe this more explicitly (LL 126):

Furthermore, the classification into limokinetic and limostatic strains (based on the motile fraction criterion) was the same under the two treatments, with a single exception (*Alteromonas* sp. 4B03 was classified as limostatic in stationary phase and limokinetic in starvation medium, **and we note that this strain has a motile fraction close to the separation value between the two classes**).

Regarding the variation within the limokinetic strains: because of the high increase of encounters between non-motile and motile populations, and the high potential reward of a successful search, a low fraction of motile cells likely already generates a population-level search advantage. Assuming a 100-fold encounter boost [35], even a motile fraction of 0.05 (the lowest value for limokinetic strains in strain *Alteromonas* sp 4B03, Table S1) would still increase the population level encounter rate by a factor of 5 compared to a non-motile population. We therefore have added a paragraph on the fraction of cells that is not swimming (LL 374-386; also relating to question 1 from Reviewer 3):

Even in limokinetic populations, the average motile fraction per strain during starvation varied from 0.05 to 0.55, and hence a considerable fraction of the cells were not swimming. One reason could be that limokinetic strains are hedging their bets, where the non-motile fraction of the population preserves resources while the motile fraction searches. If the potential reward is high, even a few successful searches could propagate the population [12]. Alternatively, it is possible that cells in limokinetic strains are only motile during a fraction of the time. Many species of bacteria are able to pause motility without losing flagella [71, 28, 72], likely through a ‘clutch-like’ mechanism [73, 74]. Our observation that during starvation the flagellated fraction of cells in limokinetic strains is higher than the motile fraction indicates that these strains are also capable of pausing motility. However, the duration of these pauses, and the degree to which they are present in all of our

strains, cannot be reliably determined from our data and requires tracking the motility of individual cells over prolonged times. Furthermore, such experiments may reveal further variation in the motility and dispersal strategies of limokinetic strains.

2. The flagellar loss is a strategy, but do some bacteria use proteases to remineralise the flagella and take up the amino acids? This question is for a future paper but illustrates just one line of enquiry this paper reveals.

To our knowledge, the only example of flagellar filament resynthesis is in the predatory bacterium *Bdellovibrio bacteriovorus* (Kaplan et al, Nat Microb 2023), but there the filament is not shed but re-absorbed into the periplasm. As indicated in our manuscript (LL 132), flagellar shedding has been observed in a range of other bacteria (with loose flagellar filaments being found in the samples). Remineralisation after shedding seems difficult for bacteria. As the flagellar shedding certainly seems wasteful from a resource perspective, it is also possible that this reveals non-energetic aspects that shape the trade-off such as predation (as highlighted in the Discussion, LL 387-398).

3. The authors have clearly considered alternative strategies, albeit from the dichotomy viewpoint. They convincingly rule out alternative energy sources, such as necromass recycling and light. This confirms that motility is fueled by internal biomass conversion. Showing that cell numbers remain the same but that biomass decreases is a worthwhile result rarely or never highlighted in the literature, adding significance to the manuscript. Their meticulous approach to this and the entire manuscript strengthens the study's conclusions.

We thank the Reviewer for highlighting this result and its novelty. We were in fact able to further strengthen this aspect. In our original manuscript, the data indicated that in limokinetic strains the biomass decreased during starvation, whereas the number of cells did not, implying a decrease in single-cell biomass. We have since continued to seek more direct evidence for this conclusion and have recently succeeded in using digital holographic microscopy (DHM) to perform quantitative phase (QPI) measurements of the biomass of single cells in both a limokinetic and a limostatic strain. These measurements revealed that the average single-cell dry mass indeed decreases with starvation time for the limokinetic strain, but not for the limostatic strain. We added a paragraph describing these new results in the main text (LL 215-233) and two panels in Figure 3 (reproduced here as Fig. R1):

To further investigate the biomass loss at the single cell level, we measured the dry mass distributions from quantitative phase imaging (QPI) on individual bacterial cells [53, 54]. We compared the biomass response during starvation of limokinetic *Vibrio coralliilyticus* YB1 and limostatic *Vibrio cyclitrophicus* ZF270. During growth we found average biomass of respectively 308 ± 100 (mean \pm s.d. of three biological replicates; $1 \text{ fg} = 1 \times 10^{-15} \text{ g}$) and $333 \pm 60 \text{ fg}$ (Extended Data Figure 6f). We then measured the biomass at approximately 5, 76 and 125 h after starvation onset (Fig 3E). The biomass of limokinetic YB1 and limostatic ZF270 5h after starvation onset were similar ($p = 0.9$, Tukey HSD) at 139 fg and 138 fg , respectively. The reduction in dry mass compared to the dry mass during growth indicates that the cells have engaged in at least 1 reductive division during the first hours of starvation. After 5 days of starvation (120 hours), the biomass of limostatic ZF270 did not significantly change ($p = 0.9$, Tukey HSD) but the average dry mass of limokinetic YB1 decreased to 86 fg ($p < 0.001$, Tukey HSD) (Fig 3e). So, the limokinetic strain lost on average 54 fg/cell (Fig. 3f), corresponding to approximately 11 fg/cell/day . Such a biomass loss would be insurmountable for marine oligotrophs, with typical cell mass of 20 fg [55]. For copiotrophs, instead, it represents a daily loss of

only about $11/139=8\%$ (close to the estimate based on population biomass loss of 9.4 % per day). Assuming all converted biomass is stored as glucose (yielding ~ 30 ATP per molecule), this would yield an energy flux of about $1 \cdot 10^4$ ATP/s, equivalent to 30 % of the population to be motile at swimming velocity of $\sim 40 \mu\text{m/s}$, and equal to the average energy requirement of limokinetic strains (Fig 3a). Hence, biomass conversion can fuel bacterial motility for several days.

Figure R1: A: single-cell dry mass distributions for YB1 (blue) and ZF270 (orange) as measured using quantitative phase imaging. Number of cells per condition (n) and statistical significance using a post-hoc Tukey test are indicated (n.s.: $p > 0.05$, **: $p < 0.001$). B: Average dry mass of a population (solid lines) as a function of starvation time for ZF270 (orange) and YB1 (blue). Error bars denote the 95 % CI on the average of all the single-cell data and the circles denote the average of a single biological replicate.

4. The authors' ability to predict the strategy of other taxa is outstanding and speaks to the thoroughness of their work. Their investigation of their own biases further underscores the meticulousness of their approach.

We thank the Reviewer for the appreciation of the genomic analysis.

5. Extrapolating carbon cycling and adaptations to oligotrophic environments is an overreach without experiments on bacteria directly from the ocean. Having the genetic database is excellent. However, their experiments minimised or eliminated nutrient gradients, and the ocean and other environments are full of them. This leads to the question of whether chemotaxis in the ocean alters behaviour, possibly breaking down the dichotomy.

As suggested by the Reviewer, we have toned down the predictions for natural environments, as we agree this requires a better verification with field data, and hence we have modified the main text in several places. First, we have slightly adjusted the wording on the prediction in the abstract (LL 25). Second, regarding our hypothesis that limokinetic strains can survive in very dilute environments, we have added explicitly that this is a hypothesis that requires further testing (LL 364). Fourth, we have added a caveat that our predictions are limited to gammaproteobacteria (LL 311) and our model would need further training for predictions of other bacterial classes. Fifth, we have added a statement (LL 332) and a paragraph to the discussion that argues for the need for field data, especially outside of surface coastal waters: (LL 367-386)

Testing our predictions of the motility in natural environments requires more direct observations of motility from the field [19]. Our current prediction of the prevalence of the limokinetic strategy stands in contrast to the findings from most laboratory-based studies predicting motility loss upon nutrient depletion [27, 28, 29, 30, 31, 32, 33]. Direct measurements of motility in field samples report variable fractions of motile cells (<10 % up to 70 %) [65, 69, 70], but are limited to coastal surface waters, where the concentration of dissolved nutrients are often high. Therefore, a more systematic mapping of motility is required that extends to the open ocean and ocean interior.

We appreciate the Reviewer's concern about whether the lack of nutrient gradients in our experiments affects the representativeness of our findings. The ocean bulk concentrations of labile metabolites are low, typically in the nanomolar to picomolar range (see the review in *Nat. Microb.* by Moran and colleagues, Ref. [5]), insufficient to fuel motility. Gradients exist around hotspots, but these are highly localized and ephemeral (e.g. Brumley et al 2019, Ref. [85]) and the hotspot concentration is low outside of productive waters (see Supplementary Discussion for numbers). Hence, our experiments are representative of the prevailing bulk conditions in oligotrophic waters, where such gradients are rare and cannot affect the dichotomy. Our work shows that there are strains that are motile in these conditions (limokinetic) and hence are expected to be able to perform chemotaxis upon encountering a gradient, but strains that lose motility (limostatic) will not. Therefore, chemotaxis in the ocean cannot break down the dichotomy.

Response to Reviewer 2

Reviewer comments in black, **Author response in blue**

Text citations indented, with highlighted changes in bold.

In this manuscript, the authors ask if and how long (copiotrophic) marine bacteria retain motility after sudden starvation. They highlight that motility (and chemotaxis), although metabolically extremely costly, may allow bacteria to find nutrient hotspots in a dilute environment. Therefore, there is likely to be a high risk/reward trade-off between retaining motility or not during starvation.

To investigate the strategy adopted by copiotrophic marine bacteria, the authors selected 26 isolates (18 species) and measured their motile fraction and swimming speed just before and after carbon starvation (up to almost 2 days). This approach is a major strength of the paper, as motility studies are often limited to either (i) different isolates of the same species (with more homogeneous behaviors), (ii) only a few species, or (iii) laboratory type strains.

Interestingly, rather than a continuous behavioral landscape, they observed two distinct poles: either cells 'persist' and retain flagellation for long periods of time, or they rapidly lose their flagella. They suggest names for these two categories and propose a genome-based classifier with reasonable performance against phenotypic determination to assign a marine species to one of the two categories. While I will have comments on such a classification, I note here that in microbiome fields it is much easier to simply highlight diversity than to propose overarching categories in a reasonably convincing way, as the authors do here.

The authors then extend their work by noting that the distribution of these strategies (as detected by their classifier in the GOMicrobiome) is not homogeneous, suggesting that local environmental constraints shape the precise risk/reward trade-off.

Overall, the writing is clear, the methods and results are extremely well described, and the discussion is of high interest. I would like to thank the authors for making it such an easy and insightful read. The supplementary material is also of high quality and interest, with careful considerations on numerous topics, and further compensates for the need for simplifications in the main text (in part imposed by the heterogeneous dataset of isolates).

We thank the Reviewer for their comments and evaluation. This has helped us to improve the manuscript.

Major remarks

(1) Authors provide an excellent technical summary of the experiments done to support their claims, in Figure S2C. I notice that two strains only have one experiment (replicate), and these two strains show a very low motility fraction even before carbon starvation. Given the important variability observed in motile fraction even in showcase examples (Fig.2B), it seems to me like insufficient data.

We thank the Reviewer for pointing this out and we have performed more tracking experiments with strains D2M19,12G01,5F79 and ZF211, such that there are at least 2 biological replicates per strain. We have adapted the figures and tables with this information. Apart from these numerical changes, none of the statements or conclusions were affected by these additional data.

(2) Methods for tracking: a. (l. 386) “Bacteria were observed within 15 minutes after sampling.” Cells can stay up to 15 minutes inside the closed motility chamber before an acquisition is done, somewhere in that chamber. Where? In the bulk or along glass surfaces? The movie suggests the first option, but the Methods should mention such information. More importantly, I am concerned about biases induced by the time for a non-negligible part of the motile population to accumulate at surfaces. Bacteria tend to remain swimming along surfaces once they reach them, exiting occasionally (especially due to turns): altogether, this usually biases bulk acquisition (made several minutes after injecting the cells) against fast low-turning-rate bacteria and will also bias (lower) the bulk motile/non-motile ratio. Did the author assess such biases? I believe it could explain part of the variability they observe. How would that impact their conclusion?

We have performed additional experiments that reveal that such bias is minor in comparison to strain-strain differences. To probe the effect described by the Reviewer, we performed tracking experiments on 24-h starved cultures of two limokinetic strains (YB1 and 3B05) and one limostatic strain (ZF270) at different times within a chamber, by acquiring a video at mid plane (the standard z-position for our starvation experiments — we now added this information to the methods section, LL 465) and one at the bottom surface. We analyzed these videos using the same protocol as the other data. This new data has been added as Supplementary Figure S1 (here reproduced as Figure R2).

Figure R2: the fraction of motile cells (A), the average speed of motile cells (B) and the reorientation frequency (C) of limostatic strain ZF270 (orange) and limokinetic strains 3B05 (green) and YB1 (blue). The grey area indicates the experimental time window for observations used in the manuscript (15 minutes) that starts immediately after the chamber is closed. For each case, observations are made at midplane (‘bulk’, solid lines) and at the bottom surface (‘surface’, dashed lines).

From these results, we note that:

- For none of the three strains we observed a substantial reduction in motility at midplane compared to the surface. No motility was detected for the limostatic strain (ZF270), neither at the surface nor in the bulk. Thus, we can rule out that limostatic strains may have appeared immotile because motile cells had “escaped” to the surfaces, indicating that the dichotomy is robust to this potential artifact.
- Motility at midplane is relatively stable over time, indicating that sampling time is not the origin of the observed dichotomy. The average fraction of motile cells, velocity and reorientation frequencies of YB1 and 3B05 remain well separated, indicating that the

variation in motility parameters among limokinetic strains (see Table S1) is not dominated by variation in sample timing.

- For limokinetic strain YB1, the fraction of motile cells and their speed were typically higher at the surface than in the bulk. In contrast, for limokinetic strain 3B05 the fraction of motile cells was lower at the surface. The fact that YB1 seems to associate more strongly with surfaces agrees with the Reviewer’s intuition, based on YB1’s faster speed and less frequent reorientations. However, as noted above, the surface association does not lead to depletion of the motile bulk population.

In summary, these data show that during the first 15 minutes of acquisition, the motile fraction, swimming speed and turning frequency indeed exhibit some variation, but the differences between the two groups of strains are robust. Most importantly, the motile fraction of the limo-static strains remains zero or near-zero, regardless of observation at the surface or midplane. The dichotomy is hence not affected by this source of variation.

We have added this information in the Methods section (LL 438):

Bacteria were observed within 15 minutes after sampling (**their motility parameters were relatively stable during this period of time, Fig. S1c**).

b. (ll.417-418) “Cell tracking was performed [...] after removing the background from each 418 image by subtracting the median image computed over the entire video.” If using a background correction on the whole movie, aren’t the authors removing a part of the non-motile cells? The described method takes the median value of each pixel over ~ 14 s (25-30 fps, 418 frames), a duration I do not expect to be sufficient for a non-motile cell to diffuse enough to not be erased by the background correction. How is that not biasing the motile fraction computation?

The “418” must be a misunderstanding (it is likely the line number in the manuscript file that has been copied with the sentence). We use the full 30 s length of the video for background correction, corresponding (at 25 fps) to 750 frames.

A calculation based on diffusion of non-motile bacteria suggests this time is sufficient to prevent erosion of non-motile cells. The diffusion of a non-motile bacterial cell is approximately $0.5 \mu\text{m}^2/\text{s}$ (Nossal and Chen, 1972) (using the Stokes-Einstein relation for the diffusion coefficient of a sphere, $D = kT/6r$, this corresponds to an equivalent sphere radius of $r=0.5 \mu\text{m}$ diffusing in water at room temperature). Over 30 seconds, this means a 2D diffusion distance of nearly $\text{RMS}=\sqrt{4Dt} = 8 \mu\text{m}$ (or 25 pixels at 20X), exceeding the dimension of the bacterium several-fold. Hence, this calculation indicates that displacement by diffusion over the timescale is sufficient to prevent erosion of non-motile bacteria.

To further investigate potential biases of background detection we selected a video with very little drift (any drift present would separate non-motile cells even more clearly from the background) for further analysis. We have added a representative video consisting of the raw and corrected images displayed side by side, which has been added to the manuscript as Supplementary Video S11. (For clarity we zoomed in on an area of 900 by 900 pixels). This video shows that the background correction properly removes any artifacts of the optical path, but non-motile cells remain properly visible. To quantify the effect the correction has on the optical prominence of a bacterium, we compared the ratio of the intensity peak height (normalized to the average pixel value) of the particle in both the raw and background-corrected images. We found an average peak ratio of 0.99 (95 % CI = [0.96,1.04]), indicating that particle intensity peaks are only very slightly dimmer than in the uncorrected image. These results have been

Figure R3: A: Example non-motile cell of strain F3R08 without background correction. B: intensity profile along the red line in panel A for both the raw data (blue) and background-corrected data (red). C: The normalized maximum intensity of raw and background-corrected images. The average value is 0.99, indicating that the background correction only marginally affects peak intensity.

added to the manuscript as Supplementary Figure S1) and reproduced here as Figure R3.

(3) (ll. 82-83) “All strains exhibited a run-reverse-flick motility pattern (Supplementary Videos S5-S10), a hallmark of single-flagellated bacteria [30, 31].” While most of the video do show cases run-reverse-flick (RRF) noticeable by eye, it does not appear to me as sufficient proof to sustain an exclusive claim. Indeed, the authors will know that under a speed threshold (dependent on species) the buckling instability causing the flick every other turn will not happen, therefore leading to a run-reverse (RR) motility pattern. RR patterns can cohabit with RRF patterns. RR patterns notably have a different (lower) diffusion coefficient.

We agree that RR and RRF motility likely co-exist in our data and we therefore removed the statement here. We meant to say that the existence of some instances of RRF per strain suggests they possess a single flagellum. But instead, we performed some additional SEM experiments, directly observing the number of flagella, bringing the total of strains for which we performed SEM imaging to 4 limostatic and 7 limokinetic strains (Extended Data Figure 4a). On average, we found 97.2% of the cells possessing a zero or one flagellum. Although we did not measure every single strain, these data confirm that single-flagellation is the dominant flagellation mode in the strains we tested.

We have modified the first paragraph discussing the flagellation and added the following text (LL 132-137):

We measured the flagellation of in total 4 limostatic and 7 limokinetic strains. The average fraction of cells with 0 or 1 flagella was 97.2 %, indicating that the dominant mode of flagellation was a single polar flagellum (Extended Data Figure 4), as is the case for most other marine bacteria [40]. This is also consistent with bacteria performing run-reverse or run-reverse-flick random walks in our tracking experiments (Supplementary videos S1-S10), the hallmark of single-flagellation [41, 42].

For the difference in diffusion coefficient between RR and RRF we refer to our reply to the next point.

(4) Authors make use of an equation from Taktikos *et al.* (eqn 23 in the article) for computing the diffusion coefficient from rotational diffusion, average swimming speed and turning frequency. This equation assumes RRF motility with identical forward and backward run duration and constant speed. I find the usage of the diffusion coefficient as a quantitative metric in this manuscript very dubious due to (1) the likelihood of RR patterns coexisting with RRF patterns, (2) the potential biases in turning frequency and swimming speed mentioned above, (3) the absence of justification regarding a symmetrical pattern (likely beyond capabilities of the tracking method here), (4) the usage of a single rotational diffusion value despite strong reasons to think it would vary from species to species (e.g. different body sizes as shown in Figure S10 A). I understand the need for estimates of such value in order to develop supplementary discussion, and I appreciate it as done in Supplementary Discussion 2.2, but it is clearly a stretch to provide the diffusion coefficient of all strains at different time points (Supplementary Discussion 2.1, SFigure 7C) and comment on differences between limostatic and limokinetic strains.

We thank the Reviewer for exposing the weakness of this analysis. We largely agree with these points, but we argue that as an approximate estimation this is still useful. We consider the equation from Taktikos *et al.* (Ref. [83]):

$$D = \frac{v^2}{6} \frac{R + 4D_R}{(R + 2D_R)^2} \quad (1)$$

where D is the diffusion coefficient, v the velocity of cells and R the reorientation frequency and D_R the rotational diffusion coefficient. We agree that the rotational diffusion depends on the shape and size of cells, and assuming a single value is introducing inaccuracies. However, given that the reorientation frequencies of the marine strains we measured are high (typically 2-3 s^{-1}), this exceeds the value of a typical rotational diffusion by nearly 10-fold and the precise value of the rotational diffusion then does not matter. Hence, the expression simplifies to

$$D = \frac{v^2}{6R} \quad (2)$$

as stated in Taktikos *et al.*, this relation is actually equal to the diffusion coefficient for RR and we can use this relation to estimate the effective diffusion coefficient for both RR and RRF.

This being said, we agree that Figure S7c suggests that detailed quantitative comparisons are possible, which is not warranted. We have therefore decided to remove former Fig. S7C from the results (note we changed the figure organization and former panels S7A and B are now in Extended Data Figure 6). Removing the panel does not affect any of the conclusions presented in the main manuscript. In the supplementary Discussion 2.1, we have changed the text as follows:

The encounter rate of bacteria with larger particles scales linearly with the effective diffusion coefficient that describes the bacteria's random walk [35]. Hence, the effective diffusion coefficient of bacteria is a quantitative measure of how much space they explore. The average diffusion coefficient D of the motile fraction of a population **performing a random walk by run-reverse-flick** can be estimated from measured bacterial trajectories, according to $D = (1/6)v^2(R + 4D_R)/(R + 2D_R)^2$ [83], where v is the average velocity of motile cells (Fig. Extended Data Figure 6a), R is the reorientation frequency (Extended Data Figure 6b), and $D_R = 0.035\text{rad}^2/\text{s}$ is the rotational diffusion coefficient [86]. **Because the typical reversal frequencies (2-3 s^{-1} are much higher than the rotational diffusion D_R , the relation can be simplified to $(1/6)v^2/R$, which is also the effective diffusion coefficient for cells performing a run-reverse random walk (without flicks) [83], which is the prevailing random walk of single-flagellated**

bacteria at low velocities [41]. Hence we can use this expression to approximate the effective diffusion coefficients of the cells' random walk. The average effective diffusion coefficient of all limokinetic strains decreased from $1.5 \pm 0.5 \cdot 10^{-6} \text{ cm}^2\text{s}^{-1}$ in nutrient-rich media to $0.8 \pm 0.4 \cdot 10^{-6} \text{ cm}^2\text{s}^{-1}$ during starvation (t -test: $p = 0.003$)

Finally, we thank the Reviewer for bringing up the issue of asymmetric RR random walks (where the forward and backward velocities are different) that have typically higher diffusion coefficients than symmetric RR random walks. Although this mode of random walk seems more prominent in some lophotrichous some bacteria (having multiple flagella), more advanced tracking experiments that yield longer trajectories could reveal asymmetries for monotrichous marine bacteria, too. We have added this possibility to the Supplementary Discussion with appropriate citations, to mention another alternative strategy for cells to enhance exploration without spending more energy.

Minor remarks

(5) Could the author comment on the (less likely) possibility that the dichotomy limostatic/limokinetic they observe is just a difference in response to the sudden withdrawal of an unknown trace elements initially present in MB, rather than carbon source?

1. The starvation responses were highly reproducible if the cells were not washed into carbon starvation buffer (F/2 medium without carbon), but instead observed in stationary phase after growth in 2% marine broth (Extended Data Figure 3). This shows that the exact composition of the starvation buffer is not important.

2. Our experiments use the full F/2 medium except for silica source but with the addition of trace elements. Only the carbon source is absent.

We have added a short statement in this section (LL 126):

These results indicate that the loss of motility **is not specific to our starvation medium**, and the dichotomy is robust to differences in the mode in which carbon starvation is imposed and is primarily a species-specific trait.

(6) Is there any hint within the data for a difference in dispersal strategy between different limokinetic strains?

There is certainly variation within the limokinetic strains, but we have found no evidence for grouping this behavior further into subclasses. As can be seen from the tracking parameters in Table S1, there seem to be strains with high velocities and low reorientation frequencies during starvation, and slower moving strains with higher reorientation frequencies. Effective diffusion coefficients can be inferred from the tracking parameters obtained from our short videos, but as indicated by this Reviewer (point 4), these should not be taken for detailed quantitative comparisons, which will be better addressed using long-term tracking.

Nonconsequential remarks

(7) Export issues that could alter understanding, find the “;” symbols (in SI)

We thank the Reviewer for pointing these issues out. They have been fixed in the revised version.

(8) Bottom panels of Fig. 5 are hard to read on paper. Figure 5 has been revised.

Response to Reviewer 3

Reviewer comments in black, Author response in blue

Text citations indented, with highlighted changes in bold.

This is a fascinating study with a clear hypothesis, a suitable experimental approach, and interesting results. The text is clear and well supported by figures, videos and other supplementary material.

We thank the Reviewer for their positive assessment and for sharing their expertise.

Specific comments

1. In some videos, some cells were practically immotile while others were actively swimming. I presume each video represents one strain. Is there also a systematic dichotomy of motility and starvation response within individual strains?

We thank the reviewer for this question regarding the potential role for a within-species dichotomy of starvation responses. First, in the case of limostatic strains, which cease motility within a few hours, there seems to be no such systematic dichotomy, as the fraction of motile cells (and also the fraction of flagellated cells) decreases to zero or near-zero. Second, limokinetic strains do have a mixed population of motile and immotile cells. One explanation for this could be bet hedging, where a part of the population remains non-motile and preserves resources, while another fraction engages in search behavior. If the potential reward is large, even a small fraction of motile cells could then propagate the genotype (see the section in the Discussion, LL 353–366). Alternatively, individual cells could be switching between motile and non-motile states, as suggested partly by the fact that the fraction of flagellated cells is higher than the fraction of motile cells (Fig. 2). However, this switching, if present, would happen on timescales much longer than our tracking videos (30 seconds). Therefore, in future studies long-term tracking could unravel these scenarios. We have modified the Discussion and added a paragraph on this topic:

Even in limokinetic populations, the average motile fraction per strain during starvation varied from 0.05 to 0.55, and hence a considerable fraction of the cells were not swimming. One reason could be that limokinetic strains are hedging their bets, where the non-motile fraction of the population preserves resources while the motile fraction searches. If the potential reward is high, even a few successful searches could propagate the population [12]. Alternatively, it is possible that cells in limokinetic strains are only motile during a fraction of the time. Many species of bacteria are able to pause motility without losing flagella [71, 28, 72], likely through a ‘clutch-like’ mechanism [73, 74]. Our observation that during starvation the flagellated fraction of cells in limokinetic strains is higher than the motile fraction indicates that these strains are also capable of pausing motility. However, the duration of these pauses, and the degree to which they are present in all of our strains, cannot be reliably determined from our data and requires tracking the motility of individual cells over prolonged times. Furthermore, such experiments may reveal further variation in the motility and dispersal strategies of limokinetic strains

2. Line 16: It sounds contradictory that bacteria contribute to carbon storage by remineralizing organic carbon.

We were too succinct here. We meant to say they control carbon storage (remineralization prevents carbon sinking to the ocean floor). We modified the sentence as:

Coptiotrophic marine bacteria contribute **to the control** of carbon storage in the ocean by remineralizing organic carbon in nutrient-rich hotspots amidst oligotrophic waters

3. Line 49: “a marine particle” is so poorly defined that the calculation of carbon supply for 10^3 to 10^4 new bacteria is not meaningful.

We have now modified this, adding a citation to a mesocosm experiment with a better quantification that suggests an even higher benefit:

These hotspots provide marine bacteria with rich nutrient resources: a marine polysaccharide particle of **40 μm in diameter** can provide enough carbon for **10^4 to 10^5** new bacteria, meaning a successful colonization may lead to a manyfold increase in biomass [38, 14].

4. Line 136: “bacteria can control motility independently from flagellation” sounds misleading since cells are immotile without flagella (but in the further context it becomes clear).

To prevent any misunderstandings, we have changed it as follows:

Comparing the fraction of flagellated cells with the fraction of motile cells shows that bacteria can cease motility without losing flagellar filaments: for all strains the fraction of flagellated cells was higher than the fraction of motile cells (Fig. 2B).

5. Line 338: How does the additional energy cost by swimming faster balance out with the increasing chance/frequency of encountering a food particle?

Resistive force theory predicts that the power spent scales with the square of the swimming speed. The encounter rate scales linearly with the diffusion coefficient, which in random walks also scales with the square of the swimming speed. This suggests that the energy per encounter has no dependence on the swimming speed. However, it seems reasonable to assume that the power available for motility is also constrained by the metabolism of the starving cell (for example by the rate of biomass conversion). We therefore focused on the potential boost of reorientation frequencies, as this comes at no energetic cost but increases the encounter frequency (linearly).

6. Line 340: What is the unit of reorientation frequency?

We regret this oversight. The unit (“1/s”) has now been added (LL 407-408)

7. Line 411 etc.: How does the vertical component of swimming affect the calculated swimming speed?

In 2D tracking like the method employed in this study, the vertical component of the swimming trajectory is not measured, leading to a bias (underestimation of) in the swimming speed. It is difficult to precisely calculate the bias in the swimming speed. A full projection of all 3D trajectories onto a 2D plane results in a $\sim 30\%$ underestimate of the average swimming speed, because the vertical velocity component is ignored (Taute et al. 2015, [110]). However, if the analysis is confined to trajectories of cells in the focal plane (“2D slicing”) the bias in the absolute velocities is much lower ($< 2\%$), but this comes at the cost of reducing the amount of data. Our tracking algorithm is likely positioned between these two extremes, but

probably more towards ‘full projection’. Importantly, the differences between strains, and between time points is still meaningful as all strains were subjected to the same protocol, but the absolute numbers of swimming speed need to be interpreted with some caution. We therefore have added a statement in this regard in the caption of Table S1, which documents all velocities:

Motility parameters for each strain during growth in carbon-replete medium (g) and during carbon starvation* (s). Fraction of motile cells f , reorientation frequency R (1/s), and average swimming velocity of motile cells $\langle v \rangle$ ($\mu\text{m/s}$). Since velocities are computed from trajectories in 2D, absolute velocities can be underestimated by up to 30 % [110].

8. Fig. 1A legend: It is not clear how orange color illustrates a ceased motility in this frame. Fig. 1D legend: Mention the meaning of orange vs. blue species names. Fig. 2A: It is difficult to distinguish the purple color of flagellar filaments. Fig. 5D: Please explain the red X's.

We thank the reviewer and we have implemented these changes.

9. Videos: The names of the videos (e.g. 292010 video 336844 sgljty.mp4) do not relate to their names in the text (e.g. Supplementary Video S1).

We understand that it seems inconvenient that the file names are not the same as their label, but these file names are generated by the submission system and not under author control.

Remarks on code availability

I am not sure which code is referred to here.

Please refer to the code on the figshare repository. It can be accessed through a private link available in the online revision and submission portal and will be replaced with a public link upon publication.

Additional references in this letter

R. Nossal, S. Chen, Light Scattering from Motile Bacteria, *J. Phys. Colloques* **33** (1972) C1-171-C1-176 DOI: 10.1051/jphyscol:1972131

Kaplan, et al. *Bdellovibrio* predation cycle characterized at nanometre-scale resolution with cryo-electron tomography, *Nature Microbiology* **8**, 1267-1279 (2023)

References

- [1] F. M. Lauro, *et al.*, *Proceedings of the National Academy of Sciences* **106**, 15527 (2009).
- [2] J. P. Zehr, J. S. Weitz, I. Joint, *Science* **357**, 646 (2017).
- [3] S. Srinivasan, S. Kjelleberg, *Journal of Biosciences* **23**, 501 (1998).
- [4] M. Sebastián, *et al.*, *Frontiers in Microbiology* **10**, 760 (2019).
- [5] M. A. Moran, *et al.*, *Nature Microbiology* **7**, 508 (2022).
- [6] M. Bergkessel, D. W. Basta, D. K. Newman, *Nature Reviews Microbiology* **14**, 549 (2016).
- [7] J. Dworkin, C. S. Harwood, *Annual Review of Microbiology* **76**, 91 (2022).
- [8] N. Jiao, *et al.*, *Nature Reviews Microbiology* **22**, 408 (2024).
- [9] M. Bressac, *et al.*, *Nature* **633**, 587 (2024).
- [10] F.-Q. Wang, *et al.*, *Microbiome* **12**, 32 (2024).
- [11] F. Azam, F. Malfatti, *Nature Reviews Microbiology* **5**, 782 (2007).
- [12] B. S. Lambert, V. I. Fernandez, R. Stocker, *Limnology and Oceanography Letters* **4**, 113 (2019).
- [13] T. N. Enke, *et al.*, *Current Biology* **29**, 1528 (2019).
- [14] U. Alcolombri, *et al.*, *Nature Geoscience* **14**, 775 (2021).
- [15] G. D'Souza, *et al.*, *The ISME Journal* (2023).
- [16] T. T. H. Nguyen, *et al.*, *Nature Communications* **13**, 1657 (2022).
- [17] N. Wadhwa, H. C. Berg, *Nature Reviews Microbiology* (2021).
- [18] R. Stocker, *Science* **338**, 628 (2012).
- [19] J. M. Keegstra, F. Carrara, R. Stocker, *Nature Reviews Microbiology* (2022).
- [20] N. I. Wisnoski, J. T. Lennon, *Trends in Microbiology* **31**, 242 (2023).
- [21] T. M. Hoehler, B. B. Jørgensen, *Nature Reviews Microbiology* **11**, 83 (2013).
- [22] C. P. Kempes, *et al.*, *Frontiers in Microbiology* **8** (2017).
- [23] E. Biselli, S. J. Schink, U. Gerland, *Molecular Systems Biology* **16** (2020).
- [24] B. Ni, R. Colin, H. Link, R. G. Endres, V. Sourjik, *Proceedings of the National Academy of Sciences* **117**, 595 (2020).
- [25] M. H. Larsen, N. Blackburn, J. L. Larsen, J. E. Olsen, *Microbiology* **150**, 1283 (2004).
- [26] E. Yam, K. Tang, *Aquatic Microbial Ecology* **48**, 207 (2007).
- [27] M. A. Lever, *et al.*, *FEMS Microbiology Reviews* **39**, 688 (2015).
- [28] X. Wei, W. D. Bauer, *Applied and Environmental Microbiology* **64**, 1708 (1998).
- [29] K. Malmcrona-Friberg, A. Goodman, S. Kjelleberg, *Applied and Environmental Microbiology* **56**, 3699 (1990).
- [30] X. Zhuang, *et al.*, *Molecular Microbiology* **114**, 279 (2020).
- [31] J. L. Ferreira, *et al.*, *PLOS Biology* **17**, e3000165 (2019).
- [32] E. Ledieu-Dhehércourt, Lost in Starvation: How the interplay between physiology and ecology impacts bacterial persistence in a patchy landscape, Ph.D. thesis, MIT, Cambridge, USA (2022).
- [33] S. Stretton, S. J. Danon, S. Kjelleberg, A. E. Goodman, *FEMS Microbiology Letters* **146**, 23 (2006).
- [34] J.-B. Raina, *et al.*, *Nature* **605**, 132 (2022).

-
- [35] T. Kiørboe, *A mechanistic approach to plankton ecology* (Princeton University Press, 2008).
- [36] J. Słomka, U. Alcolombri, E. Secchi, R. Stocker, V. I. Fernandez, *New Journal of Physics* **22**, 043016 (2020).
- [37] B. Borer, I. H. Zhang, A. E. Baker, G. A. O’Toole, A. R. Babbin, *PNAS Nexus* **2**, pgac311 (2023).
- [38] M. S. Datta, E. Sliwerska, J. Gore, M. F. Polz, O. X. Cordero, *Nature Communications* **7**, 11965 (2016).
- [39] T. M. Steinum, *et al.*, *Applied and Environmental Microbiology* **82**, 5496 (2016).
- [40] E. Leifson, B. J. Cosenza, R. Miurichelano, I. C. Cleverdon, *Journal of Bacteriology* **87**, 652 (1964).
- [41] K. Son, J. S. Guasto, R. Stocker, *Nature Physics* **9**, 494 (2013).
- [42] L. Xie, T. Altindal, S. Chattopadhyay, X.-L. Wu, *Proceedings of the National Academy of Sciences* **108**, 2246 (2011).
- [43] T. T. Renault, *et al.*, *eLife* **6**, e23136 (2017).
- [44] L. Turner, A. S. Stern, H. C. Berg, *Journal of Bacteriology* **194**, 2437 (2012).
- [45] X.-Y. Zhuang, C.-J. Lo, *Biomolecules* **10**, 1528 (2020).
- [46] M. Chen, *et al.*, *eLife* **6**, e22140 (2017).
- [47] S. Chattopadhyay, R. Moldovan, C. Yeung, X. L. Wu, *Proceedings of the National Academy of Sciences* **103**, 13712 (2006).
- [48] J. R. Taylor, R. Stocker, *Science* **338**, 675 (2012).
- [49] P. S. Amy, R. Y. Morita, *Applied and Environmental Microbiology* **45**, 1109 (1983).
- [50] M. Sebastián, *et al.*, *Environmental Microbiology* **20**, 713 (2018).
- [51] J. G. Mitchell, *Microbial Ecology* **22**, 227 (1991).
- [52] B. R. K. Roller, *et al.*, *mBio* **14**, e01585 (2023).
- [53] Z. Monemhaghdoost, F. Montfort, Y. Emery, C. Depeursinge, C. Moser, *Biomedical Optics Express* **5**, 1721 (2014).
- [54] E. R. Oldewurtel, Y. Kitahara, S. Van Teeffelen, *Proceedings of the National Academy of Sciences* **118**, e2021416118 (2021).
- [55] N. Cermak, *et al.*, *The ISME Journal* **11**, 825 (2017).
- [56] W. R. Shoemaker, *et al.*, *Proceedings of the National Academy of Sciences* **118**, e2101691118 (2021).
- [57] S. J. Schink, E. Biselli, C. Ammar, U. Gerland, *Cell Systems* **9**, 64 (2019).
- [58] D. Kadouri, E. Jurkevitch, Y. Okon, *Applied and Environmental Microbiology* **69**, 7 (2003).
- [59] M. H. Rashid, N. N. Rao, A. Kornberg, *Journal of Bacteriology* **182**, 225 (2000).
- [60] K. Sekar, *et al.*, *Applied and Environmental Microbiology* **86**, e00049 (2020).
- [61] L. Bourassa, A. Camilli, *Molecular Microbiology* **72**, 124 (2009).
- [62] J. J. Morris, A. L. Rose, Z. Lu, *Redox Biology* **52**, 102285 (2022).
- [63] L. Paoli, *et al.*, *Nature* **607**, 111 (2022).
- [64] C. D. Amsler, M. Cho, P. Matsumura, *Journal of Bacteriology* **175**, 6238 (1993).
- [65] J. G. Mitchell, *et al.*, *Applied and environmental microbiology* **61**, 877 (1995).
- [66] T. Honda, *et al.*, *Proceedings of the National Academy of Sciences* **119**, e2110342119 (2022).
- [67] J. Cremer, *et al.*, *Nature* **575**, 658 (2019).
- [68] S. Gude, *et al.*, *Nature* **578**, 588 (2020).
- [69] T. Fenchel, *Aquatic Microbial Ecology* **24**, 197 (2001).
- [70] H. Grossart, L. Riemann, F. Azam, *Aquatic Microbial Ecology* **25**, 247 (2001).
- [71] Y. Yawata, F. Carrara, F. Menolascina, R. Stocker, *Proceedings of the National Academy of Sciences* **117**, 25571 (2020).
- [72] M. Grognot, A. Mittal, M. Mah’moud, K. M. Taute, *Applied and Environmental Micro-*

-
- biology* **87**, e01293 (2021).
- [73] R. Sathyamoorthy, *et al.*, *The ISME Journal* **15**, 109 (2020).
- [74] S. Subramanian, D. B. Kearns, *Annual Review of Microbiology* **73**, 225 (2019).
- [75] S. Zhu, B. Gao, *Trends in Microbiology* **28**, 785 (2020).
- [76] J. Pernthaler, *Nature Reviews Microbiology* **3**, 537 (2005).
- [77] C. Matz, K. Jürgens, *Applied and Environmental Microbiology* **71**, 921 (2005).
- [78] F. de Schaetzen, *et al.*, *Proceedings of the National Academy of Sciences* **119**, e2122659119 (2022).
- [79] R. Lewin, *Microbial Ecology* **34**, 232 (1997).
- [80] Y.-W. Lien, *et al.*, *Science* **386**, eadp0614 (2024).
- [81] S. Z. Schade, J. Adler, H. Ris, *Journal of Virology* **1**, 599 (1967).
- [82] J. Y. Yen, K. M. Broadway, B. E. Scharf, *Applied and Environmental Microbiology* **78**, 7216 (2012).
- [83] J. Taktikos, H. Stark, V. Ziburdaev, *PLoS ONE* **8**, e81936 (2013).
- [84] N. W. Frankel, *et al.*, *eLife* **3** (2014).
- [85] D. R. Brumley, *et al.*, *Proceedings of the National Academy of Sciences* **116**, 10792 (2019).
- [86] K. Son, F. Menolascina, R. Stocker, *Proceedings of the National Academy of Sciences* **113**, 8624 (2016).
- [87] D. E. Hunt, *et al.*, *Science* **320**, 1081 (2008).
- [88] O. X. Cordero, *et al.*, *Science* **337**, 1228 (2012).
- [89] Y. Yawata, *et al.*, *Proceedings of the National Academy of Sciences* **111**, 5622 (2014).
- [90] J.-H. Hehemann, *et al.*, *Nature Communications* **7**, 12860 (2016).
- [91] B.-H. Y., R. E., *Marine Biology* **141**, 47 (2002).
- [92] R. M. Welsh, *et al.*, *The ISME Journal* **10**, 1540 (2016).
- [93] R. Stocker, J. R. Seymour, A. Samadani, D. E. Hunt, M. F. Polz, *Proceedings of the National Academy of Sciences* **105**, 4209 (2008).
- [94] E. C. Kaepfel, A. Gärdes, S. Seebah, H.-P. Grossart, M. S. Ullrich, *International Journal of Systematic and Evolutionary Microbiology* **62**, 124 (2012).
- [95] G. Loy, A. Zelinsky, *Computer Vision ECCV 2002* **2350**, 358 (2002). Series Title: Lecture Notes in Computer Science.
- [96] D. B. Allen, T. Caswell, N. C. Keim, C. M. van der Wel, R. W. Verweij, Trackpy (2021). 10.5281/zenodo.4682814.
- [97] E. E. Clerc, *et al.*, *Nature Communications* **14**, 8080 (2023).
- [98] A. Savitzky, M. J. E. Golay, *Analytical Chemistry* **36**, 1627 (1964).
- [99] F. Asnicar, *et al.*, *Nature Communications* **11**, 2500 (2020).
- [100] M. Streichan, J. R. Golecki, G. Schön, *FEMS Microbiology Letters* **73**, 113 (1990).
- [101] D. P. Mesquita, *et al.*, *Water Science and Technology* **69**, 2315 (2014).
- [102] J. Kacmar, R. Carlson, S. J. Balogh, F. Sreenc, *Cytometry Part A* **69A**, 27 (2006).
- [103] S. Berg, *et al.*, *Nature Methods* **16**, 1226 (2019).
- [104] A. Hernández-Plaza, *et al.*, *Nucleic Acids Research* **51**, D389 (2023).
- [105] M. Kanehisa, S. Goto, *Nucleic Acids Research* **28**, No. 1, 27 (2000).
- [106] G. Hutson, Feature Selection Engine to Remove Features with Minimal Predictive Power (2022).
- [107] M. Majka, naivebayes: High Performance Implementation of the Naive Bayes Algorithm in R (2019).
- [108] M. Kuhn, *Journal of Statistical Software* **28**, 1 (2008). 10.18637/jss.v028.i05.
- [109] L. S. Tung Ho, C. Ané, *Systematic Biology* **63**, 397 (2014).
- [110] K. Taute, S. Gude, S. Tans, T. Shimizu, *Nature Communications* **6** (2015).

We thank all Reviewers for their time to review our manuscript and accompanying code. We thank Reviewers 2 and 3 for their positive assessment and we focus below in our response to the remaining open issues raised by Reviewer 4.

Response to Reviewer 4

Reviewer comments in black, Author response in blue

General assessment

The authors provided a fascinating analysis of bacterial strategy with a large set of experiments to evaluate their prediction. I would like to thank the editor for appointing a reviewer specifically to have a closer look at the code in our data-centric biology. I thank the authors for providing the data and the code to the reviewers as part of the review process. However, the code lacks documentation in its current state to allow for an easy replication of the code or understanding of the shared data. I understand that the main research outcome of the authors is not a software, but all items in the Code and Software Submission Checklist from Nature have been ticked when not all the information is actually provided. A couple of improvements regarding the documentation and sharing the model would benefit the transparency of the methods as well as increase the scientific outreach of the work. I answered to the short feedback questions first and in more details below.

We thank the Reviewer, Dr. Pauvert, for sharing his time and expertise. We agree that the main research outcome of this work is not the code, but rather how we have used the code to unravel a principle that guides the behavior of marine bacteria. This being said we are more than happy to improve the accessibility and we explain what we have done below.

Remarks on code

I had a particular look at the R code for the classifier as it supports a main claim of the paper and I could run the scripts after manual corrections.

We agree that this is the main code of this paper, and we are pleased to learn it can be successfully run by a different user. We have now improved the documentation and addressed the issues listed below.

Whilst the R packages are cited, there is no indication of the versions used, neither in the manuscript nor in the code nor the README. A specific suggestion to help here would be for the authors to use the `renv` package (<https://cloud.r-project.org/package=renv>) that tracks versions of packages within a project. It makes installing the set of packages way much easier for collaborators, reviewers and readers. This would also spot earlier on two issues I faced:

- a missing R dependency not specified in the code: the package `klaR`. Error in : task 1 failed "there is no package called 'klaR'"
- a remaining typo in the file `classifier/classifier/training_prediction/limostatic.R`

We appreciated the suggestions of the reviewer and we implemented them. The scripts have been reviewed and simplified where possible. A table of all `ko` features is provided as raw data, as are the original `eggnoG-mapper` search results. Most importantly, an `renv.lock` file has been provided that contains the information needed to recreate the environment, including dependencies and package versions. Instructions are provided in the `readme` file as to how the '`renv`' package can be installed and initiated to recreate the environment. In addition, a `*.rproj` file has been provided that will allow the project to be opened locally in Rstudio and a small script

to manually install the dependencies (although without versioning) are provided as a backup.

Documentation is scarce and scattered in multiple readme files (readme.rtf, README.md, readme_DHM_images.rtf, readme_20241128.rtf), making it hard for reviewers and readers to know where to start. The main readme file at the root of the code and data archive should point to others READMEs (perhaps with consistent naming and format) to guide the reviewers and readers of the type of input needed, how to install the dependencies, what are the scripts and which output files are going to be written. Many files present within the code and data archives are actually not described in the documentation/README files when entering an analysis directory.

We thank the Reviewer for this remark and we have followed the suggestion. Each folder, representing different data and analyses of the study, has its own readme file that explains all files in that folder. The root directory readme file refers the reader to the other files as suggested.

I found the readme for tracking code to be nicely written as a guide, but the authors could have provided a list of Python packages to install the dependencies (in the form of a command in the README to copy in order to create a conda/mamba environment for instance).

We have copied the required python packages and used version numbers in the readme file for the tracking, including instruction on how to install them (with a command).

There are some nestedness problems with the directories that needs to be fixed to evaluate the code. A specific suggestion to help here would be for the authors to use the here package (<https://cloud.r-project.org/package=here>) to refer to paths in the archives. For example, the script 'classifier/classifier/with_depth/depth.R' requires data that is not available in the working directory.

- L25 or L31: `load('limostatic.RData')` when it should be `load('../training_prediction/limostatic.RData')`.

- L36-38: where tsv files are expected to be under "depth" directory which had been renamed to "with_depth".

We regret these nestedness problems have caused delay of the Reviewer in being able to run the code, and we have addressed all of the above points in the updated code. The readme file now clearly states the code needs to be run from the same folder in which case all paths are correctly stated.

Precomputed data in the form of RData files is presumably shared which is a good idea. However, it is made worthless without explanation (nor in any README or in code comments), for instance loading `limokinetic.RData` in R allows to explore the following objects:

-master
-nb
-x

These objects should be documented in the README within the training_prediction to understand them and to allow reviewers and readers that could not re run the model fit to evaluate the model.

We have added an explanation of all the objects in the files in the README file of the genomic analysis.

In that regard, if the authors provided few key instructions on how to predict the studied

motility behaviour using their model, this would extend the scientific outreach of the work and help understanding the methods. For instance, making available a small demo file of KEGG orthologs to be loaded, next to a precomputed model, from which the motility behaviour could be predicted. This detailed steps would add smaller use case compared to the ones on the enteric species (that is included in the training) or the MAGs. This would allow other groups to easily use their model to assess the prediction with different strains, taxa or biomes.

Even though our code is not the main ‘product’ of this paper, we agree that applying the classifiers to other datasets is useful. We have therefore added instructions to the readme file how to apply the classifier to different strains, or train a new classifier. In addition, the code has been commented to make it more accessible.

A minor point: The authors could have excluded unnecessary files from the archives for clarity (e.g., the ones generated by Mac OSX: `._MACOSX`, `.DS_Store`)

We removed any instances of hidden files we encountered.